# Regional, multi-decadal analysis on the Loire River basin reveals that stream temperature increases faster than air temperature

Hanieh Seyedhashemi[1,2], Jean-Philippe Vidal[1], Jacob S. Diamond[1], Dominique Thiéry[3], Céline Monteil[4], Frédéric Hendrickx[4], Anthony Maire[4], and Florentina Moatar[1]

[1]INRAE, UR RiverLy, 5 rue de la Doua CS 20244, 69625 Villeurbanne, France
[2]EA 6293 GéoHydrosystèmes COntinentaux, Université François-Rabelais de Tours, Parc de Grandmont, 37200 Tours, France
[3]BRGM, Bureau de Recherches Géologiques et Minières, BP 6009 45060 Orléans Cedex 2, France
[4]EDF – Recherche et Développement, Laboratoire National d'Hydraulique et Environnement, Chatou, France

**Correspondence:** Hanieh Seyedhashemi (hanieh.seyedhashemi@inrae.fr)

**Abstract.**

   Stream temperature appears to be increasing globally, but its rate remains poorly constrained due to a paucity of long-term data and difficulty in parsing effects of hydroclimate and landscape variability. Here, we address these issues using the physically-based thermal model T-NET (Temperature-NETwork) coupled with the EROS semi-distributed hydrological model to reconstruct past daily stream temperature and streamflow at the scale of the entire Loire River basin in France ($10^5 \, km^2$ with 52 278 reaches). Stream temperature increased for almost all reaches in all seasons (mean = +0.38 °C/decade) over the 1963–2019 period. Increases were greatest in spring and summer with a median increase of +0.38 °C (range=+0.11 to +0.76 °C) and +0.44 °C (+0.08 to +1.02 °C) per decade, respectively. Rates of stream temperature increases were greater than for air temperature across seasons for majority of reaches. Spring and summer increases were typically the greatest in the southern part of the Loire basin (up to +1 °C/decade) and in the largest rivers (Strahler order $\geq$ 5). Importantly, air temperature and streamflow could exert joint influence on stream temperature trends, where the greatest stream temperature increases were accompanied by similar trends in air temperature (up to +0.71 °C/decade) and the greatest decreases in streamflow (up to -16 %/decade). Indeed, for the majority of reaches, positive stream temperature anomalies exhibited synchrony with positive anomalies in air temperature and negative anomalies in streamflow, highlighting the dual control exerted by these hydroclimatic drivers. Moreover, spring and summer stream temperature, air temperature, and streamflow time series exhibited common change-points occurring in the late 1980s, suggesting a temporal coherence between changes in the hydroclimatic drivers and a rapid stream temperature response. Critically, riparian vegetation shading mitigated stream temperature increases by up to 16 % in smaller streams (i.e. <30 km from the source). Our results provide strong support for basin-wide increases in stream temperature due to joint effects of rising air temperature and reduced streamflow. We suggest that some of these climate change-induced effects can be mitigated through the restoration and maintenance of riparian forests.

# 1 Introduction

Stream temperature is a critical water quality parameter affecting the distribution of aquatic communities (Poole and Berman, 2001; Ducharne, 2008), but its future under global change remains uncertain. As air temperature (Ta) increases worldwide due to climate change, stream temperature (Tw) is expected to follow a similar trajectory (Mohseni et al., 1999; Kaushal et al., 2010; Van Vliet et al., 2011; Isaak et al., 2012; Arora et al., 2016). Indeed, there is growing evidence that stream warming is occurring around the world, affecting freshwater ecosystems through structural and functional changes in biological communities throughout the food web (Woodward et al., 2010; O'Gorman et al., 2012; Scheffers et al., 2016). Deleterious warming effects are documented from bottom-dwelling microorganisms (e.g. Romaní et al., 2016; Majdi et al., 2020) up to macroinvertebrates (e.g. Floury et al., 2013; Bruno et al., 2019) and fish communities (e.g. Maire et al., 2019; Stefani et al., 2020). However, the paucity of long-term time series of Tw (Webb and Walling, 1996; Nelson and Palmer, 2007; Webb et al., 2008; Arora et al., 2016) has impaired the larger scale assessment of such trends, especially in light of confounding factors like hydrological changes and land use change. Hence, analyses of Tw trends, especially at large spatiotemporal scales, remain scarce (but see Kaushal et al., 2010; Orr et al., 2015; Arora et al., 2016; Michel et al., 2020; Wilby and Johnson, 2020).

To overcome the lack of Tw data, large-scale ecological studies typically use Ta as a proxy for Tw to assess the impact of climate change on the spatial distribution of aquatic organisms (e.g. Buisson et al., 2008; Buisson and Grenouillet, 2009; Tisseuil et al., 2012; Domisch et al., 2013), but Ta can over-predict changes to stream fish assemblages with climate warming compared with Tw (Kirk and Rahel, 2022). Indeed, Ta can be an imprecise surrogate for Tw (Caissie, 2006) since many landscape and basin characteristics (e.g. stream discharge (Q), streambed morphology, karst resurgences, topography, and vegetation cover) contribute to the response of Tw to climate change over time and space (Stefan and Preud'homme, 1993; Webb and Walling, 1996; Webb et al., 2008; Hannah and Garner, 2015). For instance, riparian vegetation can obstruct solar radiation, which is the dominant heat flux at air-water surface (Hannah et al., 2004; Caissie, 2006), and therefore decrease Tw response to Ta (Johnson, 2004; Loicq et al., 2018). However, while riparian vegetation shading can greatly decrease the temperature of small rivers (Dan Moore et al., 2005; Loicq et al., 2018), it has limited effects on larger rivers since the width of such rivers is large enough that only a small part of it can be shaded. Rising groundwater temperature (Taylor and Stefan, 2009; Kurylyk et al., 2013, 2014) and reduced groundwater flows (Kurylyk et al., 2014) due to climate change may further contribute to Tw trends (Meisner, 1990; Arora et al., 2016), leading to asymmetric controls (vis-à-vis Ta) on Tw (Moatar and Gailhard, 2006), especially in headwaters (Caissie, 2006; Kelleher et al., 2012; Mayer, 2012). Finally, intensification of the water cycle (Huntington, 2006), with more frequent and severe droughts (Mantua et al., 2010; Giuntoli et al., 2013; Prudhomme et al., 2014), as well as more intense and sudden floods (Blöschl et al., 2019) may decouple Ta-Tw trends, exacerbating Tw increases that will most likely be evident during low summer flows when thermal inertia and flow velocity are at their minima (Webb, 1996; Webb et al., 2008).

There is thus a clear need to improve our estimates of Tw trends to assess how stream ecosystems will respond to the climate change. Unfortunately, extrapolating trend estimates derived from short time series may lead to paradoxical results, e.g. cooling streams in a warming world (Arismendi et al., 2012). This discrepancy in short- and long-term dynamics is likely

due to confounding influences of Ta and hydrology, with implications for the persistence of specialized aquatic organisms (e.g., for cold-water biota, Arismendi et al., 2013b) and the completion of their life cycle (e.g., for diadromous fish, Arevalo et al., 2020). Hence from an ecological perspective, it will be critical to understand and deconvolve the joint influences of changing Tw and Q regimes. In the absence of more robust data sources, modeling is thus an indispensable tool in meeting these goals.

Tw models output data sets that can then be used to assess the magnitude of long-term trends, but model selection entails important considerations. For example, Tw can be estimated by developing a statistical, or stochastic, model based on multiple independent drivers (Benyahya et al., 2007), which is a common practice for large scale studies (e.g. Mantua et al., 2010; Jackson et al., 2017, 2018). However, these statistical models lack mechanism; they cannot reveal the specific energy transfer mechanisms responsible for the spatiotemporal patterns of Tw (Dugdale et al., 2017). They are also unable to predict Tw for periods other than those used for their calibration due to, for instance, the non-stationary relationship between Ta and Tw over time (Arismendi et al., 2014). Alternatively, physically-based, or deterministic, models are entirely mechanistic; they predict Tw dynamics through a heat budget, accounting for energy exchanges and effects of landscape characteristics on energy transfer (Sinokrot et al., 1995; Webb and Walling, 1997; Yearsley, 2009; van Vliet et al., 2013). Critically, such process-based models can be used not only to reconstruct past time series, but they can be used in forecasting, or in predicting Tw response to projected climate or land-use changes (Caissie et al., 2007; Dugdale et al., 2017).

Here, we used a physical process-based thermal model coupled with a semi-distributed hydrological model to understand how Tw has responded to recent climate change at a large scale. To do so, we first assessed the performance of the models against field observations over the Loire basin, France, then reconstructed daily Q and Tw over the past 57 years over the whole hydrographic network. We then used reconstructed time series to compute the magnitude of decadal trends in seasonal and annual Tw, Ta and Q. To understand the relative influences of Ta and Q (as the main hydroclimate drivers of Tw) on Tw, we compared their trends, and assessed their spatial and temporal links. Finally, we sought to understand variation in Tw trends as a function of stream size, landscape attributes, and riparian shading.

## 2  Study area

The Loire River basin is one of the largest in Europe ($10^5 \, \mathrm{km}^2$) encompassing an area with starkly contrasting HydroEco Regions ("HER"), land use/land cover, and climatic conditions (Moatar and Dupont, 2016), providing an ideal case study to disentangle the drivers of the spatial heterogeneity of trends in Tw. Mean annual precipitation (549–2130 mm), mean annual potential evapotranpiration (PET) (550–850 mm), mean annual Ta (6.0–12.5 °C) (see Fig. S1), and altitude (0–1850 m) (see Fig. 1, right) provide spatially variable controls on stream thermal regimes. Wasson et al. (2002) identified several Hydro-Ecoregions (HER) over France, representing homogeneous areas in terms of land use/land cover, geology, and climate conditions. A grouping over the Loire basin was done to identify three contrasted HERs (identified as A, B, and C in Fig. 1) that can be used to describe the spatial heterogeneity in Tw at the basin scale. Granite and basalt dominate the southern headwaters of the basin (mostly in the Massif Central, HER A), whereas sedimentary rocks occupy the middle reaches with a potential for groundwater input (HER B), followed by granite and schist in the lower reaches (HER C) (see Fig. 1, left). The percentage of

riparian vegetation cover (mean over both sides of a river bank at a buffer of 10 m, Valette et al., 2012) is more important in HER A (median=73%) and in HER B (median=68%) (Fig. 1, middle). In HER C, the presence of riparian vegetation is quite heterogeneous (median=50%).

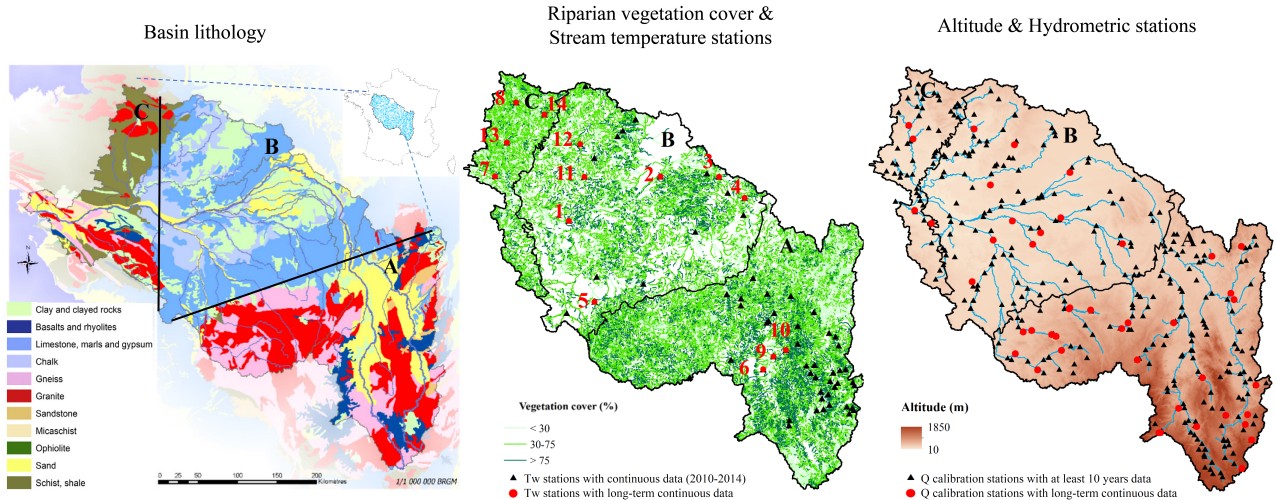

**Figure 1.** Maps of the Loire basin. Left panel: basin lighology adapted from Moatar and Dupont (2016), based on original data from BRGM (French Geological Survey). The figure in the top right corner of left panel shows the position of the Loire River basin in France. Middle panel: riparian vegetation cover (provided by Valette et al., 2012) and Tw stations. All Tw stations were used to assess the performance of the T-NET thermal model (see Table S5), but only the ones with long-term data (in red) were used to assess the T-NET thermal model accuracy for long-term trends. Complete information on Tw stations with long-term continuous data are provided in Table 1. The numbers in red in the middle panel correspond to the ID of long-term Tw stations in Table 1. Right panel: altitude (IGN, 2011) and hydrometric stations (extracted from French national Banque Hydro database: http://www.hydro.eaufrance.fr/). All hydrometric stations were used for calibrating the EROS hydrological model (see Table S2), but only stations on the French Reference Hydrometric Network (RHN) with long-term data (in red) were used to assess the EROS hydrological model performance on long-term trends (see Table S4).

## 3  Method and data

We assessed how Tw responded to climate change over the past 57 years in the Loire River basin in four steps. First, we applied physically based Q and Tw models for the period 1963–2019. Second, we assessed both hydrological and thermal models performance by comparing simulated seasonal and annual Q and Tw to data from observation stations. Third, we assessed Q and Tw long-term trends. Fourth, we analyzed how hydro-climatic changes and landscape features could explain reconstructed trends in Tw. We performed all analyses on both seasonal and annual basis, where winter is December-February (DJF), spring is March-May (MAM), summer is June-August (JJA), and fall is September-November (SON). Analyses were also performed by HER (see Fig. 1, and section 2) to look at the large-scale influence of lanscape characteristics.

## 3.1 Modeling daily Q and Tw

We used two models to calculate Tw in the Loire River basin according to the method developed by Beaufort et al. (2016). The first model is the EROS semi-distributed hydrologic model, which estimates daily Q at sub-basin outlets. The second model is the fully distributed, mechanistic temperature model called Temperature-NETwork (T-NET; Beaufort et al., 2016; Loicq et al., 2018) that uses Q from EROS and meteorological reanalysis data from Safran atmospheric reanalysis data (Quintana-Segui et al., 2008; Vidal et al., 2010) to estimate Tw at each reach in the Loire River basin. These models are briefly described below, and are detailed in Thiéry (1988); Thiéry and Moutzopoulos (1995); Thiéry (2018) for EROS, and Beaufort et al. (2016); Loicq et al. (2018) for T-NET.

### 3.1.1 EROS hydrological model principles and input data

The EROS semi-distributed hydrological model simulates daily Q at the outlet of 368 sub-basins (ranges between 40 and $1600\,\mathrm{km}^2$; mean drainage area = $300\,\mathrm{km}^2$), designed to be as homogeneous as possible with respect to land use and geology. On each of these sub-basins, the water balance is modelled by a lumped model using three reservoirs (see Fig. S2) and a routing function for propagation across sub-basins (Thiéry, 1988; Thiéry and Moutzopoulos, 1995; Thiéry, 2018). Water abstractions are not considered in EROS, and it produces natural Q. This hydrological model has already been used in several other studies including climate change impact studies (Ducharne et al., 2011; Habets et al., 2013; Bustillo et al., 2014).

EROS used daily Ta (°C), solid and liquid precipitation (mm), and reference evapotranspiration ($ET_0$, mm) to produce mean daily Q and groundwater flows (Thiéry, 1988; Thiéry and Moutzopoulos, 1995; Thiéry, 2018). Meteorological inputs were provided by the 8 km gridded Safran atmospheric reanalysis data released by Meteo-France over the 1958–2019 period (Quintana-Segui et al., 2008; Vidal et al., 2010). $ET_0$ was computed from Safran variables with the Penman-Monteith equation (Allen et al., 1998).

### 3.1.2 T-NET thermal model principles and input data

The T-NET (Temperature-NETwork) thermal model is a fully mechanistic, 1D model that simulates hourly Twi,j at distance, i, along reach, j, by solving the local heat budget. The heat budget of each reach includes six fluxes ($\mathrm{W.m}^{-2}$): net solar radiation, atmospheric longwave radiation, longwave radiation emitted from the surface water, evaporative heat flux, convective heat flux, and groundwater flux. Briefly, the model calculates the longitudinal change in Tw at time t (dTw/dx) for steady-state conditions in order to achieve thermal equilibrium (i.e. $\Sigma H_{i,j} = 0$, where H is a heat flux), while accounting for confluence mixing. Detailed information about the T-NET model principles and calculation of the six heat fluxes at the water-air and water-stream bed interfaces, and thermal propagation were provided in Beaufort et al. (2016) and Loicq et al. (2018).

The hydrographic network of the model over the Loire basin consists of 52 278 reaches delimited either by confluences of two streams or a headwater source (i.e. first order reaches) (Beaufort et al., 2016; Loicq et al., 2018). The mean reach length is $1.7\,\mathrm{km}$ and 74 % of the reaches have a Strahler order lower than 3 (see Fig. S3). To compute the six heat fluxes and the water travel time for each reach, the following input data were used:

- Meteorological variables: hourly Ta (°C), specific humidity (g.kg$^{-1}$), wind velocity (m.s$^{-1}$), shortwave radiation (W.m$^{-2}$) and longwave radiation (W.m$^{-2}$) were provided by the 8 km gridded Safran atmospheric reanalysis (Vidal et al., 2010). All reaches within a grid cell were assigned the values of the meteorological variables in that grid cell. For reaches flowing through more than one grid cell, meteorological variables were weighted by the relative length of the reach within each grid cell.

- Riparian vegetation shading: riparian vegetation is one of the major regulators of shortwave radiation. In the current study, patches of wooded area provided by the BD TOPO® (IGN, 2008) database were used as a proxy of vegetation. The vegetation species and length of each wooded patch in a buffer of 10 m were extracted for both right and left river banks (van Looy and Tormos, 2013). The vegetation density ($vc$) was then calculated as the ratio of patch length to reach length for both right and left river banks. In case of multiple wooded patches in any side of a river bank, the average vegetation density of the patches was considered. Then, the model proposed by Li et al. (2012) was used for the calculation of the dynamic shading factor ($SF$) at hourly time step. The required average tree height for both right and left river banks was estimated based on vegetation species (see Table S1).

In the presence of different vegetation species, the average tree height (m) for each side of a river bank was calculated as follows:

$$H = \frac{1}{n}\Sigma_{i=1}^{n} H_i \frac{L_i}{L} \tag{1}$$

with $H_i$ and $L_i$ the height and length of the tree patch $i$, respectively, and $L$ the reach length. Next we calculated the proportion of the river width that was shaded ($W_{\text{shaded}}$) and the dynamic shading factor ($SF$) as follows:

$$W_{\text{shaded}}(t) = \frac{H_{\text{left/right}} \times \cot \Psi(t) \times \sin \delta(t)}{W(t)} \tag{2}$$

$$SF_{\text{right}}(t) = (W_{\text{shaded}})_{\text{right}}(t) \times (vc)_{\text{right}} \tag{3}$$

$$SF_{\text{left}}(t) = (W_{\text{shaded}})_{\text{left}}(t) \times (vc)_{\text{left}} \tag{4}$$

$$SF(t) = Max(SF_{\text{right}}(t), SF_{\text{left}}(t)) \tag{5}$$

where $H$ is the average tree height (see Eq. 1), $W$ is the river width (see Eq. 7), $\Psi$ is the solar altitude angle, $\delta$ is the angle between solar azimuth and the mean azimuth of T-NET reach, and $vc$ is the vegetation density. Loicq et al. (2018) already tested this method and showed that Tw simulated by the above method was close to observed Tw. To take into account the phenology and stages of leaf growth, a coefficient corresponding to each season and transmissivity was applied to $SF$ to calculate the final shading factor: $SF_{final} = SF \times (1 - \text{transmissivity})$. The transmissivity in leafless months (Jan, Feb, Nov and Dec), months of leaf growth (Mar and Apr), and full-leaf months (May-Sep) was fixed to 0.3, 0.2 and 0, respectively, following Hutchison and Matt (1977). The shortwave radiation was lastly regulated by a factor of $1 - SF_{final}$.

- Reach streamflow: the daily Q simulated at the outlet of 368 homogeneous sub-basins by the EROS model (see section 3.1.1) were redistributed along the river network inside each sub-basin according to the reach drainage area for

informing the T-NET model at the reach scale. The ratio of sum of the lengths of all reaches upstream of a reach to the sum of the lengths of all reaches located in a sub-basin was used as a proxy for the drainage area of a reach. To have Q at hourly time step, Q was assumed to be constant over 24 hours.

– River hydraulic geometry: stream depth and width were calculated using a hydraulic model assuming a rectangular river section being constant over 24 hours (Morel et al., 2020):

$$D(t) = D50 \times \left[ \frac{Q(t)}{Q50} \right]^{f} \tag{6}$$

$$W(t) = W50 \times \left[ \frac{Q(t)}{Q50} \right]^{b} \tag{7}$$

with $f$ and $b$ being at-a-reach exponents previously predicted by climate, hydrological, topographic and land use descriptors (see Morel et al., 2020, for more details). $Q(t)$ is the daily mean streamflow provided by the EROS hydrological model. The Q50, W50, D50 (the median of Q, width and height, respectively) and the exponents were available on the Theoretical Hydrographic Network for France (RHT, Pella et al., 2012). There was about 50 % correspondence between the reaches of the T-NET and RHT networks. For the rest of them, required hydraulic geometry variables for the T-NET reaches were extrapolated from the nearest RHT reach. These river hydraulic geometries allowed us to calculate the water velocity by the ratio of $Q(t)$ to rectangular wetted cross-section. The travel time was also defined by the ratio of water velocity to reach length.

## 3.2 Model assessment and validation

### 3.2.1 Calibration and assessment of EROS hydrological model

For calibration, 352 hydrometric stations with observed Q data (see Table S2) extracted from the French national Banque Hydro database (http://www.hydro.eaufrance.fr/), were used (Fig. 1, right). Stations along the main Loire (14 stations) and Allier (11 stations) rivers were influenced by operations of 4 large dams for hydropower, flood control and low-flow management, notably through summer releases (see Table S3). Time series at these stations have been first naturalized by EDF (Électricité de France) based on variations in water storage in reservoirs due to operations based on target storage curves (Naussac, Villerest) and optimization of the hydropower energy generation for the electric grid needs (Grangent, Montpezat). EROS was then calibrated over the 1971–2018 period (which maximizes the number of available streamflow observations) against daily Q at all 352 sub-basins with at least 10 years of daily observations. Note that in the current study, calibrating EROS by performing a classical split-sample test was not possible due to the numerous and diverse gaps in streamflow observations at this large scale. Therefore, calibration was done on the complete set of available information over the past decades which is a standard approach for calibrating a hydrological model in climate change impact studies (e.g. Habets et al., 2013; Vidal et al., 2016).

The calibration aimed at optimizing all unknown parameters (soil capacity, recession times, and propagation times) through maximizing the Nash-Sutcliffe Efficiency (NSE) criterion on the square root of streamflow and minimizing the overall bias. Considering the NSE criterion on the square root of the flows provided an estimate of model performance without favoring

neither high flows nor low flows. The overall calibration criterion, C, was: C = mNSE – w.mRB where mNSE is the mean NSE, mRB is mean relative bias, and w is a weighting factor fixed at 0.05. A 3-year warm-up period (1971–1974) was discarded from the overall calibration period for the next assessments.

The calibrated model was then used to simulate streamflow over the whole 368 homogenous sub-basins in the Loire basin over the whole 1963–2019 period. Note that although meteorological variables were available from 1958 onwards, the first years (1958-1962) were discarded from the analysis to ensure the EROS model convergence.

Of the 352 calibration stations considered by EROS, 44 were part of the French Reference Hydrometric Network (RHN) described by Giuntoli et al. (2013) with long-term continuous high-quality data over the 1968–2019 period, especially for low-flows (see Table S4). These stations are shown with red points in Fig. 1, right. Seasonal and annual relative biases, together with Nash-Sutcliffe efficiency on Q, $ln(Q)$, and $\sqrt{Q}$ were computed on both 352 calibration (over the 1975–2018 period) and 44 RHN stations (over the 1968–2019 period) to provide an overview of the performance of the EROS model. Moreover, to asses EROS model ability to capture long-term trends, decadal trends (in % per decade) over the 1963–2019 period on seasonal and annual averages from the EROS simulation were compared to corresponding observed trends at each of the 44 RHN stations with long-term continuous daily data.

### 3.2.2 Validation of T-NET thermal model

T-NET does not consider the influence of impoundments on thermal energy balance, and thus produces "natural" thermal regimes. Therefore, model performance was assessed on near-natural Tw stations, which are also weakly influenced by impoundments i.e. at 67 stations with continuous daily data over the 2010–2014 period (see Fig. 1, middle, and Table S5). These stations were identified using the thermal signatures approach that allows distinguishing between natural and altered thermal regimes (see Seyedhashemi et al., 2020, for more details). Of these identified natural stations, 55 were located on small/medium streams (with distance from the source <100 km), while the remaining were located on large rivers. The mean catchment area of natural stations was $151 \, \text{km}^2$ (range=7-1342 km$^2$) for small/medium streams, and $18\,926 \, \text{km}^2$ (range=1931-57 043 km$^2$) for large rivers. Tw data at these 67 stations were provided by different organizations: EDF (Électricité de France), OFB (Office français de la biodiversité), and DREAL (Direction régionale de l'environnement, de l'aménagement et du logement), and FD (Fédération Nationale de la Pêche) (see Tables 1 and S5). Tw data provided by OFB can be downloaded from http://www.naiades.eaufrance.fr/. The rest of Tw data are not archived for the public use. Seasonal and annual absolute biases, and Root mean square error (RMSE) were assessed at these 67 stations. Note that unlike the EROS hydrological model, the T-NET thermal model does not have any free parameter, hence, it did not require calibration.

Long-term continuous data was available at 14 of the 67 near-natural stations, including 9 stations with 8-13 years data and 5 stations with 20-40 years data. These 14 stations are represented as red points in Fig. 1, middle, and listed in Table 1. The long-term evolution of annual mean Tw at these stations was shown in Fig. S4. These 14 near-natural stations with long-term continuous data comprised the validation dataset for the seasonal and annual trend assessment (see Table 1).

**Table 1.** Characteristics of the 14 long-term Tw stations. See Fig. S4 for the evolution of observed annual Tw at these stations. The ID of stations correspond to the numbers (in red) shown in the Fig. 1, middle. The coordinate system is Lambert 93.

| ID | River (Location) | Catchment area (km$^2$) | Record period | Total years | Source of data | X | Y |
|----|------------------|------------------------|---------------|-------------|----------------|-----|-----|
| 1 | Loire (Chinon) | 57043 | 1977–2019 | 43 | EDF | 493337 | 6688043 |
| 2 | Loire (St-Laurent) | 38088 | 1977-2019 | 43 | EDF | 596822 | 6737144 |
| 3 | Loire (Dampierre) | 36212 | 1977–2019 | 43 | EDF | 663009 | 6736208 |
| 4 | Loire (Bellevile) | 35172 | 1979–2019 | 41 | EDF | 691553 | 6712421 |
| 5 | Vienne (Civaux) | 5795 | 1997–2017 | 21 | EDF | 521421 | 6596684 |
| 6 | Artière (Clermont-Ferrand) | 48 | 2005–2017 | 13 | DREAL Auvergne | 710982 | 6519098 |
| 7 | Oudon (Segré) | 1342 | 2004–2014 | 11 | DREAL PDL | 410465 | 6738872 |
| 8 | Mayenne (Ambrières-les-Vallées) | 825 | 2004–2014 | 11 | DREAL PDL | 434920 | 6822096 |
| 9 | Bedat (Saint-Laure) | 419 | 2008–2017 | 10 | DREAL Auvergne | 722677 | 6533379 |
| 10 | Credogne (Puy-Guillaume) | 84 | 2008–2017 | 10 | DREAL Auvergne | 736684 | 6540613 |
| 11 | Loir (Flée) | 6215 | 2010–2017 | 8 | DREAL PDL | 511117 | 6737501 |
| 12 | Huisne (Montfort-le-Gesnois) | 1931 | 2010–2017 | 8 | DREAL PDL | 506734 | 6774430 |
| 13 | Jouanne (Forcé) | 413 | 2010–2017 | 8 | DREAL PDL | 423991 | 6776828 |
| 14 | Merdereau (Saint-Paul-le-Gaultier) | 123 | 2010-2017 | 8 | DREAL PDL | 466815 | 6808016 |

### 3.3 Time series reconstruction and assessment of long-term trends

Daily Q and Tw were reconstructed over the 57-yr period 1963–2019 using the EROS hydrological model and the T-NET thermal model. For each of the 52 278 reaches, daily time series of Ta (from the Safran reanalysis), Q (from EROS), and Tw (from T-NET) were reconstructed. Seasonal and annual averages of these three variables were considered in the trend assessment. Map of annual mean Tw, Ta and Q over the 1963–2019 period were presented in Fig. S5 with annual Tw $> 11.5\,°C$ for large rivers (annual Ta $> 10.5\,°C$ and log(annual Q)$> 1\,\mathrm{m^3/s}$).

We estimated the magnitude of trends in these time series with the non-parametric Theil–Sen estimator (Sen, 1968), and evaluated their significance with the Mann-Kendall test (Mann, 1945), commonly used in hydrological analyses (see e.g. Giuntoli et al., 2013) but also thermal analyses (e.g. Kaushal et al., 2010; Arismendi et al., 2013a; Arevalo et al., 2020). This test is robust to non-normal data, non-linear trends, and series with outliers and missing values. Trend magnitudes were reported in $°C/\mathrm{decade}$ for Ta and Tw, and in %/decade for Q (percentage of changes over whole period) to help for comparisons across the basin.

### 3.4 Hydroclimatic drivers of Tw trends

Many factors affect the spatio-temporal variability of Tw. In the current study, we considered Ta as a proxy for heat fluxes and meteorological variables, and Q as a proxy for thermal inertia and hydraulic geometries (which depends on Q, see Eq. 6 and 7).

To understand variability in Tw trends, we assessed spatial coherence and temporal synchronicity between trends in Tw and trends in Ta and Q as the main hydroclimate drivers influencing Tw. To do so, we first assessed the spatial coherence between these variables. In this regard, distributions of trends in Tw and Ta were first compared for the whole basin at the seasonal and annual scales using the non-parametric Wilcoxon signed rank test (Bauer, 1972) to determine whether median Tw trends were

greater than median Ta trends. Then, the spatial coherence across reaches in terms of difference in trends between Tw and Ta on one hand, and sign of trend in Q on the other hand, was assessed to explain the discrepancy between Tw and Ta found in the previous step with respect to Q.

Then, we assessed the temporal link between these three variables (i.e. Tw, Ta and Q). To do so, for each identified reach, we first evaluated the strength and direction of joint trends using Pearson correlation between i) Tw and Ta, and ii) Tw and Q.

Ta, Tw, and Q seasonal and annual anomalies – with respect to the 1963–2019 interannual mean – were then used to assess the synchronicity of extreme years and change-point in time series. Change-points in time series of anomalies at each reach were computed with the non-parametric Pettitt test that considers no change in the central tendency as a null hypothesis (Pettitt, 1979). Change points were reported at the 95 % confidence level.

### 3.5 Landscape drivers of Tw trends

Stream size, within individual large-scale homogeneous HERs, was selected as the first major potential landscape driver. The Strahler order of each reach was used as a proxy for stream size. Reaches with Strahler order 5–8 were combined into a single group termed "large rivers". The Spearman correlation was computed between decadal trends in Tw (i.e. across all reaches) and Strahler order. Such correlations were computed across HERs and at seasonal and annual scales to evaluate the spatial heterogeneity and seasonality. Finally, in order to better illustrate the relationship between trends in Tw and Strahler order,

median Tw trends of each group of rivers with respect to Strahler order was presented.

Second, the influence of riparian vegetation shading on trends in Tw was assessed using the daily average of the riparian vegetation shading ($SF$) simulated by T-NET. Seasonal shading was computed as the average of the daily $SF$ over each season. For this analysis, only low-order reaches – distance from the source $< 30\,\mathrm{km}$ – were considered, based on previous observations that riparian shading primarily influences Tw at this scale (Dan Moore et al., 2005; Loicq et al., 2018). Then, as

for the previous analysis for the influence of stream size, the correlation between decadal Tw trends and five levels of riparian shading (<15 %; 15-25 %; 25-40 %; 40-60 %; >60 %) was computed across HERs and seasons. Finally, median Tw trends were compared for each level of riparian shading.

## 4   Results

### 4.1   Performance of models against observations

The EROS model performed well at the 352 calibration stations at the annual scale with a median relative bias close to 0% (see Fig. S6, left). It slightly underestimated winter Q (median relative bias (across stations)=-6.27%) and spring Q (-3.47%),

and overestimated summer Q (+34.7%) and fall Q (+20.9%). The EROS model also performed well at 44 RHN stations with long-term continuous daily data at the annual scale with a median relative bias of 0.37% (see Fig. S6, right). It slightly underestimated winter Q (median relative bias (across stations)=-7.26%) and spring Q (-6.79%), and overestimated summer Q (+37.7%) and fall Q (+24.7%). The mean NSE criteria for Q, $ln(Q)$, and $\sqrt{Q}$ was $> 0.7$ for at least 75 % of both calibration and RHN stations (see Fig. S7). A good performance of the EROS model in reconstructing daily Q was also seen at three different hydrometric stations located in the upstream, the middle, and the downstream part of the Loire River basin (see Fig. S8).

No systematic bias was found for Tw modeled by T-NET at the stations located on small and medium rivers (see Fig. S9, top left). Median Tw bias ranged from -0.26 °C (in fall) to 0.8 °C (in winter). Large rivers exhibited a small Tw underestimation (see Fig. S9, top right), with a median bias ranging from -0.29 °C (in fall) to +0.15 °C (in winter), and the overall biases were still quite small across seasons (Interquartile Range (IQR)=0.4–0.7 °C depending on season). On the other hand, the median RMSE of the T-NET thermal model, for small and medium rivers, ranged between 0.52 °C (in annual) and 0.91 °C (in DJF and JJA) across seasons (see Fig. S9, bottom left). For large rivers, the median RMSE of the T-NET thermal model ranged between 0.38 °C (annual) and 1.11 °C (SON) across seasons (see Fig. S9, bottom right). Moreover, 53-83% stations (resp. 50-100%) on small and medium (resp. large) rivers had a RMSE $< 1$ °C across seasons.

Trends in observed and modeled Q were significantly correlated for all seasons (Fig. S10), with the highest correlation across stations found in spring and fall (r=0.69 and 0.71, p $< 0.05$) and the lowest correlation found in summer, which was also non-significant (r=0.17, p=0.26). The trends of both modeled and observed Q were slightly decreasing (up to -11 %/decade) for the majority of stations across all seasons, but the trend was significant for a very few of them (and mostly at the annual scale), all located in the southern part of the basin in HER A (red points of Fig. S10). Moreover, there were only a few discrepancies between estimates of trend significance in modeled and observed Q across seasons (11-18 % of stations).

Modeled and observed Tw trends also correlated significantly (see Fig. S11) across seasons, with the highest correlation in summer (r=0.94, p $< 0.001$) and the lowest correlation in fall, which was also non-significant (r=0.29, p=0.32). There was also good agreement between estimates of trend significance in modeled and observed Tw across seasons. Indeed, the trends in observed and simulated Tw were either both significant or neither was significant. Contrasting with trends in Q, trends for Tw increased for most stations across seasons, but the short period of record led to mostly non-significant trends. However, stations with long-term data showed significant increasing trends for all seasons, with the exception of winter (Fig. S11). A visual comparison of observed and modeled Tw time series and trends at stations with long-term data ($> 20$ years) indeed suggested a strong coherence and agreement between observations and simulations provided by T-NET for all seasons (Fig. 2). For the four stations along the main stem of the Loire River (Fig. 2), the greatest increase occurred in spring – +0.61 (resp. +0.71) °C/decade in observations (resp. simulations) – and summer – +0.62 (resp. +0.58) °C/decade in observations (resp. simulations). The smallest increase was found in winter – +0.22 (resp. +0.28) °C/decade in observations (resp. simulations).

### 4.2   Spatial reconstruction of long-term trends

Tw significantly increased in almost all modeled reaches for all seasons (Fig. 3, left, and Fig. S12, left). Depending on the season considered, 62 % to 80 % of reaches showed trends in the range of +0.2 to +0.4 °C/decade (i.e. +1.14 to +2.28 °C over

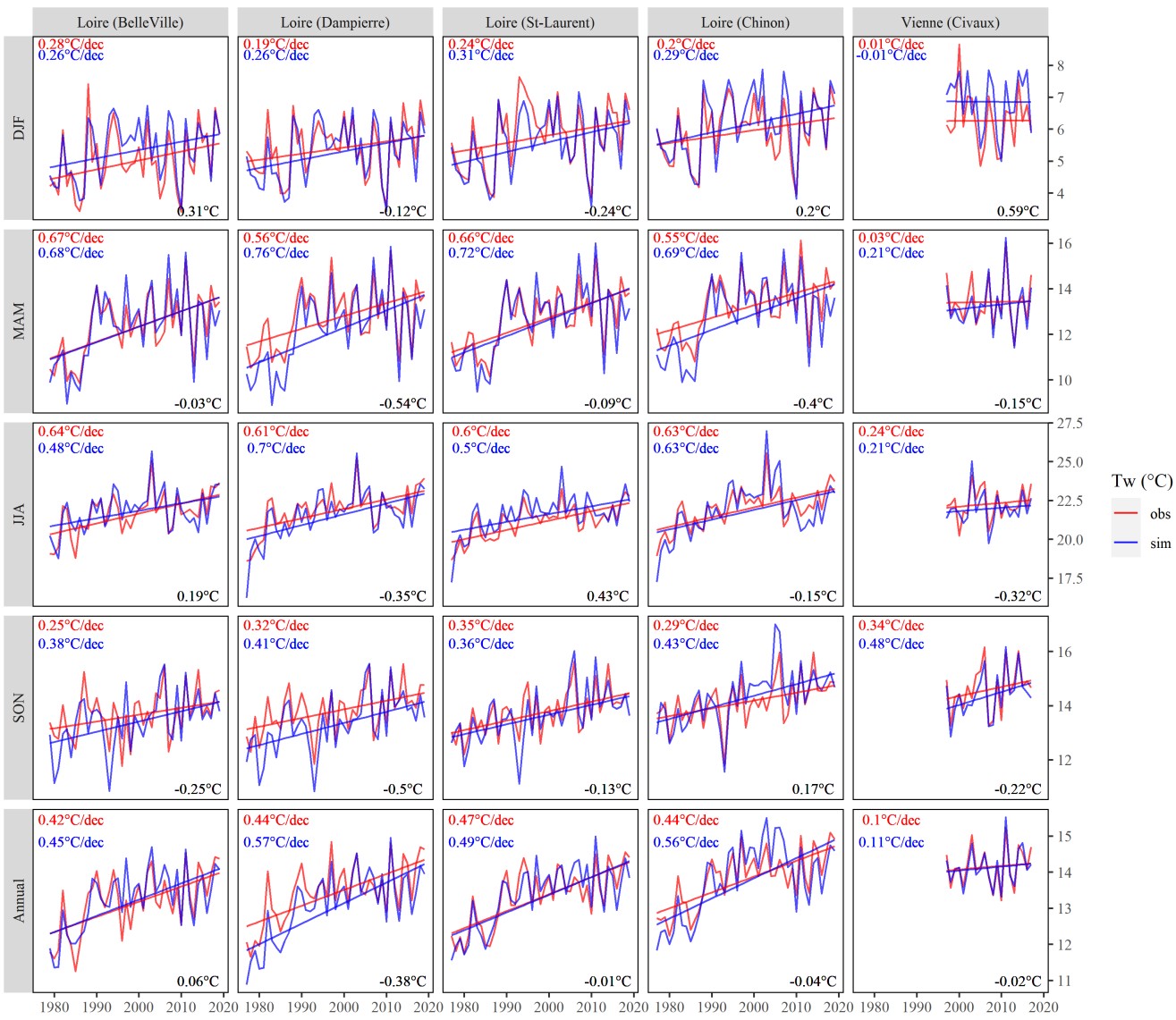

**Figure 2.** Seasonal and annual time series of observed and simulated Tw at stations with at least 20 years long-term continuous data between 1977 and 2019 (see Table 1 for more information). The red and blue solid lines show trend in observed and simulated Tw. Numbers in red and blue in the top left corner of each graph show trend magnitudes (Sen's slope) in observed and simulated Tw. Numbers in black in the bottom right corner of each graph show the mean bias of the reconstruction.

the whole 1963–2019 period, see Fig. S13). Summer Tw trends were more spatially variable than in other seasons, with more than 50 % of reaches showing values higher than +0.4 °C/decade (see Fig. S13). Such reaches were mainly located in the southern part of the basin, in HER A (see Fig. 3, left). Spring Tw trends showed a similar spatial pattern, but with lower trend values.

Likewise, Ta exhibited increasing trends for 99 % of all reaches across spring, summer, and the whole year (Figs. 3, middle and S12, middle). Values were mainly in the range of +0.2 to +0.4 °C/decade (see Fig. S13). The highest Ta trend values were found in summer and spring, when 67 % and 22 % of reaches, respectively, showed values higher than +0.4 °C/decade. Such reaches were mainly located in HER A. Non-significant trends were found over the whole basin in winter, and in the southern part of the basin in fall.

In contrast to Ta and Tw, trends in Q were highly variable in magnitude and direction across the basin and across seasons (Fig. 3, right), and most were not significant at p=0.05 (Fig. S12, right). However, significant decreasing trends were found in the southern part of the basin (HER A) in spring, summer, fall, and at annual time scale (Fig. S12). Decreasing trends were observed for a majority of reaches across seasons (66-83% of reaches), with the exception of winter (37%) (see Fig. S14). Decreasing trends could have magnitudes greater than -5 % per decade, implying a -28 % loss in Q over the whole 1963–2019 320   period (see Fig. S14).

## 4.3   Hydroclimatic drivers of Tw trends

**Spatial coherence**

The medians of Tw trends were higher than those of Ta trends for every season (p<0.001 according to the Wilcoxon signed rank test), except for summer when the median trend values for Tw and Ta were very similar, but more variable for Tw (+0.08 325   to +1.02 °C/decade) (Fig. 4). The greatest increase in median Tw over the basin was found in summer (+0.44 °C/decade).

    Overall, Tw trends were more spatially variable than Ta trends (see Figs. 3 and 4), suggesting the conditional influence of other factors like Q trends. Indeed, for the majority of reaches, Tw trends exceeded Ta trends (see Fig. 5), with the exception of summer when it was the case for the half of the reaches. The difference between Tw and Ta trends could go up to 0.38 °C/decade (i.e. up to 2.2 °C over the whole 1963–2019 period) (see Fig. 5). At the majority of such reaches (where Tw 330   trend > Ta trend) decreasing Q trends occurred coincidentally (43-94% of all reaches) for all seasons (with the exception of winter) irrespective of whether all significant and non-significant trends were considered (Fig. 6, top) or only significant trends of all three variables (i.e. Tw, Ta and Q) were considered (Fig. 6, bottom). Of these specific reaches where Tw trend > Ta trend and Q trend <0, most were located in HER A (see Figs. S15 and S16). Moreover, wherever Tw trends < Ta trend, increasing Q trends occurred coincidentally (see dark blue bars in Fig. 6), suggesting again the influence of Q trends as a possible 335   explanation for discrepancy found between Tw and Ta trends.

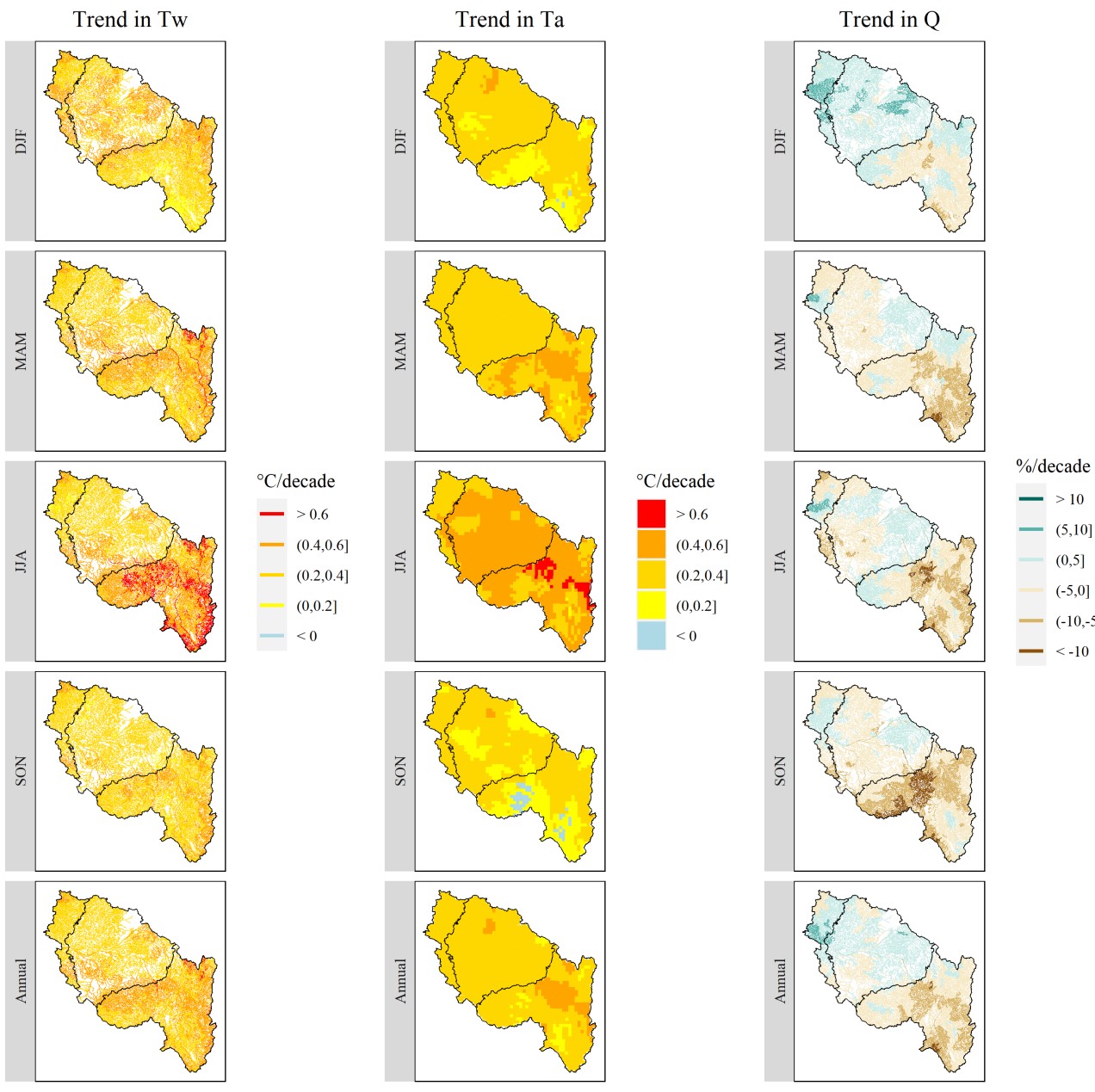

**Figure 3.** Spatial variability of trends in seasonal and annual Tw, Ta and Q over the 1963–2019 period, based on the Sen's Slope estimator. Solid black lines show the Hydro-Ecoregion (HER) delineation (see Fig. 1). The statistical significance of these trends is given in Fig. S12.

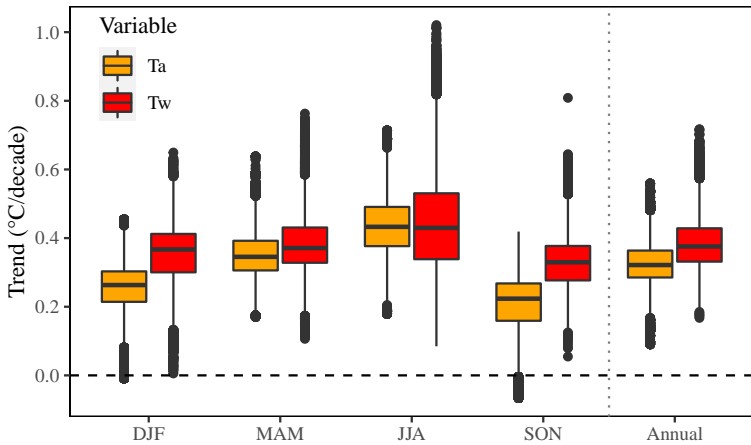

**Figure 4.** Distributions of seasonal and annual trends in Tw and Ta for all 52 278 reaches over the 1963–2019 period. Sen's slope is used as trend value estimate (see text). This representation includes both significant and non-significant trends.

## Temporal synchronicity

We observed strong positive correlations between seasonal and annual averages of Tw and Ta across seasons (r=+0.72 to +0.82 depending on the season; see Table 2). We further observed strong negative correlation between summer Tw and Q time series (r=-0.40).

**Table 2.** Pearson correlation over the 1963–2019 period between seasonal and annual Tw and Ta and/or Q time series, averaged over all reaches. Percentages in brackets show the proportion of reaches with a significant correlation at the 95 % confidence level.

| Season | Tw and Ta | Tw and Q |
|--------|-----------|----------|
| DJF | +0.73 (100%) | +0.52 (94%) |
| MAM | +0.78 (100%) | -0.02 (25%) |
| JJA | +0.82 (100%) | -0.40 (79%) |
| SON | +0.72 (99.6%) | -0.01 (19%) |
| Annual | +0.83 (100%) | -0.01 (22%) |

Annual anomalies of Tw, Ta, and Q exhibited variable patterns, with Tw and Ta generally increasing and Q remaining relatively stationary (Fig. S17). Tw anomalies were generally more variable than Ta, especially in summer, but both time series appeared to exhibit synchronous behavior. Change-point analysis supported this visual observation, where change-points in seasonal and annual averages were largely coincident across these time series (Fig. 7). Tw and Ta anomalies exhibited clear negative-to-positive change-points in the late 1980s at nearly all reaches, with median values increasing by approximately +2 °C

after the change-point. These change points were observed in all seasons, but were most pronounced and synchronous around

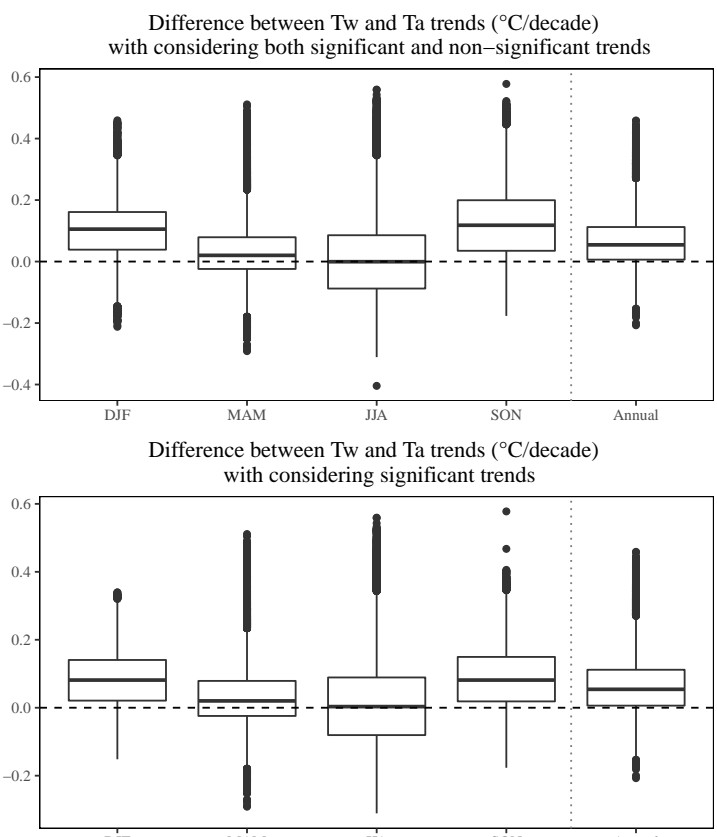

**Figure 5.** Difference between Tw and Ta trend at each reach in °C/decade for the whole 52 278 reaches.

1988 in spring and summer. The change-points detected in winter Tw and Ta time series were less concomitant, occurring mostly in the early 1990s (1992 and 1993) for Tw and in the late 1980s (1986-1989) for Ta. The fall change-points were distributed between 1980 and 1994 for both Tw and Ta. The significant change-points in seasonal Q time series were detected for a substantially smaller proportion of reaches, e.g. less than 40 % of reaches for spring and summer. The majority of these

350 reaches were located in HER A across seasons (66-86 % of such reaches), with the exception of winter (49 %) (see Fig. S18). In spring and summer, they occurred in the late 1980s, similarly to Tw and Ta. Conversely, the significant change-points detected in other seasons were much more scattered in time, probably due to the high interannual variability of Q.

Critically, the largest summer Tw and Ta positive anomalies over the study period were observed in 1976, 1994, 1995, 2003, 2005, 2006, 2015, 2017, 2018 and 2019, which corresponded to years with the largest negative anomalies in summer Q with

355 the exception of 1994 and 2018 (Fig. 8). Note that this signal was much less clear for the other seasons (see Fig. S19).

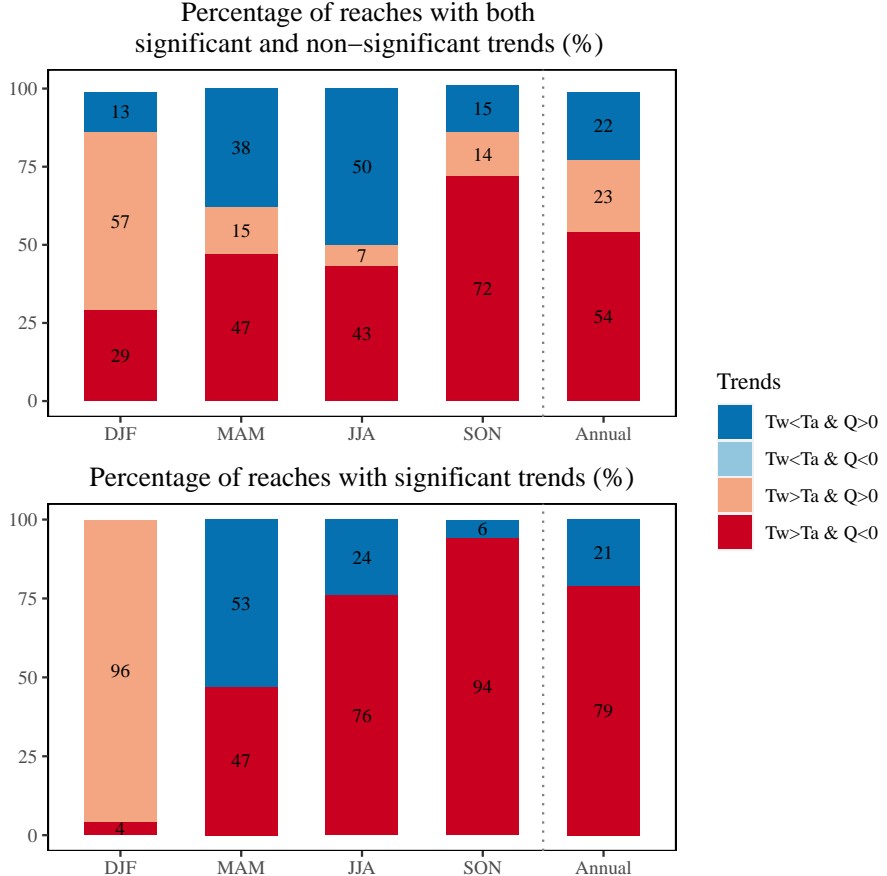

**Figure 6.** Percentage of reaches with consistent 1963–2019 trends in Tw, Ta, and Q, categorised with respect to two criteria: (1) comparison between Tw and Ta trends, and (2) sign of Q trend. Sen's slope is used as trend value estimate.

## 4.4 Landscape drivers of Tw trends

### 4.4.1 Stream size and HER

Strahler order was strongly ($p<0.001$) and positively correlated with Tw trends for all HERs in spring, and for HER A in summer and fall and at annual scale. HER A exhibited the highest positive correlations in spring (r= +0.32) and summer (r=+0.15) (Fig. S20). In other words, larger rivers tended to exhibit the largest increases in spring and summer Tw, especially for reaches located in the HER A. There, median trends in spring (resp. summer) ranged from +0.37 to +0.55 °C/decade (resp. from +0.49 to +0.64 °C/decade) from small streams (Strahler order 1) to large rivers (Strahler order $\geq$ 5).

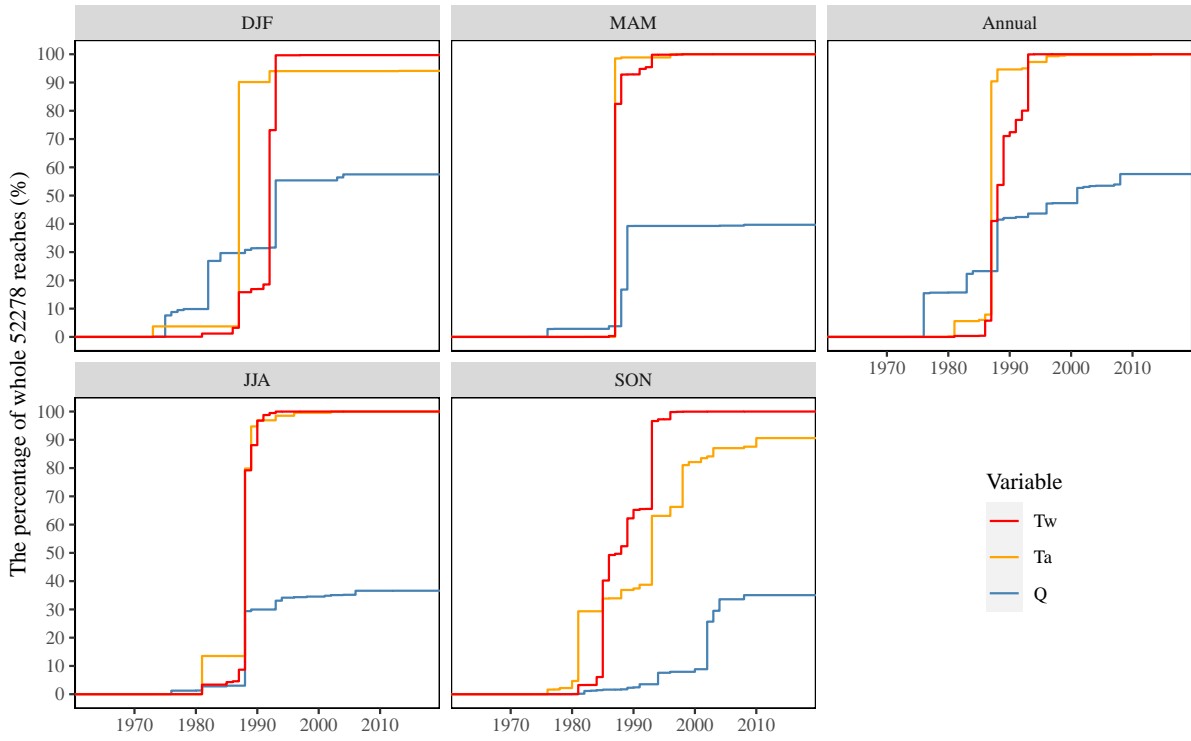

**Figure 7.** Change-point in Tw, Ta, and Q time series at the seasonal and annual scales, plotted as a proportion of reaches experiencing a shift in a given year. Only the first change-point detected at the 95% confidence level is considered, and non-significant change-points were removed, leading to curves not reaching 100 %.

### 4.4.2 Riparian shading and HER

For small streams, i.e. reaches located closer than $30\,\mathrm{km}$ from the source, the shading factor ($SF$) and trends in Tw were significantly (p<0.001) and negatively correlated in HER A in all seasons, as well as in HER B and C in spring and in HER B in fall (Fig. 9). Indeed, across seasons, the highest negative correlation was found in HER A (r=-0.56 to -0.37 depending on the season). Unsurprisingly, the mitigating effect of shading on trends in Tw for small streams was observed for all HERs in spring, and only for HER A in summer and, to a lesser extent, in fall and winter. The median spring Tw trend was mitigated by $0.12\,^{\circ}\mathrm{C}/\mathrm{decade}$ from sparsely shaded reaches ($SF < 15\%$) to highly shaded reaches ($SF > 40\%$). For summer Tw in HER A, the median trends was mitigated by $0.16\,^{\circ}\mathrm{C}/\mathrm{decade}$ from the lowest shaded reaches to the highest shaded reaches.

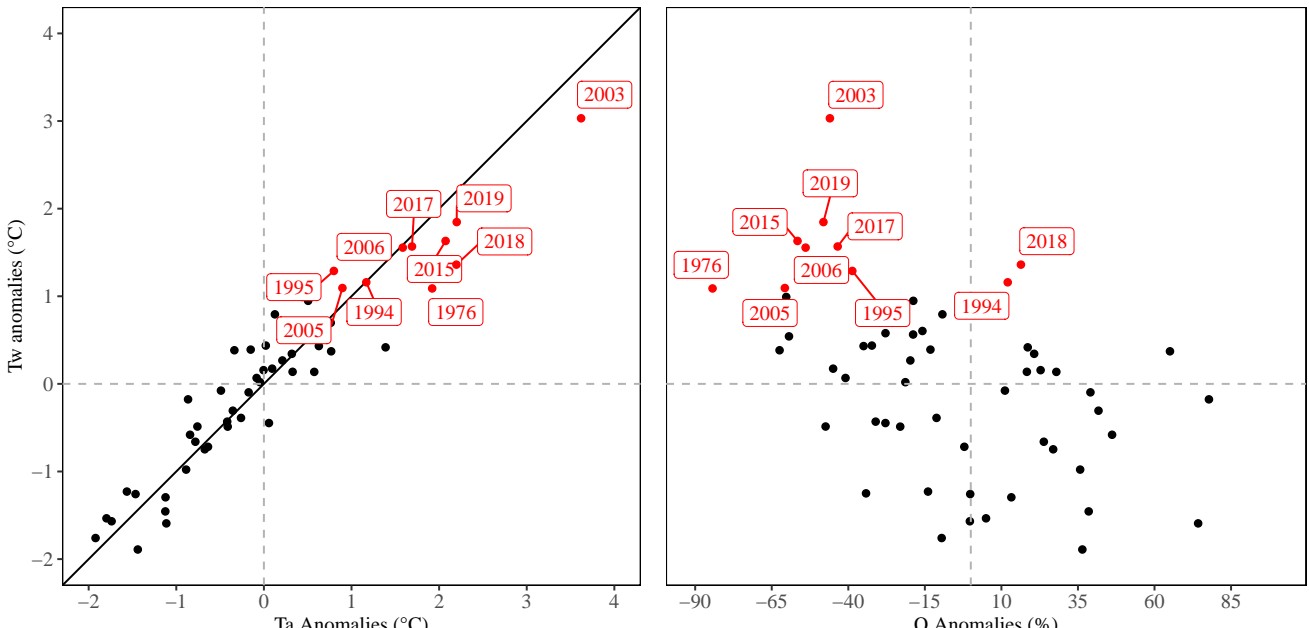

**Figure 8.** Relationship between summer anomalies in Tw on one side and summer Ta and Q anomalies on the other side. Individual years are identified from their median values across all reaches. Years with the highest anomalies in Tw ($> +1\,°C$), and corresponding anomalies in Ta, and Q are identified in red.

## 5   Discussion

### 5.1   Quality and suitability of simulated Tw and Q

Although some biases were observed for both Q (Fig. S6) and Tw (Fig. S9), we found significant correlations between modeled and observed trends in seasonal and annual Q, with the exception of summer (Fig. S10), and Tw, with the exception of fall
(Fig. S11). The low correlation value found in summer Q (Fig. S10) originated from poor simulation at very few stations all located in HER B. Two of these stations gauged catchments where numerous small ponds were found and the highly decreasing observed trends might be due to the increasing evaporation from these ponds which were obviously not included in the EROS hydrological model. This was also true in a lesser extent for the other hydrometric station, in which a canal followed a large part of the course of the river and might play a similar role with respect to summer evaporation trends. Apart from these
specific stations, in summer, a coherence as good as in other seasons was found between trends in simulated and observed Q. Moreover, the spatial pattern in simulated Q trends, with significant decreases in the southern headwaters, was consistent with observations at the high-quality reference hydrometric stations (Giuntoli et al., 2013, their figure 5).

The low correlation between simulated and observed Tw trend found in fall (Fig. S11) originated from two stations with 8-13 years Tw data while such correlation was really good at stations on the Loire and Vienne rivers with longer (20-40 years)

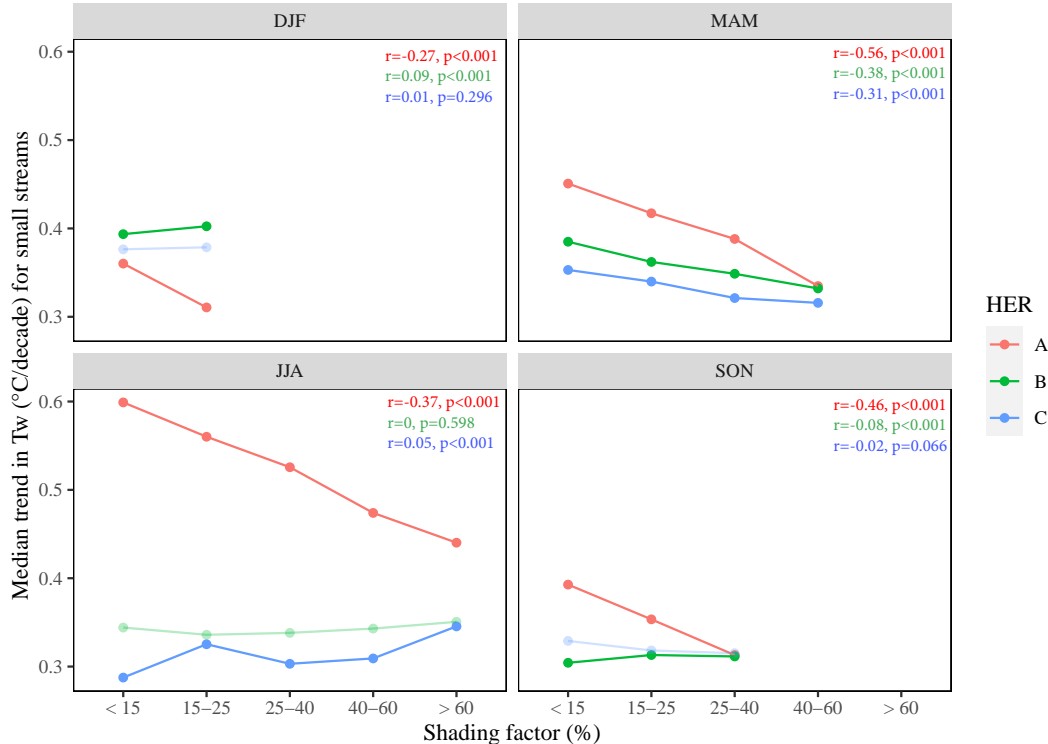

**Figure 9.** Relationships between shading factor and median trends in Tw over the 1963–2019 period for small streams, by HER and by season. Note that some shading factor classes are not observed in fall and winter. Correlations and associated p-values are shown on the top-left corner of each graph, and significant relationships at the 95 % confidence level are identified by full solid lines.

Tw data. Therefore, poor correlation in fall could be due to insufficient Tw data at these two stations. Moreover, consistent with Moatar and Gailhard (2006) and Arevalo et al. (2020), we found no trend (p>0.05) in both observation and simulation at Loire (Dampierre) in winter.

## 5.2   Comparison with Tw trends in other European basins

T-NET simulations over the 1963–2019 period showed significantly increasing trends in Tw for almost all reaches over the
Loire basin across seasons (see Fig. 3, left, and Fig. S12, left), with an increase of +0.38°C/decade on average at the annual scale. To the best of authors' knowledge, the present study was one of the few studies used modeled Tw to investigate past trends at a large scale (but see Van Vliet et al., 2011; Isaak et al., 2012, 2017). Table 3, summarizing recently published findings based on observations, demonstrate that the present results are consistent with trends observed in other European basins with clear increases in Tw over the recent decades. It also shows that the much larger scale and finer spatial resolution of the current
study clearly stands out as unique. Although start year, end year and length of the study period can have a significant influence

on trend estimates and trend detection (Arora et al., 2016), comparing trends with other studies conducted over different periods gives us a comprehensive view upon the overall magnitude of changes in Tw and possible related drivers.

Global-scale stream temperature modelling suggests trends in annual averages ranging from +0.2 to +0.5°C/decade over France (Wanders et al., 2019), which is consistent with our findings (mostly in the range of +0.2 to +0.4°C/decade, Fig. S13).
We found more pronounced trends in spring and summer, which was also found in other parts of Europe (e.g. Kędra, 2020; Arora et al., 2016; Michel et al., 2020). Considerable discrepancies were also found between Tw and Ta trends across seasons for the majority of the reaches (see Figs. 3, and 5), which is a common finding for other sites around the world (Arora et al., 2016; Wanders et al., 2019). This highlights that changes in Ta may not be the only driver of changes in natural Tw.

## 5.3 Drivers and spatial patterns of trends in Tw

Consistent with our findings (see Fig. 5), Moatar and Gailhard (2006) found Tw increased faster than Ta in spring and summer and at the annual scale for all 4 stations on the Loire River. Arora et al. (2016) also found Tw trends > Ta trends in summer. In Switzerland, Michel et al. (2020) described an increase of $+0.33 \pm 0.03$°C/decade in Tw, resulting from the joint effects of an increase in Ta ($+0.39 \pm 0.14$°C/decade) and a decrease in Q ($-10.1 \pm 4.6$%/decade) over the 1979–2018 period. In contrast with our results, they found Tw trends lower than Ta trends due to influence of snow melt and glacier melt in Alpine catchments. Consistent with their findings, Orr et al. (2015) also found Tw trends < Ta trends in UK. They suggested the such difference between Tw and Tw trends could be as a result of different processes driving Tw. In the current study, we found spatial coherence between trends in Tw and trends in Ta and Q. Indeed, the greatest increases in Tw (up to +1 °C/decade) were predominately located in the southern part of the basin, in HER A (Massif Central) where a greater increase in Ta (up to +0.71 °C/decade) and a greater decrease in Q (up to -16 %/decade) occurred jointly. We also found, at the majority of reaches where Tw trend>Ta trend, decreasing Q trends occurred coincidentally for all seasons (with the exception of winter) (Fig. 6).

The decrease in Q could itself be due to a significant increase in potential evapotranspiration (PET) (up to +10 %) over the whole of basin and a decrease in total precipitation (P) (up to -5 %/decade) (Figs. S21 and S22). Such trends, computed here based on variables from the Safran surface meteorological reanalysis (Vidal et al., 2010), are consistent with larger-scale studies (see e.g. Spinoni et al., 2017; Tramblay et al., 2020; Hobeichi et al., 2021). Moreover, Vicente-Serrano et al. (2019) attributed annual streamflow trends in southern France mostly to trends in precipitation and potential evapotranspiration, as opposed to irrigation and land-use changes that have additional strong effects e.g. in the Iberian peninsula. We also observed, for the majority of reaches where Tw increased less than Ta, an increase in Q occurred jointly suggesting that increase in Q or in other words increase in thermal inertia could also explain the discrepancy between Tw and Ta trends at these reaches (see Fig. 6).

A strong synchronicity between Ta and Tw anomalies was observed in the present study in the warmest years, and these years were also among those with the largest negative Q anomalies (see Fig. 8). Indeed, increase in summer Tw could be due to co-occurrence with the increase in summer Ta (average correlation: +0.82), and with the decrease in summer Q (average correlation: -0.40). These findings are consistent with those of Michel et al. (2020): average Tw-Ta correlation: +0.61, and average Tw-Q correlation: -0.66. For the middle Loire River, Moatar and Gailhard (2006) found that the increase in Ta (resp.

**Table 3.** Recent studies on Tw trends in Europe. A comprehensive review of the relevant literature published prior to 2016 can be found in Arora et al. (2016). Magnitudes with unspecified season are related to annual scale. Notice that except present study, the others used observed Tw.

| Country | Sites | Period | Rate of change (°C/decade) | Reference |
|---|---|---|---|---|
| **France** | **52 278 reaches in the Loire basin** | **1963–2019** | **+0.17 to +0.72 (Mean=+0.38)** | **Present study** |
| | | | **+0.01 to +0.65 (+0.35) in winter** | |
| | | | **+0.11 to +0.76 (+0.38) in spring** | |
| | | | **+0.08 to +1.02 (+0.44) in summer** | |
| | | | **+0.05 to +0.81 (+0.33) in fall** | |
| Austria | 18 rivers | 2010-2017 | +1.9 to +3.2 in summer | Niedrist and Füreder (2021) |
| England | 6148 sites | 2000-2018 | -0.4 | Wilby and Johnson (2020) |
| Switzerland | 31 rivers | 1979-2018 | +0.33 (±0.03) | Michel et al. (2020) |
| | | | +0.6 to +1.1 in summer | |
| Poland | 5 Carpathian rivers | 1984–2018 | +0.33 to +0.92 | Kędra (2020) |
| | | | +0.82 to +0.95 in spring | |
| | | | +0.75 to +1.17 in summer | |
| | | | +0.51 to +1.08 in fall | |
| | | | +0.25 to +0.29 in winter | |
| France | 11 stations on Loire, Vienne, Rhône, Seine, Meuse | 1980-2015 | +0.79 in spring | Maire et al. (2019) |
| Poland | 6 stations on Warta River | 1960-2009 | +0.096 to +0.28 | Ptak et al. (2019a) |
| Croatia | 6 stations on Kupa River | 1990-2017 | +0.23 to +0.796 | Zhu et al. (2019) |
| Switzerland | Rhine, Rhône, Aar, and Thur rivers | 1983-2013 | +0.27 (± 0.03) | Zobrist et al. (2018) |
| Northern Germany | 132 sites | 1985-2010 | +0.3 (±0.03) | Arora et al. (2016) |
| | | 1985-1995 | +0.69 (±0.10) in spring | |
| | | | +0.78 (±0.06) in summer | |
| | | | +0.75 (±0.09) in fall | |
| | | | +0.39 (±0.23) in winter | |
| | | | +0.81 (±0.2) | |
| | 475 sites | 2000-2010 | +0.9 (±0.07) | |
| England and Wales | 2773 sites | 1990–2006 | +0.3 (±0.02) | Orr et al. (2015) |
| Poland | Coastal rivers (Rega, Parsęta, Słupia, Łupawa, Łeba) | 1971-2015 | +0.26 to +0.31 | Ptak et al. (2016) |
| | | | +0.46 in April (the month with the highest trend) | |
| France | 4 stations on Loire River | 1976–2003 | +0.61 to +0.71 | Moatar and Gailhard (2006) |
| | | | +0.86 to +1.07 in spring and summer | |

decrease in Q) explain 60 % (resp. 40 %) of the increase in Tw. Moreover, the significant change-point in Tw, Ta and Q time series in the late 1980s has also been found in other studies in Europe (Moatar and Gailhard, 2006; Arora et al., 2016; Zobrist et al., 2018; Ptak et al., 2019b; Michel et al., 2020). Long-term observational Tw time series like that of the Loire at Dampierre also display a similar change-point.

Trends in Tw might also be explained by trends in additional drivers (not considered directly in the current study), like
shortwave radiation (Wanders et al., 2019), which is the dominant flux at air-water surface, and is notably increasing over Europe (Sanchez-Lorenzo et al., 2015). This might explain why Tw increased faster than Ta in spring, summer and fall, when no decreasing trend in Q were found (see Fig. 6).

The current study suggests that Ta and Q could exert a joint influence on Tw, based on an analysis of the spatial coherence and temporal synchronicity of these variables. Assessing causal influence of these factors on Tw trends goes beyond the scope
of this paper and is left for future research. In this regard, one could devise a formal attribution framework where one may e.g. remove trends in Q and trends in Ta alternatively in T-NET inputs.

## 5.4 Natural trends and anthropogenic influence on Tw

Natural Tw time series were used in the current study for detecting trends, as both EROS and T-NET models are used in a non-influenced set-up (see Sect. 3.1). However, anthropogenic impoundments (e.g. large dams, small reservoirs, and ponds)
influence downstream Tw regimes in a diversity of ways that depend on their structure and position along the river continuum (Seyedhashemi et al., 2020). In this regard, on the one hand, large dams, by releasing cold hypolimnetic water in summer, can lower downstream Tw (Olden and Naiman, 2010), and mitigate increasing trend in Tw (Cheng et al., 2020). Nevertheless, it is anticipated that a considerable proportion of streams regulated by large dams may still experience high temperatures and low flows under future climate change (Cheng et al., 2020). The mitigating influence of dams could be of importance for streams in
the southern headwaters of the Loire basin since this area both experienced the greatest Tw trends and gathers most of existing large dams.

On the other hand, ponds and shallow reservoirs, by releasing warm water can increase downstream Tw (Chandesris et al., 2019; Seyedhashemi et al., 2020; Zaidel et al., 2020) and exacerbate increasing trends in Tw (Wanders et al., 2019; Michel et al., 2020). The warming effect of such surface waters in the current study can be significant for streams located in lowlands
in the middle and north of the Loire River basin where most of the shallow reservoirs and ponds are located. In these streams, anthropogenically-induced trends in Tw may be greater than natural ones, and the warming process can get worse through the increasing demand for storing water in small reservoirs for irrigation. Nevertheless, the warming effect can also be local, and streams being located closer to such regulated streams may show limited to no warming (Wanders et al., 2019).

Note that although there are nuclear power plants in the Loire basin, their impacts on Tw is considered negligible according
to Moatar and Gailhard (2006) and Bustillo et al. (2014). Moreover, it was observed in the current study that the Tw trend at Belleville located in upstream of power plants and consequently, not influenced by them, had the same trend magnitude as the other three stations located downstream (see Fig. 2), showing a negligible influence of nuclear power plants on Tw trends.

## 5.5 Implications for river management and aquatic biota

The removal of riparian vegetation can increase Tw (Caissie, 2006), and changes in Tw can be even more sensitive to changes in riparian vegetation than to changes in Ta or Q (Wondzell et al., 2019). We showed that in small streams, an increase of > 25 % of riparian shading (from 15% to 40%) could mitigate the median trend in spring and summer Tw by up to 16 °C/decade (Fig. 9). Spring and summer Tw trends were more pronounced in large rivers, especially in the south of the basin, with a difference in median Tw trends of up to +0.18 °C/decade (Fig. S20), probably due to decrease in Q (up to -2 %/decade, see Fig. S23), a greater thermal sensitivity, and the absence of mitigating factors like riparian vegetation shading or groundwater inputs (Kelleher et al., 2012; Beaufort et al., 2020).

Restoring riparian vegetation and shading can therefore substantially mitigate future increases in Tw. In addition, riparian restoration may lessen the impacts of climate change on flood damage to human infrastructure, on riparian biodiversity, on ecosystem vulnerability and on changes in Q (Palmer et al., 2009; Seavy et al., 2009; Perry et al., 2015). However, riparian restoration is not an easy task since the survival, persistence, growth rate of planted trees as well as required time for thermal regime recovery under possibly severe future conditions should be studied beforehand (Perry et al., 2015). For instance, it may take between 5 and 15 years for rivers to recover their thermal regime following vegetation growth (Edmonds et al., 2000; Caissie, 2006). Moreover, the efficacy of riparian planting is also highly dependent upon the type and structure of forest stands (Dugdale et al., 2018), and this should also be considered in long-term projects.

Stream warming affects cold-water fish populations negatively at the warmer boundaries of their habitat (Hari et al., 2006). Furthermore, changes in spawning and migration timing (McCann et al., 2018; Arevalo et al., 2020), decreases in habitat availability and freshwater quality for organisms (Lennox et al., 2019), and shifts in species distribution (Comte et al., 2013) are already observed as consequences of the long-term increase in Tw. Some major changes in fish density and community structure has already been reported in large rivers over France (Maire et al., 2019) for which we also found greater trends in Tw compared to small ones. Therefore, physical process-based thermal models like T-NET can also be used to assess the various stresses on freshwater habitat sustainability due to changes in Q and Tw (Morales-Marín et al., 2019).

## 6 Conclusions

Regional trends in Tw at the reach resolution were detected and assessed by using the T-NET physical process-based model coupled with the EROS hydrological model over the Loire basin. Using model outputs across 52 278 reaches over the Loire basin, for 3 variables (Ta, Q, and Tw), and 5 time scales (seasons plus annual), we found consistent increasing Tw trends at the scale of the entire Loire River basin, regardless of the season. Critically, the rate of warming for stream temperature was in the majority of reaches higher than the rate of atmospheric warming, suggesting a decoupling of thermal trajectories linked to decreasing Q, especially in the southern part of the basin supported by observed coherent spatial and temporal patterns, and well-understood physical processes. Moreover, in the southern part of the basin, Tw trends in all seasons except winter were greater in rivers with Strahler order $\geq 5$, which we attributed to the mitigation effect of riparian shading for small rivers.

The synchronicity of extreme events of low flows and high stream temperature in the southern part of the basin will likely generate a double penalty for existing cold-water aquatic communities. However, riparian shading in small streams in the southern part of the basin may mitigate such warming. These findings underscore that air temperature alone is likely not an adequate proxy to explain stresses and shifts experienced by aquatic communities over time and space, especially in regions with more pronounced stream warming, and thus there is a need to grow and maintain Tw sensor networks. It also highlights that the physical process-based thermal models like T-NET can also be used to assess the various stresses on freshwater habitat sustainability due to changes in Q and Tw. This knowledge can help develop appropriate management strategies to conserve thermal refugia and mitigate extreme thermal events induced by climate change.

*Data availability.* The seasonal and annual Tw and Q time series at 67 stations (see Tables 1 and S5) can be downloaded from: https://doi.org/10.15454/PTY9R7. Data for other reaches in the basin and codes are available from the corresponding author, [HS], upon reasonable request.

*Author contributions.* HS: Conceptualization, Methodology, Software, Formal analysis, Writing - original draft preparation JPV: Conceptualization, Methodology, Writing - review and editing; JD: Methodology; Writing - review and editing; DT: Resources, Writing - review; CM: Resources; FH: Resources; AM: Writing - review and editing; FM: Conceptualization, Methodology, Writing - review and editing, Supervision

*Competing interests.* The authors declare that they have no conflict of interest.

*Acknowledgements.* This work was realized in the course of a doctoral project at University of Tours, funded by European Regional Development Fund (Fonds Européen de développement Régional-FEDER) POI FEDER Loire nᵒ2017- EX001784, Le plan Loire grandeur nature, Agence de l'eau Loire-Bretagne (AELB) and EDF (Hynes Team). The SAFRAN database was provided by the French national meteorological service (Météo-France). We are grateful to Electricité de France (French Electricity, EDF) for providing long-term data on observed stream temperature and naturalized streamflows. We also thank André Chandesris for his assistance in the incorporation of vegetation cover into the T-NET thermal model.

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
