# Peer review of "Regional, multi-decadal analysis on the Loire River basin reveals that stream temperature increases faster than air temperature"

_Hydrology and Earth System Sciences, 2021_

## Referee Comment (RC1)

**Review of "Regional, multi-decadal analysis reveals that stream temperature increases faster than air temperature"**

Dear Authors, dear Editor,

The submitted paper presents a large-scale modelling effort of river discharge and temperature for the Loire River basin in France. The modelling is carried out in two steps, first for discharge and then for water temperature. The models are forced with reanalysis data provided by Météo-France and validated on a large number of measurement stations. The model results are then used to analyse and discuss the trends in discharge and water temperature, the factors influencing the modelled trends in water temperature, the spatial patterns of the resulting trends, and the role of riparian vegetation and Strahler order.

This article presents an important work and a valuable contribution to his field of research. In general, the paper is well written in terms of language and has clear and nice figures. However, I have some concerns which I detail below. The first concern is the clarity, completeness, and organisation of section 3 "Method and data", where, in my opinion, some crucial information is missing. The second concern is the robustness of some of the results, in particular the validation of the model and the strength of some of the analysis performed. Indeed, the results and discussion sections tend to analyse many different aspects and perhaps only the most robust ones could be kept. In addition, since quite simple models are used, I would be more careful for some interpretation of the results.

I also regret that no data or models are shared along the article. I imagine that sharing of forcing data may not be allowed, however I would strongly encourage the addition of a "Data and code availability" section to detail how this work can be replicated. If possible, sharing with the community the time series produced by the model would also be appreciated. In the same vein, the supplementary material could be improved to provide interested readers with a more complete overview of the large number of results obtained.

Despite these concerns, this paper has the potential to bring interesting new results. I think that a revised version would be suitable for publication in HESS. Water temperature is now a rapidly expanding area of research, and with climate change underway, further research efforts are needed. This article is certainly a valuable contribution.

Please do not hesitate to contact me for further discussions.

Best regards,

Adrien Michel (adrien.michel@epfl.ch)

**Main comments:**

**1. Details of methods and data**

**1.1 Data description**

My first recommendation is to create a "Data" section (with sub-sections) where all details on forcing, validation and geographical data are grouped. Currently, the information is spread over P3L83-86, P4L97-P5L104, P5L118-121, L123-124, L125-128, P7L165-167 (I may have forgotten some). Grouping them together would make for easier reading as many details now appear in the models' description sections.

In addition, a comprehensive table of all stations used for calibration and validation should be provided in the supplementary material (SM later), with coordinates and station names, and also indicating the data provider. The coordinates should be added in Table 1. For long-term stations, one could even consider numbering them in the table and then indicating the numbers on Figure 1. Maps similar to Figure 3 and S5, but showing the annual and seasonal average of Tw, Ta and Q over the whole catchment area could be a useful addition to the SM to capture the different local conditions in the catchment. A map similar to Figure 2, middle panel, with a colour indicating the Strahler order could be added in the SM. For non-French readers, a map showing the location of the Loire basin in France (or in Europe) should be added. It could be integrated as a new panel in figure 1.

In P5L103 it is said that the Q time series are "naturalised". However, no reference or details of the procedure used are provided. This information should be included because, as you note later in section 5.4, anthropogenic disturbances are of major importance. In addition, I did not find in the document the source of the time series for water temperature. Is it the same supplier as for the discharge? Is it also "naturalized"?

**1.2 Description of the EROS model**

Only a few details are given on how the model actually works. In addition, the two main references given for the model are in French. Although it is not necessary here to describe all the details of the models, the main points should be provided to the reader. In particular, details of the mass balance should be given. Is it precipitation - evaporation, or can some of the mass be lost through deep soil infiltration? How is the water transported through the soil? Does EROS use a reservoir model? Finally, is there any routing of water into the stream network carried out in EROS or is only the release of water at the sub-catchment scale simulated? P5L109-111 suggest that routing is done in T-NET, but this is not clearly stated, nor is it mentioned in section 3.1.2. On the other hand, section 3.2.1 suggests that Q is obtained directly from EROS, which means that the routing is done in EROS.

It is not mentioned which parameters are calibrated (this should be stated), which values are tested for calibration, and which values are finally chosen. This should be stated in the SM to allow reproducibility. P5L105-107 states that: *"The calibration aimed at maximizing the Nash-Sutcliffe efficiency criteria (Nash and Sutcliffe, 1970) on the square root of streamflow and minimizing the overall bias, in order to simulate correctly the whole range of Q values".* How are the two metrics combined (NSE and bias) to assess the quality of the calibration? Why is the square root used? How many calibrations are performed, how are the calibration values chosen (pure chance or more advanced algorithm)? Again, the only non-French peer-reviewed source provided is Thiéry (1988). This paper describes a water level model for an unconfined aquifer and I imagine that EROS is a model based on this work, but rather different from this 1988 version.

There is also no mention of the other input parameters of the model. The authors describe three different HydroEco regions (HER), but there is no information on how the region influences the model parameters (and thus the model outputs).

Overall, more details about the model are needed to enable the reader to understand how the EORS model works. The main assumptions and principle of the model should be stated, as well as details on the calibration procedure and parameters (a part can be added to the SM). I have found many applications of this model in the literature; these could also be cited as application examples.

**1.3 Description of the T-NET model**

Here too, important pieces of information are missing. This is even more problematic than for the EROS model as no references are given for this model. P5L115-116 says: "*To simulate Tw, the equilibrium temperature (Te) is first computed, the temperature at which the net heat flux across the surfaces of the stream is null*". Then details of how some of the energy fluxes are calculated are given, but the above sentence is never followed by a "then" or "next". So, if I understand correctly, the temperature of the water being modelled is the equilibrium temperature? Or are you using a formulation similar to Bustillo's (2014) eq. (6). A crucial piece of information is missing here. And if only Te is used, this critical assumption and its consequences need to be discussed.

In section 3.1.1 it says that EROS is used to calculate discharge over 368 sub-catchments, whereas in section 3.1.2 52'278 reaches are mentioned. So, there are several reaches per sub-catchment and within each sub-catchment, the water supply simulated by EROS is distributed to the reaches using the drainage area? But again, how and in which model is the routing calculated? P6L156-157 clearly states that "*Q is the daily mean streamflow provided by the EROS model*". But how can this be done if there is no routing in EROS? P7L160-161 explains that the travel time is calculated in T-NET, but this information is not used for the calculation of the water temperature (at least that is my understanding), but would be mandatory information for routing.

No details are provided on how sensible, latent and groundwater heat fluxes are calculated. Equations or references should be provided. I understand that no calibration is done for T-NET, is this correct? Are there any other parameters used in the model? For example, as with EROS, how are the properties of HERs taken into account in the model? The results are discussed for the different HERs, but this discussion only makes sense if the HERs are somehow parameterised in the model.

Finally, in addition to my questions about routing, there is no information about reach-to-reach heat advection. Is there reach-to-reach heat transfer? If not, this would considerably weaken the analysis carried out in terms of the Strahler order and of the whole model in general.

To conclude this first part of the discussion, I would really encourage the authors to add a few paragraphs better detailing the data sources, the model workflow, and the main assumptions of the models. As I say below, it would also be important to indicate the limitations of the models.

**2. Model validation and robustness of the results**

**2.1 Calibration and validation of EROS and T-NET**

Firstly, details of the calibration procedure and parameters should be provided (see above). Secondly, why is no validation period used to infer the quality of the calibration? Indeed, time series are usually divided into a calibration period and a validation period. By using only a calibration period without validation, we have no information on the potential overfitting of the model at the calibration station during the calibration period. Depending on the modelling effort required, I would strongly recommend recalibrating the model over a shorter period and using a few years for validation. Furthermore, looking at Figure 3 of Thiéry (1988), we see a clear decrease in the performance of the model during the validation period.

In section 3.2.1, NSE on Q, ln(Q) and sqrt(Q) are mentioned, does this correspond to the "the mean NSE criteria for low, medium and high flows" mentioned in P9L221? Little detail is given on the quality of the calibration of Q. Indeed, only the mean NSE is given (no details on the variance), a graph of the distribution of values should be added in SM, as well as a graph showing the simulated and measured time series for some stations. In Figure S2, the bias is only shown for the 44 NHN stations, why not show all 352 stations (using a shorter time period)?

In Section 4.1, it is shown that most of the simulated summer trends are not significant and that the simulated summer trends are poorly correlated with the observed trends. However, these summer trends are used extensively later in the document (e.g. Figure 5, Table 2). Given the uncertainty around these trends over the calibration period, I doubt that they are robust enough to perform such an analysis.

For Tw, only the performance in terms of trends and biases is presented. A presentation of performance in terms of mean square error would also be informative in assessing the performance of the model as this metric is commonly used in the literature. The bias is discussed for Q and Tw. However, as the subsequent analysis is mainly trend based, and bias has no impact on trend, I am not sure of the relevance of this metric here.

Section 5.1 does not add much to section 4.1. I would like to see in these sections more discussion of where the models are underperforming and therefore the limitation of the dataset obtained. Indeed, the models used, like all models, are not perfect. Identifying the limitations and thus focusing the analysis and discussion on the part that proved to be within the radius of validity of the models is a mandatory step in modelling studies.

Finally, the part concerning riparian vegetation seems to be a new addition to the model in this article. My concern is that no validation is shown. The approach should be validated using a Tw measurement station in a shaded area and compare the model performance with and without shading. In the absence of validation and evaluation of the effectiveness and limitations of the approach, I find it difficult to proceed with the analysis.

**2.2 Link between Ta, Tw, and Q**

The comparison of Ta and Tw trends and the potential impact of Q are widely discussed. I appreciate this effort and think that there is still much to understand about the interaction between Tw and Q.

First of all, great importance is attached to the finding that Tw increases faster than Ta for most seasons and on an annual basis. This result has already been found in some regions (see e.g., Webb and Nobilis, 2007; and Arora et al., 2016), but is in contradiction with other studies (see e.g., Moatar and Gailhard, 2006; Orr et al., 2015; and Michel et al., 2020). I believe this result would merit further discussion to assess its strength.

The main factor used to explain why the trends in Tw are more important than those in Ta is the discharge. However, trends in other forcing variables should be shown. Indeed, in winter, Figures 3 and S5 show that the variable explaining why TW trends > TA is probably not

discharge (since no significant trend in discharge is found). Similarly, summer and spring show marked negative trends in discharge in some parts of the catchment, whereas these are the seasons where the trends in Ta and Tw are most similar. The addition of the seasonal discharge trend in the boxplot in Figure 4 would facilitate the analysis.

Figure 5 is used to support the hypothesis that Q is the main driver, see P11L259-263: "*Overall, Tw trends were more spatially variable than Ta trends, suggesting the conditional influence of Q trends (Fig. 4). Indeed, where Tw trends exceeded Ta trends, decreasing Q trends occurred coincidentally at the majority of reaches for all seasons – with the exception of winter – (43-72 %, depending on season; Fig. 5. Of these specific reaches where all factors converged (trend in Tw higher than trend in Ta, and decreasing trend in Q)*" (please note that there are some grammatical problems in the second sentences). There are many shortcuts here. Firstly, the larger scatter suggests that factors other than Ta have an impact on water temperature, but this does not in itself show that Q is responsible, it could be any other forcing variable. Secondly, and more importantly, Figure 5 shows the percentage of reaches where "all factors converge", which I interpret as "all trends are significant". Please clarify this point, as when comparing Figures 5 and 3, I rather understand that all trends are included in Figure 5. Including non-significant trends would be a significant bias here, as many non-significant trends are just above or just below zero and the figures are based on trend signs.

For Figure 5 to be complete, the distribution between Q>0 and Q<0 should also appear in the blue part. Indeed, the figure now shows that Tw>Ta in most cases when Q<0, however Figure 4 shows that Tw>Ta in most cases anyway. This figure could be used to show the impact of Q if, and only if, we can see that the proportion of Ta>Tw vs Ta<Tw changes if Q>0 and Q<0. Also, the number of catchments used probably differs significantly between seasons (at least if only significant trends are shown). Here I see that there may be something irreconcilable in this figure:

   - Either all reaches are kept, including those with insignificant trends, but then the figure itself will lose much of its meaning for the reason mentioned above.

   - Either only significant trends are retained, but only a small minority of reaches show significant Q trends in certain seasons, and the figure would then only show a small subset of reaches.

This question of the relationship between Tw and Q trends is not straightforward (you can look at the introduction of Arora et al. (2016) for a good review of the literature available by then) and, it is important to note, seeing a correlation between Tw trends and Q trends does not imply any causality. This is what we see in the subsection "Synchrony of annual anomalies" and in Figure 8: low Q summers correspond to high Ta summers, so it is difficult to assess which of the two, or the combination, leads to an increase in Tw. Furthermore, Figures S10 and S11 suggest that the negative discharge trend is caused by an increase in ET rather than a decrease in P. The regions concerned (see Figure 3), are those where the increase in Ta is most significant. Thus, the causal chain here appears to be Ta increasing → ET increasing → Q decreasing. So, even if Q is shown to be a factor influencing Tw, it originates (mostly) in the increase in Ta. Thus, it might be misleading to say that a decrease in Q is a contributing factor, as I think that for most readers a decrease in Q would mean a decrease in precipitation, whereas here no significant decrease in precipitation is found, and thus no impact of the precipitation regime on Tw can be assessed. The real impact of ET can be assessed by comparing the measured trends of discharge to the trends on measured precipitation in the catchment, in order to confirm if the increase in ET modelled is correct.

With all this discussion, I really question this sentence in the abstract: *"Importantly, air temperature and streamflow exerted joint influence on stream temperature trends, where the greatest stream temperature increases were accompanied by similar trends in air temperature (up to +0.71 °C/decade) and the greatest decreases in streamflow (up to -16 %/decade)".* Indeed, as discussed above the discharge decrease might just be a "side-effect" of the increase in Ta through increased ET, but since it is also the region with the highest Ta increase

(P16L229), I don't think that we can conclude that "*air temperature and streamflow exerted joint influence on stream temperature*".

Continuing with the subsection "Synchrony of annual anomalies", I do not really see the added value of the change point analysis. Furthermore, as it seems to illustrate an abrupt change in the late 1980s, I would think that a trend analysis is then not appropriate for these time series.

Going back to the Tw>Ta question more generally (sorry, all the variables are related and I had trouble organising my comments), it would be interesting to calculate trends at long-term water temperature stations to see if the same trend is observed, to reinforce the results. In addition, a map like the one in Figure 3, but showing the difference between Ta and Tw, would be very informative about the spatial distribution of catchments where Ta trends are more important than Tw and help the comparison with discharge trends.

This result receives a lot of attention in the article (see for example the title). The details (Figure 4) show that this is mainly due to the winter and autumn seasons (although the difference is statistically significant in spring, it is still really small). However, Figure S4 and the indicated R coefficient shows that the simulated trends in autumn have the lowest correlation with the measured trends (this is also stated in P9L232), so autumn is the season where we have the lowest confidence in the results. In any case, I would condition the general statement of Tw>Ta on the seasonal aspect since it is not general (and I would add this information in the title) and I would really stress the uncertainties about it. This result is interesting, and certainly deserves attention, but in my opinion not robust enough to be asserted in the way it is in the title of the paper (at least with what is currently shown). The discussion around the cause can also be enhanced.

**2.3 HER, Strahler order and riparian vegetation**

The study is complemented by an assessment of the influence of HERs, Strahler order and riparian vegetation on Tw. I have already raised some concerns about HERs (how they are accounted for in the models), and about riparian vegetation (the model is not validated for this). These two points need to be addressed in order to present the analysis. In general, all these topics are interesting, but I have the impression that they are treated only superficially and not with the necessary rigour. Moreover, there are already a lot of results presented and I think the article would still be interesting, and perhaps easier to read, if these parts were removed.

Regarding the Strahler order analysis, this brings back to the issue of the reach-to-reach heat advection mechanism that is missing in the model description. Also, showing a Strahler order map for each reach in the SM would be really informative. As mentioned in in P22L397-398, the correlation between Strahler order and Tw may be due to riparian vegetation (and not a concentration of the warming when going downstream). However, this should be analysed in more details, e.g., by looking at the correlation between Strahler order and Tw separately for different shading factor values (e.g., the categories of Figure 10).

For riparian vegetation, do you have a mechanism for why it would change the trend in Tw? I understand that adding or removing vegetation would change the absolute value of temperature, but by what mechanism would the trends be affected? If the riparian temperature sections were to be retained, this should be addressed.

To conclude this second part, I would recommend that the authors revise and strengthen the analysis of Ta vs Tw trends and the impact of Q. With the results presented, such strong statements as those in the title and abstract do not entirely hold water. I would also recommend mentioning in the title the location where the study was conducted (Loire, France). Perhaps some of my reservations stem from a lack of understanding of exactly how the models work and a better description might help to alleviate these reservations. Along with the emphasis

on the main message that I recommend, perhaps some of the "secondary analyses" could be removed from the document. I think that a more thorough discussion of the limitations of the methods and results should be provided. I have not commented in detail on all the discussion, summary and conclusion sections, but certainly some of them are relevant to my comments above.

A final comment concerns the real added value of such a major modelling effort. A significant part of France is modelled; however, I feel that some of the results could be obtained by analysing only past measurements (and getting rid of all the modelling uncertainty). The added value could come from the whole analysis of the riparian vegetation for example, but as mentioned these analyses need to be strengthened. One solution (but which would involve a lot of extra work), could be to first publish just the data set in a journal like ESSE (https://www.earth-system-science-data.net/), where all the modelling and validation aspects are discussed in detail, and then have an article in HESS focused on the data analysis only allowing for an in-depth analysis. This would also allow for elaboration on some aspects that are not yet discussed (*e.g.*, elevation, impact of snow if relevant for this catchment, more detailed spatial analysis). I think this option would really increase the potential impact of the significant modelling effort that has been made. But I would understand if the authoring team do not want to go through this extra work.

**Minor comments:**

**P1L38** [Q] should be (Q)

**P1L52-52** "*in the face of a changing climate*", maybe just say "to climate change"

**P3L79-80** Maybe define exactly what HERs are, or give a reference

**Figure 1** Add HERs "borders" also in left panel. In the manuscript, this figure is not vectorized and small text are really pixelized when zooming to reads them. Maybe provide a vectorized figure or a higher resolution bitmap figure. Add in the figure or caption the source for the maps shown

**P5L101** "*bottom*" should be "right"

**P5EQ(5)** Can't it be written in the more compact form $max(SF_{Left}, SF_{Right})$?

**P8L200** Why using $log(Q)$ in this analysis?

**P9L224** What does IQR mean?

**P9L226** "r" is used here, while "R" is used in figures

**Figure 2** There is a "a" on the right below the colour legend

**P11L267-269** Please add a reference here to support this statement

**P16L306-308** How negative Q trends, just by themselves, suggest an effect on Tw? "suggested" should be "suggesting".

**P20L335-338** First, despite it is clearly stated on the abstract of the paper Michel (2020) that: "*The mean trends for the last 20 years are + 0.37 ± 0.11 ◦ C per decade for water temperature, resulting from the joint effects of trends in air temperature (+0.39 ± 0.14 ◦C per decade), discharge (−10.1 ± 4.6 % per decade), and precipitation (−9.3±3.4% per decade)*", I think now that this paper does not show a real impact of Q on Tw, but rather a correlation between Q and Tw in summer. If I had to rewrite this paper today, I would not be so categorical (and this why I also question it in your paper). Second, when you say: "In *contrast with our results, they found Tw trends lower than Ta trends due to influence of snow melt and glacier melt"*, this is not totally exact. Indeed, trends found in Alpine catchments are lower due to the mentioned effects. For the low-altitude catchments where snow plays no role, trends are indeed closer, but Tw trends remain slightly slower than Ta trends (compare Figures S17 and S18). However,

on an annual basis, we are talking about a few tenths of a degree less in my article and a few tens of a degree more in yours, so taking into account all the uncertainties involved, I see no contradiction. In addition, different regions are studied.

**Introduction and Section 5.5** Nuclear plants cooling is never mentioned in the paper. This might not be relevant for the Loire (but Bustillo et al. (2014) mention some plants in the catchment), but in general in France the question of cooling nuclear plants in the future with increasing air and water temperature will be a real challenge and I think it is worth mentioning it (see e.g. Bourqui et al., 2011).

**P22L398** Shouldn't it be "small rivers"?

**References**

Arora, R., Tockner, K., and Venohr, M.: Changing River temperatures in northern Germany: trends and drivers of change, Hydrol. Process., 30, 3084–3096, DOI: 10.1002/hyp.10849, 2016.

Bourqui, M., Hendrickx, F., and Le Moine, N.: Long-term forecasting of flow and water temperature for cooling systems: Case study of the Rhone River, France, AHS Publ, 348, 135–142, 2011.

Bustillo, V., Moatar, F., Ducharne, A., Thiéry, D. and Poirel, A.: A multimodel comparison for assessing water temperatures under changing climate conditions via the equilibrium temperature concept: case study of the Middle Loire River, France. Hydrol. Process., 28: 1507-1524. DOI: 10.1002/hyp.9683, 2014.

Michel, A., Brauchli, T., Lehning, M., Schaefli, B., and Huwald, H.: Stream temperature and discharge evolution in Switzerland over the last 50 years: annual and seasonal behaviour., Hydrology & Earth System Sciences, 24, DOI: 10.5194/hess-24-115-2020, 2020.

Moatar, F. and Gailhard, J.: Water temperature behaviour in the River Loire since 1976 and 1881, Comptes Rendus Geoscience, 338, 319–328, DOI: 10.1016/j.crte.2006.02.011, 2006.

Orr, H. G., Simpson, G. L., Clers, S., Watts, G., Hughes, M., Han- naford, J., Dunbar, M. J., Laizé, C. L. R., Wilby, R. L., Bat- tarbee, R. W., and Evans, R.: Detecting changing river temperatures in England and Wales, Hydrol. Process., 29, 752–766, DOI: 10.1002/hyp.10181, 2015.

Thiéry, D.: Forecast of changes in piezometric levels by a lumped hydrological model, Journal of Hydrology, 97, 129–148, DOI: 10.1016/0022-1694(88)90070-4, 1988.

Webb, B. W. and Nobilis, F.: Long-term changes in river tempera- ture and the influence of climatic and hydrological factors, Hy- drolog. Sci. J., 52, 74–85, DOI: 10.1623/hysj.52.1.74, 2007.

---

## Author Comment (AC1)

We thank the reviewer for his very useful comments. By addressing these comments, we believe the paper will be substantially improved, particularly with respect to a refined focus and a more detailed description of hydrological and thermal models. The reviewer comments are in italics, our responses are in normal font, and the proposed text additions and modifications are in bold.

Please note that in the following, "P", "L" and "SM" stand for page number, line number and Supplementary Materials, respectively.

**#Reviewer 1:**

General comments:

*The submitted paper presents a large-scale modelling effort of river discharge and temperature for the Loire River basin in France. The modelling is carried out in two steps, first for discharge and then for water temperature. The models are forced with reanalysis data provided by Météo-France and validated on a large number of measurement stations. The model results are then used to analyse and discuss the trends in discharge and water temperature, the factors influencing the modelled trends in water temperature, the spatial patterns of the resulting trends, and the role of riparian vegetation and Strahler order.*

*This article presents an important work and a valuable contribution to his field of research. In general, the paper is well written in terms of language and has clear and nice figures.*

The authors would like to thank the reviewer for this assessment.

*However, I have some concerns which I detail below. The first concern is the clarity, completeness, and organisation of section 3 "Method and data", where, in my opinion, some crucial information is missing.*

We believe that the responses to the following comments address this comment.

*The second concern is the robustness of some of the results, in particular the validation of the model and the strength of some of the analysis performed. Indeed, the results and discussion sections tend to analyse many different aspects and perhaps only the most robust ones could be kept. In addition, since quite simple models are used, I would be more careful for some interpretation of the results.*

We believe that the responses to the following comments address this comment.

*I also regret that no data or models are shared along the article. I imagine that sharing of forcing data may not be allowed, however I would strongly encourage the addition of a "Data and code availability" section to detail how this work can be replicated. If possible, sharing with the community the time series produced by the model would also be appreciated. In the same vein, the supplementary material could be improved to provide interested readers with a more complete overview of the large number of results obtained.*

We completely agree on the data sharing idea, and this is why we already planned to write down a data paper on simulated streamflow and stream temperature time series for reaches corresponding to location of water quality stations (around 1000 stations) over the 1963-2019 period. Please note that we indeed have no right to publish the Safran meteorological reanalysis data belonging to Météo-France.

*Main comments*

*1.1 Data description*

*My first recommendation is to create a "Data" section (with sub-sections) where all details on forcing, validation and geographical data are grouped. Currently, the information is spread over P3L83-86, P4L97-P5L104, P5L118-121, L123-124, L125-128, P7L165-167 (I may have forgotten some). Grouping them together would make for easier reading as many details now appear in the models' description sections.*

Please first note that P3L83-86 is about HERs (Hydro EcoRegions) that are not input of any of the models. This confusion has been repeated in other comments by the reviewer. They are simply regions within which land use/land cover, and climatic conditions are rather homogeneous. They were developed for the Water Framework Directive to give ecological reference conditions. Such information has been already specified in P3L80. HERs were used in the current study to be able to consider the similar landscape characteristics for reaches belonged to the same HER. Thus, it is used to compare reaches with the same landscape characteristics together like what has been done in assessing the influence of reach size and vegetation cover. Moreover, it helps to report and compare the trends over the basin in a simple way at this large scale.

Please note that the rest of the data pointed out by the reviewer (*P4L97, P5L118-121, L123-124, L125-128, P7L165-167*) are the part of inputs of the hydrological and thermal models. We believe that we cannot talk about such data before explaining each model Nevertheless, in the revised manuscript, agreed with the reviewer, we will add a data section before the model description section (where we just specify subsets of data that are used for simulation/calibration/validation). Moreover, to clarify that each model functions separately with their own specific input data, the following paragraph will be added P4L90 Section 3.0:

**"We used two models to calculate stream temperature in the Loire River basin according to the method developed by Beaufort et al (2016). The first model is the semi-distributed hydrologic model EROS, which estimates daily streamflow at sub-basin outlets. The second model is the fully distributed, mechanistic temperature model Temperature-NETwork (T-NET; Beaufort et al. 2016 and Loicq et al. 2018) that uses streamflow from EROS and meteorological reanalysis data to estimate Tw at each reach in the Loire River basin. These models are briefly described below, and are detailed in Thiéry (1988), Thiéry and Moutzopoulos (1995), and Thiéry (2018) for EROS, and Beaufort et al. (2016) and Loicq et al. (2018) for T-NET."**

*In addition, a comprehensive table of all stations used for calibration and validation should be provided in the supplementary material (SM later), with coordinates and station names, and also indicating the data provider.*

We agree. The data source of hydrometric stations used for both calibration and validation was already mentioned in P5L100. Nevertheless, a list of these stations with coordinates will be added to SM. A list of stream temperature stations including the data provider (and the corresponding link) will be also provided in the SM.

*The coordinates should be added in Table 1. For long-term stations, one could even consider numbering them in the table and then indicating the numbers on Figure 1.*

We agree and it will be added in revised manuscript**.**

*Maps similar to Figure 3 and S5, but showing the annual and seasonal average of Tw, Ta and Q over the whole catchment area could be a useful addition to the SM to capture the different local conditions in the catchment.*

We agree and we will add the following figures to SM showing annual averages.

[Figure]

Annual mean Ta, liquid and solid P, and PET over the 1958–2019 period. These data are derived from the SAFRAN reanalysis data (Quintana-Segui et al., 2008; Vidal et al., 2010)

[Figure]

Annual mean Tw and Q. Please note that for Q, log(annual Q) are presented.

*A map similar to Figure 2, middle panel, with a colour indicating the Strahler order could be added in the SM.*

Such information can already be seen on the previous figure that will be included in SM.

*For non-French readers, a map showing the location of the Loire basin in France (or in Europe) should be added. It could be integrated as a new panel in figure 1.*

It has been already presented in in the top right corner of Figure 1, left panel. We will make it explicit in the corresponding caption.

*In P5L103 it is said that the Q time series are "naturalised". However, no reference or details of the procedure used are provided. This information should be included because, as you note later in section 5.4, anthropogenic disturbances are of major importance.*

We agree, and P5L103 will be modified as follows:

"**Stations along the main Loire (14) and Allier (11) rivers are influenced by operations of 4 large dams for hydropower, flood control and low-flow management, notably through summer releases. Time series at these stations have been first naturalized by EDF (Électricité de France) based on variations in water storage in reservoirs due to operations based on target storage curves (Naussac, Villerest) and optimization of the hydropower energy generation for the electric grid needs (Grangent, Montpezat).**"

*In addition, I did not find in the document the source of the time series for water temperature. Is it the same supplier as for the discharge? Is it also "naturalized"?*

The data provider of stream temperature were already mentioned in P23L410. However, as mentioned before, the link to data provider will be added to the revised manuscript.

Please note that Tw data are raw observations, i.e. not "naturalized", as already made explicit in the manuscript P7 L172.

*1.2 Description of the EROS model*

*Only a few details are given on how the model actually works. In addition, the two main references given for the model are in French. Although it is not necessary here to describe all the details of the models, the main points should be provided to the reader. In particular, details of the mass balance should be given. Is it precipitation - evaporation, or can some of the mass be lost through deep soil infiltration? How is the water transported through the soil? Does EROS use a reservoir model? Finally, is there any routing of water into the stream network carried out in EROS or is only the release of water at the sub-catchment scale simulated? P5L109-111 suggest that routing is done in T-NET, but this is not clearly stated, nor is it mentioned in section 3.1.2. On the other hand, section 3.2.1 suggests that Q is obtained directly from EROS, which means that the routing is done in EROS.*

 *It is not mentioned which parameters are calibrated (this should be stated), which values are tested for calibration, and which values are finally chosen. This should be stated in the SM to allow reproducibility. P5L105-107 states that: "The calibration aimed at maximizing the Nash-Sutcliffe efficiency criteria (Nash and Sutcliffe, 1970) on the square root of streamflow and minimizing the overall bias, in order to simulate correctly the whole range of Q values". How are the two metrics combined (NSE and bias) to assess the quality of the calibration? Why is the square root used? How many calibrations are performed, how are the calibration values chosen (pure chance or more advanced algorithm)? Again, the only non-French peerreviewed source provided is Thiéry (1988). This paper describes a water level model for an unconfined aquifer and I imagine that EROS is a model based on this work, but rather different from this 1988 version.*

*Overall, more details about the model are needed to enable the reader to understand how the EORS model works. The main assumptions and principle of the model should be stated, as well as details on the calibration procedure and parameters (a part can be added to the SM). I have found many applications of this model in the literature; these could also be cited as application examples.*

We agree, and therefore the section "EROS hydrological model" (section 3.1.1), P4L95, is modified as following, which includes the principles of the EROS model, its input data, calibration process and variables, and applications of EROS in other studies:

**"The EROS semi-distributed hydrological model simulates the daily streamflow at the outlet of 368 sub-basins (ranges between 40 and 1600 km$^2$; mean drainage area = 300 km$^2$), designed to be as homogeneous as possible with respect to land use and geology. On each of these sub-basins, the water balance is modelled by a lumped model using three reservoirs and a routing function for propagation across sub-basins (Thiéry, 1988; Thiéry and Moutzopoulos, 1995; Thiéry, 2018). This hydrological model has already been used in several other studies including climate change impact studies (Ducharne et al., 2011; Habets et al., 2013; Bustillo et al., 2014).**

**The EROS model uses daily Ta (°C), solid and liquid precipitation (mm), and reference evapotranspiration (ET0, mm). Ta and precipitation are provided by the 8 km gridded Safran surface reanalysis from Meteo-France over the 1958–2019 period (Quintana-Segui et al., 2008; Vidal et al., 2010). ET0 was computed from Safran variables with the Penman-Monteith equation (Allen et al., 1998).**

**For calibration, 352 hydrometric stations with observed Q data were extracted from the French national Banque Hydro database (http://www.hydro.eaufrance.fr/) (Figure 1, right panel). Stations along the main Loire (14) and Allier (11) rivers are influenced by operations of 4 large dams for hydropower, flood control and low-flow management, notably through summer releases. Time series at these stations have been first naturalized by EDF (Électricité de France) based on variations in water storage  in reservoirs due to operations based on target storage curves (Naussac, Villerest) and optimization of the hydropower energy generation for the electric grid needs (Grangent, Montpezat). EROS was then calibrated over the 1971-2018 period against daily Q at all 352 sub-basins with at least 10 years of daily observations. The calibration aimed at optimizing all unknown parameters (soil capacity, recession times, propagation times) through maximizing the Nash-Sutcliffe Efficiency (NSE) criterion on the square root of streamflow and minimizing the overall bias. Considering the NSE criterion on the square roots of the flows provides an estimate of model performance without favoring neither high flows nor low flows. The overall calibration criterion C is: C = mNSE – w.mRB where the weighting factor w is fixed at 0.05 .**

**A 3-year warm-up period (1971-1974) was discarded from the overall calibration period (1971-2018, which maximizes the number of streamflow observations). The calibrated model was then used to simulate streamflow over the whole 368 homogenous sub-basins in the Loire basin over the whole 1963-2019 period. Note that although meteorological variables are available from 1958 onwards, the first years (1958-1962) were discarded from the analysis to ensure the model convergence."**

Moreover, to clarify that simulated Q by EROS used in T-NET and there is no routing for simulating Q in T-NET, P5L110 of the current manuscript will be removed and it will be added between P6L150 and L151 as follows:

**"- Streamflow: the daily streamflow simulated at the outlet of 368 homogeneous sub-basins by the EROS model (see section 3.1.1) are redistributed along the river network inside each sub-basin according to the reach drainage area for informing the T-NET model at the reach scale. The ratio of sum of the lengths of all reaches upstream of a reach to the sum of the lengths of all reaches located in a sub-basin is used as a proxy for the drainage area of a reach."**

*There is also no mention of the other input parameters of the model. The authors describe three different HydroEco regions (HER), but there is no information on how the region influences the model parameters (and thus the model outputs).*

Please see response to a comment above.

*1.3 Description of the T-NET model*

*Here too, important pieces of information are missing. This is even more problematic than for the EROS model as no references are given for this model.*

We actually did provide references for the T-NET model (original text P5L120). To clarify this point, the following paragraph will be added in P4L90 Section 3.0 (already mentioned before):

**"We used two models to calculate stream temperature in the Loire River basin according to the method developed by Beaufort et al (2016). The first model is the semi-distributed hydrologic model EROS, which estimates daily streamflow at sub-basin outlets. The second model is the fully distributed, mechanistic temperature model Temperature-NETwork (T-NET; Beaufort et al. 2016 and Loicq et al. 2018) that uses streamflow from EROS and meteorological reanalysis data to estimate Tw at each reach in the Loire River basin. These models are briefly described below, and are detailed in Thiéry (1988), Thiéry and Moutzopoulos (1995), and Thiéry (2018) for EROS, and Beaufort et al. (2016) and Loicq et al. (2018) for T-NET."**

*P5L115-116 says: "To simulate Tw, the equilibrium temperature (Te) is first computed, the temperature at which the net heat flux across the surfaces of the stream is null". Then details of how some of the energy fluxes are calculated are given, but the above sentence is never followed by a "then" or "next". So, if I understand correctly, the temperature of the water being modelled is the equilibrium temperature? Or are you using a formulation similar to Bustillo's (2014) eq. (6). A crucial piece of information is missing here. And if only Te is used, this critical assumption and its consequences need to be discussed.*

We agree, and in this regard, we will improve description of the thermal model in P5L115-116 as following:

**"TNET is a fully mechanistic, 1D model that simulates hourly Twi,j at distance, i, along reach, j, by solving the local heat budget. The heat budget of each reach includes six fluxes (W m⁻²): net solar radiation, atmospheric longwave radiation, longwave radiation emitted from the water surface, evaporative heat flux, convective heat flux, and groundwater heat inflow. Briefly, the model calculates the longitudinal change in Tw at time t (dTw/dx) for steady-state conditions in order to achieve thermal equilibrium (i.e., ΣHi,j=0, where H is a heat flux), while accounting for confluence mixing. Detailed information about the T-NET model principles and calculation of six heat fluxes at the water-air and water-stream bed interfaces, thermal propagation were provided in Beaufort et al., 2016 and Loicq et al., 2018. Note that unlike the EROS hydrological model, the T-NET thermal model does not have any free parameter, hence, it does not require calibration and it is only validated."**

*In section 3.1.1 it says that EROS is used to calculate discharge over 368 sub-catchments, whereas in section 3.1.2 52'278 reaches are mentioned. So, there are several reaches per sub-catchment and within each sub-catchment, the water supply simulated by EROS is distributed to the reaches using the drainage area? But again, how and in which model is the routing calculated? P6L156-157 clearly states that "Q is the daily mean streamflow provided by the EROS model". But how can this be done if there is no routing in EROS? P7L160-161 explains*

*that the travel time is calculated in T-NET, but this information is not used for the calculation of the water temperature (at least that is my understanding), but would be mandatory information for routing.*

We believed that responses to previous comments address these comments as well.

*No details are provided on how sensible, latent and groundwater heat fluxes are calculated. Equations or references should be provided.*

It can be found in Beaufort et al. (2016). The modifications proposed in the previous comments address also this one.

*I understand that no calibration is done for TNET, is this correct?*

Yes, there is no calibration for T-NET. The modifications proposed in the previous comments address also this one.

*Are there any other parameters used in the model? For example, as with EROS, how are the properties of HERs taken into account in the model? The results are discussed for the different HERs, but this discussion only makes sense if the HERs are somehow parameterized in the model.*

This already has been responded. As it is mentioned before, the HERs are not input of any of the models.

*Finally, in addition to my questions about routing, there is no information about reach-to-reach heat advection. Is there reach-to-reach heat transfer? If not, this would considerably weaken the analysis carried out in terms of the Strahler order and of the whole model in general. To conclude this first part of the discussion, I would really encourage the authors to add a few paragraphs better detailing the data sources, the model workflow, and the main assumptions of the models. As I say below, it would also be important to indicate the limitations of the models.*

Yes, there is an upstream-downstream thermal propagation. Please see Beaufort et al., 2016 page 5. The modifications proposed in the previous comments address also this one.

2.1 Calibration and validation of EROS and T-NET

*Firstly, details of the calibration procedure and parameters should be provided (see above). Secondly, why is no validation period used to infer the quality of the calibration? Indeed, time series are usually divided into a calibration period and a validation period. By using only a calibration period without validation, we have no information on the potential overfitting of the model at the calibration station during the calibration period. Depending on the modelling effort required, I would strongly recommend recalibrating the model over a shorter period and using a few years for validation.*

Modifications to P4L95 related to EROS model was already dealt with in previous responses. On calibration and validation: first, such a spatially distributed calibration requires a lot of streamflow observations and performing a classical split-sample test would not allow that given the numerous and diverse gaps in streamflow observations. Second, using the complete set of available information over the past decades for calibrating the hydrological model is a standard approach in climate change impact studies. And the next step after the present study is obviously to use this calibrated EROS model to run hydrological (and afterwards thermal) projections over the 21$^{st}$ century.

*Furthermore, looking at Figure 3 of Thiéry (1988), we see a clear decrease in the performance of the model during the validation period.*

Several points should be noted here: first, in Thiéry (1988), the period was much shorter (11 years) leading to a necessarily less robust (with respect to interannual varaibility) model response. Second, the assessment is here done on aquifer levels, not streamflow like in the present study. Third, validation results in Thiéry (1988) are quite satisfying with the exception of two sites impacted by one specific precipitation input, i.e. the station observation at Abbeville, which may point to a possible deficiency in these inputs. Note that in the present study, the input data are the reference Safran gridded surface reanalysis over France (Vidal et al., 2010).

*In section 3.2.1, NSE on Q, ln(Q) and sqrt(Q) are mentioned, does this correspond to the "the mean NSE criteria for low, medium and high flows" mentioned in P9L221?*

Yes. We will use the same configuration (Q, ln(Q) and sqrt(Q)) in the results instead of using low, mean and high flows in the revised manuscript.

*Little detail is given on the quality of the calibration of Q. Indeed, only the mean NSE is given (no details on the variance), a graph of the distribution of values should be added in SM, as well as a graph showing the simulated and measured time series for some stations. In Figure S2, the bias is only shown for the 44 NHN stations, why not show all 352 stations (using a shorter time period)?*

We agree and we will add some figures to SM. We will also show simulated and observed Q time series for three sub-basins in the upstream, middle and downstream part of the basin in SM. In this regard, P9L219-221 will be modified as following:

**"The EROS model performs well at the 352 calibration stations at the annual scale with a median relative bias close to 0 (see Figure 4.12, left panel). It slightly underestimates winter Q (median relative bias (across stations)=-6.27%) and spring Q (-3.47%), and overestimates summer Q (+34.7%) and fall Q (+20.9%). The overestimation in summer and fall could be due to the fact that the EROS model does not take into account water abstractions for irrigation that occur mainly during the low-flow period (for abstraction volumes, see the national abstraction database https://bnpe.eaufrance.fr/).**
**The EROS model also performs well at 44 RHN stations with long-term continuous daily data at the annual scale with a median relative bias of 0.37% (see Figure 4.12,**

**right panel). It slightly underestimates winter Q (median relative bias (across stations)=-7.26%) and spring Q (-6.79%), and overestimates summer Q (+37.7%) and fall Q (+24.7%).**

**A good performance of the EROS model in reconstructing daily Q was also seen at three different hydrometric stations located in the upstream, the middle, and the downstream part of the Loire River basin."** Please note that the corresponding figure will be added in SM between Figures S2 and S3 of the current manuscript.

[Figure]

[Figure]

[Figure]

[Figure]

Figure: Simulated and observed daily Q at three different hydrometric stations: (top) in the upstream (L'Allier à Monistrol-d'Allier), (middle) in the middle (L'Arnon à Méreau [Pont de Méreau]), and (bottom) in the downstream part (La Loire à Montjean-sur-Loire) of the Loire River basin.

*In Section 4.1, it is shown that most of the simulated summer trends are not significant and that the simulated summer trends are poorly correlated with the observed trends. However, these summer trends are used extensively later in the document (e.g. Figure 5, Table 2). Given the uncertainty around these trends over the calibration period, I doubt that they are robust enough to perform such an analysis.*

Please note that the low correlation value found in summer in Figure S3 originates from poor simulation at very few stations all located in HER B. Sauldre@Selles-sur-Cher (K6492510) and Ardoux@Lailly-en-Val (K4443010) gauge catchments where numerous small ponds are found and the highly decreasing observed trends may be due to the increasing evaporation from these ponds which are obviously not included in the EROS model. This is also true in a lesser extent for the Auron@Bourges, in which a canal follows a large part of the course of the river and may play a similar role with respect to summer evaporation trends. Apart from these specific stations, a coherence as good as in other seasons is found between trends in simulated and observed streamflow.

*For Tw, only the performance in terms of trends and biases is presented. A presentation of performance in terms of mean square error would also be informative in assessing the performance of the model as this metric is commonly used in the literature.*

We agree. The following figure (including RMSE) will be added to the bias figure in the SM (Figure S2). The following results will be also added to P9L224:

**"The median RMSE of the T-NET thermal model, for small and medium rivers, ranges between 0.52 °C (Annual) and 0.91 °C (DJF and JJA) across seasons (see Figure S2, bottom left panel). For large rivers, the median RMSE of the T-NET thermal model ranges between 0.38 °C (annual) and 1.11 °C (JJA and SON) across seasons (see Fig. S2, bottom right panel). Moreover, 53-83% stations (resp. 50-100%) on small and medium (resp. large) rivers have a RMSE<1 °C across seasons."**

[Figure]

*The bias is discussed for Q and Tw. However, as the subsequent analysis is mainly trend based, and bias has no impact on trend, I am not sure of the relevance of this metric here.*

We agree that a model with bias can also simulate temporal trends well. Nevertheless, the first step of using a model is to assess its biases. Please note that the model performance in simulating seasonal Tw and Q was not assessed before in Beaufort et al., 2016. Moreover, such a comment is rather surprising after the previous one requesting an assessment based on the RMSE, which is highly influenced by any bias.

*Section 5.1 does not add much to section 4.1. I would like to see in these sections more discussion of where the models are underperforming and therefore the limitation of the dataset obtained. Indeed, the models used, like all models, are not perfect. Identifying the*

*limitations and thus focusing the analysis and discussion on the part that proved to be within the radius of validity of the models is a mandatory step in modelling studies.*

**We agree and the limitations of the models as the one (related to trends in open-water evaporation) seen for trends of the EROS model in summer will be discussed in the revised manuscripts.**

*Finally, the part concerning riparian vegetation seems to be a new addition to the model in this article. My concern is that no validation is shown. The approach should be validated using a Tw measurement station in a shaded area and compare the model performance with and without shading. In the absence of validation and evaluation of the effectiveness and limitations of the approach, I find it difficult to proceed with the analysis.*

T-NET improvement by this riparian shading method was already shown by Loicq et al., 2018. However, a sentence referred to this reference is missing. Therefore, the following sentence will be added in P6L135**: "Loicq et al., 2018 (their Fig. 11) showed that Tw simulated by this method was close to observed Tw".**

*2.2 Link between Ta, Tw, and Q*

*The comparison of Ta and Tw trends and the potential impact of Q are widely discussed. I appreciate this effort and think that there is still much to understand about the interaction between Tw and Q.*

The authors thank the reviewer for this assessment.

*First of all, great importance is attached to the finding that Tw increases faster than Ta for most seasons and on an annual basis. This result has already been found in some regions (see e.g., Webb and Nobilis, 2007; and Arora et al., 2016), but is in contradiction with other studies (see e.g., Moatar and Gailhard, 2006; Orr et al., 2015; and Michel et al., 2020). I believe this result would merit further discussion to assess its strength.*

We agree that such findings depend on the study area and even on the study period. In this regard, we decided to specify the study area in the new manuscript as following: "**Regional, multi-decadal analysis on the Loire River basin reveals that stream temperature increases faster than air temperature**". Moreover, in the manuscript, we will better discuss Tw and Ta trends observed in the other regions like what the reviewer pointed out.

 Please note that in Moatar and Gailhard, 2006, Tw increases faster than Ta in spring and summer and at annual scale for the whole 4 stations on the Loire River.

*The main factor used to explain why the trends in Tw are more important than those in Ta is the discharge. However, trends in other forcing variables should be shown. Indeed, in winter, Figures 3 and S5 show that the variable explaining why TW trends > TA is probably not discharge (since no significant trend in discharge is found).*

We agree and we already excluded winter from this statement. Please see P11L260.

*Similarly, summer and spring show marked negative trends in discharge in some parts of the catchment, whereas these are the seasons where the trends in Ta and Tw are most similar. The addition of the seasonal discharge trend in the boxplot in Figure 4 would facilitate the analysis.*

First of all, Figure 3 clearly shows reaches in the upstream of the basin for which Ta trend (middle panel) are in range of 0.4-0.6 °C/decade (in orange) while Tw trends are >0.6 °C/decade (in red), so we do not agree that Tw and Ta trends are similar in spring and summer. Yes, the median values across the basin for Tw and Ta trends is similar (Figure 4), but the spatial pattern is clearly not, hence the added value of Figure 3. Please note that, Figure 4 compares dispersion and median values of Tw and Ta trends across the basin and not reach by reach while Figure 5 (which is modified and shown in the following) compares trends reach by reach. Figure 4 is just the first step to see whether Tw are more spatially variable than Ta trends. It therefore suggests that other factors than Ta are affecting Tw trends. Looking at the spatial pattern of trends in Figure 3 suggests spatial links between an additional increase in Tw and a decrease in Q, mainly in the upstream part of the basin (considering both significant and non-significant trends). Finally, Figure 5 compares Tw, Ta trends and the sign of Q trends reach by reach to test the hypothesis of decreasing Q as a possible controlling factor. Indeed the modified Figure 5 (shown in the following) clearly shows that regardless of the significance level, for the majority of reaches with Tw trends> Ta trends, a decrease in Q occurs coincidentally across seasons, with again the exception of winter.

With this modified Figure 5, P11L260-264 will be modified as following:

**"Indeed, where Tw trends exceed Ta trends, decreasing Q trends occur coincidentally at the majority of reaches (43-94%) for all seasons (with the exception of winter) irrespective of whether all significant and non-significant trends are considered (Figure 5, top panel) or only significant trends of all three variables are considered (Figure 5, bottom panel). Of these specific reaches where all factors converge (trend in Tw higher than trend in Ta, and decreasing trend in Q), most are located in HER A irrespective of considering all significant and non-significant trends (52-90% of such reaches across seasons; see spatial figure, left panel in SM), or considering only significant trends of all three variables (100% of such reaches across seasons; see spatial figure, right panel in SM)."**

Please note that the spatial figures showing location of these reaches will be put in SM (between Figure S7 and S8 of the current manuscript).

[Figure]

Percentage of reaches with both signifcant and non-significant trends (%)

Percentage of reaches with significant trends (%)

Trends
- Tw<Ta & Q>0
- Tw<Ta & Q<0
- Tw>Ta & Q>0
- Tw>Ta & Q<0

Significant & non-significant trends

Significant trends

[Figure]

**Trends**
Tw<Ta & Q>0
Tw<Ta & Q<0
Tw>Ta & Q>0
Tw>Ta & Q<0

Spatial figures

*Figure 5 is used to support the hypothesis that Q is the main driver, see P11L259-263:
"Overall, Tw trends were more spatially variable than Ta trends, suggesting the conditional influence of Q trends (Fig. 4). Indeed, where Tw trends exceeded Ta trends, decreasing Q trends occurred coincidentally at the majority of reaches for all seasons – with the exception of winter – (43-72 %, depending on season; Fig. 5. Of these specific reaches where all factors converged (trend in Tw higher than trend in Ta, and decreasing trend in Q)" (please note that there are some grammatical problems in the second sentences). There are many shortcuts here. Firstly, the larger scatter suggests that factors other than Ta have an impact on water temperature, but this does not in itself show that Q is responsible, it could be any other forcing variable. Secondly, and more importantly, Figure 5 shows the percentage of reaches where "all factors converge", which I interpret as "all trends are significant".*

We of course acknowledge that other factors are at play in the trends that we observed (including model error and bias), and this is also commented in the Discussion. However, we suggest that Figure 5 (and its modifications, see above) are strong supporters of our hypothesis that increasing Tw can be due to the joint effects of increasing Ta and decreasing Q. We do not claim that this is the main driver. At this point, we just state what we observed as the reviewer also referred to "…decreasing Q trends occurred coincidentally at the majority of reaches for all seasons".

Moreover, in the paragraph that the reviewer referred to, we already explained, "all factors converge" by adding "(trend in Tw higher than trend in Ta, and decreasing trend in Q)".

*Please clarify this point, as when comparing Figures 5 and 3, I rather understand that all trends are included in Figure 5. Including non-significant trends would be a significant bias here, as many non-significant trends are just above or just below zero and the figures are based on trend signs. For Figure 5 to be complete, the distribution between Q>0 and Q<0 should also appear in the blue part. Indeed, the figure now shows that Tw>Ta in most cases when Q<0, however Figure 4 shows that Tw>Ta in most cases anyway.*

Yes, the current Figure 5 include both significant and non-significant trends, but we believe that the modified Figure 5 (see above) brings an adequate response to this comment.

*This figure could be used to show the impact of Q if, and only if, we can see that the proportion of Ta>Tw vs Ta<Tw changes if Q>0 and Q<0. Also, the number of catchments used probably differs significantly between seasons (at least if only significant trends are shown). Here I see that there may be something irreconcilable in this figure:*
*- Either all reaches are kept, including those with insignificant trends, but then the figure itself will lose much of its meaning for the reason mentioned above.*
*- Either only significant trends are retained, but only a small minority of reaches show significant Q trends in certain seasons, and the figure would then only show a small subset of reaches.*

These points were already addressed in the previous comments and in the modified Figure 5 shown above.

*This question of the relationship between Tw and Q trends is not straightforward (you can look at the introduction of Arora et al. (2016) for a good review of the literature available by then) and, it is important to note, seeing a correlation between Tw trends and Q trends does not imply any causality.*

Thanks you for this suggestion, but we are already aware of the paper referred to here as we already cited it several times in the manuscript. And of course, we are aware that correlation does not imply causality, hence our position only suggesting this causality, as it is supported by coherent spatial patterns, temporal patterns, and well-understood physical processes. The aim of this article was just to find out the magnitude of trends in Tw, and possible explanations for Tw trends higher than Ta trends by assessing the spatial (Figures 3 and 5) and temporal links (Figures 6-8, and Table 2) between trends in Tw and Ta and/or Q.

*This is what we see in the subsection "Synchrony of annual anomalies" and in Figure 8: low Q summers correspond to high Ta summers, so it is difficult to assess which of the two, or the combination, leads to an increase in Tw.*

As you are probably aware, summer streamflow does not only depends on summer temperature (through summer evapotranspiration), but also from baseflow originating from precipitation in the previous months, during the recharge season. Distinguishing between Ta and Q factors on Tw in this study derives from that they are the two main inputs of the T-NET model. Of course one could devise a formal attribution framework where one may e.g. remove trends in Q and trends in Ta alternatively in T-NET inputs, but this would require much more work than the objectives set up for this manuscript.

*Furthermore, Figures S10 and S11 suggest that the negative discharge trend is caused by an increase in ET rather than a decrease in P. The regions concerned (see Figure 3), are those where the increase in Ta is most significant. Thus, the causal chain here appears to be Ta increasing → ET increasing → Q decreasing. So, even if Q is shown to be a factor influencing Tw, it originates (mostly) in the increase in Ta. Thus, it might be misleading to say that a decrease in Q is a contributing factor, as I think that for most readers a decrease in Q would mean a decrease in precipitation, whereas here no significant decrease in precipitation is found, and thus no impact of the precipitation regime on Tw can be assessed. The real impact of ET can be assessed by comparing the measured trends of discharge to the trends on measured precipitation in the catchment, in order to confirm if the increase in ET modelled is correct.*

We refer the reviewer to the response to a pervious comment on summer baseflow originating from previous recharge season. The causal chain presented by the reviewer may be effective when looking at the annual scale, but this is made much more complex with the seasonality in T and P, and of course the fact that catchment integrate theses signal in time across seasons. Moreover, we find it surprising that the reviewer suggests that " it might be misleading to say that a decrease in Q is a contributing factor" while a recent paper he was the main author of actually says right in the abstract "The mean trends for the last 20 years are $+ 0.37 \pm 0.11 \circ C$ per decade for water temperature, resulting from the joint effects of trends in air temperature ($+0.39 \pm 0.14 \circ C$ per decade), discharge ($-10.1 \pm 4.6$ % per decade), and precipitation ($-9.3 \pm 3.4$ % per decade)." (Michel et al., 2020). Such an assessment goes much further in the attribution of Tw trends than we aim to, while we only gather clues (spatial patterns, temporal patterns, physical reasoning) towards such a still distant formal attribution.

*With all this discussion, I really question this sentence in the abstract: "Importantly, air temperature and streamflow exerted joint influence on stream temperature trends, where the greatest stream temperature increases were accompanied by similar trends in air temperature (up to +0.71 °C/decade) and the greatest decreases in streamflow (up to -16 %/decade)". Indeed, as discussed above the discharge decrease might just be a "side-effect" of the increase in Ta through increased ET, but since it is also the region with the highest Ta increase (P16L229), I don't think that we can conclude that "air temperature and streamflow exerted joint influence on stream temperature".*

Again, we respectfully disagree with the reviewer and at the same time agree with Michel et al. (2020) which asserts such a joint influence while we clearly only bring forward here this joint influence.

*Continuing with the subsection "Synchrony of annual anomalies", I do not really see the added value of the change point analysis. Furthermore, as it seems to illustrate an abrupt change in the late 1980s, I would think that a trend analysis is then not appropriate for these time series.*

This is a good remark indeed. However, trend assessment and change-point assessment are definitely not incompatible, and both analyses bring forward their own contribution of our understanding. More specifically here, the trend analysis show the increase/decrease of the variables and the change-point analysis critically bring some more information on the relationship between Ta, Q, and Tw in their temporal synchronicity, again towards a still distant attribution.

Moreover, please note that a time series can have both a change-point and a trend. A visual assessment can clearly shows this point (e.g., see Tw and Ta anomalies in Figure 6).

*Going back to the Tw>Ta question more generally (sorry, all the variables are related and I had trouble organising my comments), it would be interesting to calculate trends at long-term water temperature stations to see if the same trend is observed, to reinforce the results.*

It is a good idea, but we do not have Tw long-term stations in the upstream part of the Loire catchment (HER A) where mostly Tw trend> Ta trends (see Figure 3 and modified Figure 5 above). We have long-term data for the Loire River in the downstream part of the basin, which have already been analyzed in this study and previous ones (Moatar and Gailhard, 2006; Avarello et al., 2020). Nevertheless, please note that consistent with our findings, Moatar and Gailhard, 2006 found Tw trends > Ta trends in spring, summer and at the annual scale for the whole 4 stations on the Loire River.

Moreover, it would be nice to work with observations and have less uncertainties about the results of simulations, but as seen in Table 1, only 5 stations have long-term Tw data over the whole basin. In France, the National Water temperature network was established only in 2009 (https://hubeau.eaufrance.fr/page/api-temperature-continu#/). We are facing to lack of long-term and detailed observed Tw in the basin, and the point of developing models like T-NET model was precisely to overcome this lack of data.

*In addition, a map like the one in Figure 3, but showing the difference between Ta and Tw, would be very informative about the spatial distribution of catchments where Ta trends are more important than Tw and help the comparison with discharge trends.*

We agree. Such Figure was already provided in the previous comments where modified Figure 5 was presented.

*This result receives a lot of attention in the article (see for example the title). The details (Figure 4) show that this is mainly due to the winter and autumn seasons (although the difference is statistically significant in spring, it is still really small).*

The fact that Tw trends are stronger than Ta trends is also valid at the whole annual scale. We therefore believe that the title is therefore appropriate.

*However, Figure S4 and the indicated R coefficient shows that the simulated trends in autumn have the lowest correlation with the measured trends (this is also stated in P9L232), so autumn is the season where we have the lowest confidence in the results. In any case, I would condition the general statement of Tw>Ta on the seasonal aspect since it is not general (and I would add this information in the title) and I would really stress the uncertainties about it. This result is interesting, and certainly deserves attention, but in my opinion not robust enough to be asserted in the way it is in the title of the paper (at least with what is currently shown). The discussion around the cause can also be enhanced.*

Please note that in abstract P1L10, we put the attention on spring and summer not on fall**: "Spring and summer increases were typically the greatest in the southern headwaters (up to +1 °C/decade) and in the largest rivers (Strahler order >5). Importantly, air temperature and streamflow exerted joint influence on stream temperature trends, where the greatest stream temperature increases were accompanied by similar trends in air temperature (up to +0.71 °C/decade) and the greatest decreases in streamflow (up to -16 %/decade)"**. We therefore believe that we sufficiently take account of uncertainties and agreed with the reviewer, discussion will be enhanced according to responses to previous comments.
Please also note that the low correlation between simulated and observed Tw trend in fall originates from very few stations with 8-13 years Tw data while such correlation is good at stations with longer (20-40 years) Tw data. Therefore, poor correlation in fall can be due to insufficient Tw data at very few stations. Such point will be discussed in the revised manuscript.

*2.3 HER, Strahler order and riparian vegetation*

*The study is complemented by an assessment of the influence of HERs, Strahler order and riparian vegetation on Tw. I have already raised some concerns about HERs (how they are accounted for in the models), and about riparian vegetation (the model is not validated for this). These two points need to be addressed in order to present the analysis.*

Responses to previous comments clearly stated that HER are not a conditioning/explaining factor and that the riparian vegetation influence has already been validated by Loicq et al. (2018).

*In general, all these topics are interesting, but I have the impression that they are treated only superficially and not with the necessary rigour.Moreover, there are already a lot of results presented and I think the article would still be interesting, and perhaps easier to read, if these parts were removed.*

We would like the reviewer to suggest a more "rigorous" approach than the one we used, which is rather basic and robust, derived only from model outputs and classes of vegetation and stream order. We therefore strongly disagree with the reviewer statement on a lack of rigour.

*Regarding the Strahler order analysis, this brings back to the issue of the reach-to-reach heat advection mechanism that is missing in the model description. It is answered previously. But, there is upstream-downstream thermal propagation in the model.*

This comment has already been answered previously. There is an upstream-downstream propagation of thermal signal. Please see page 4 of Beaufort et al. (2016).

*Also, showing a Strahler order map for each reach in the SM would be really informative.*

As mentioned in the previous comments, we agree and It will be added in SM.

*As mentioned in P22L397-398, the correlation between Strahler order and Tw may be due to riparian vegetation (and not a concentration of the warming when going downstream). However, this should be analysed in more details, e.g., by looking at the correlation between Strahler order and Tw separately for different shading factor values (e.g., the categories of Figure 10).*

This comment is somewhat contradictory with the previous comment relative to the lack of rigour suggested by the reviewer on this topic. Moreover, It was already shown by other studies that riparian shading influence is more important on small rivers and has no influence on large rivers (e.g., Moor et al., 2005, Loicq et al., 2018). Of course, additional analyses are always possible, and we would appreciate any help on this issue for further assessments.

*For riparian vegetation, do you have a mechanism for why it would change the trend in Tw? I understand that adding or removing vegetation would change the absolute value of temperature, but by what mechanism would the trends be affected? If the riparian temperature sections were to be retained, this should be addressed.*

We believe that the reviewer misunderstood this part of the analysis (which may have led to his previous comments) and we'll make sure to make the revised version clearer. In this analysis we compare Tw trends on headwater streams (in the same zone which implies same type of geology, proxy of groundwater contribution), with different degrees of shading,

resulting from vegetation and reach orientation. It is observed that effect of increase on radiative fluxes are less important on streams with high shading, creating resilience to warming for these type of rivers.

*To conclude this second part, I would recommend that the authors revise and strengthen the analysis of Ta vs Tw trends and the impact of Q. With the results presented, such strong statements as those in the title and abstract do not entirely hold water. I would also recommend mentioning in the title the location where the study was conducted (Loire, France). Perhaps some of my reservations stem from a lack of understanding of exactly how the models work and a better description might help to alleviate these reservations.*

The reviewer is right and we hope that our responses and proposed modifications to the manuscript put some light on the understanding of the models.

*Along with the emphasis on the main message that I recommend, perhaps some of the "secondary analyses" could be removed from the document. I think that a more thorough discussion of the limitations of the methods and results should be provided. I have not commented in detail on all the discussion, summary and conclusion sections, but certainly some of them are relevant to my comments above.*

We believe that the previous responses address these comments and that the analysis made on e.g. riparian vegetation may be interesting to readers as they suggest that increasing riparian vegetation might help reducing the impact of climate change on stream temperature (of course, independently on other consequences, notably on water use by such riparian vegetation).

*A final comment concerns the real added value of such a major modelling effort. A significant part of France is modelled; however, I feel that some of the results could be obtained by analysing only past measurements (and getting rid of all the modelling uncertainty).*

This comment is rather surprising for two reasons. First, the manuscript clearly states that much too few Tw observations are available for assessing long-term trends over the whole Loire basin. Second, the reviewer is the main author of a HESSD preprint precisely using physically-based models to assess the impact of climate change in Switzerland (Michel et al., 2021, https://doi.org/10.5194/hess-2021-194) and should understand why such a modelling effort may be needed, i.e. to infer future changes in Tw thanks to such models and climate projections (which is precisely the objective of an upcoming manuscript).

*The added value could come from the whole analysis of the riparian vegetation for example, but as mentioned these analyses need to be strengthened. One solution (but which would involve a lot of extra work), could be to first publish just the data set in a journal like ESSE (https://www.earth-system-science-data.net/), where all the modelling and validation aspects are discussed in detail, and then have an article in HESS focused on the data analysis only allowing for an in-depth analysis. This would also allow for elaboration on some aspects that are not yet discussed (e.g., elevation, impact of snow if relevant for this catchment, more detailed spatial analysis). I think this option would really increase the potential impact of the*

*significant modelling effort that has been made. But I would understand if the authoring team do not want to go through this extra work.*

We really thanks the reviewer for these suggestions, and we will take theme into account for further analyses, as this goes much further than the objective of the present manuscript.

*Minor comments:*

We agree with all comments except with ones to which we responded.

*P1L38 [Q] should be (Q)*
*P1L52-52 "in the face of a changing climate", maybe just say "to climate change"*
*P3L79-80 Maybe define exactly what HERs are, or give a reference*
*Figure 1 Add HERs "borders" also in left panel. In the manuscript, this figure is not vectorized and small text are really pixelized when zooming to reads them. Maybe provide a vectorized figure or a higher resolution bitmap figure. Add in the figure or caption the source for the maps shown*

**P5L101** *"bottom" should be "right"*
**P5EQ(5)** *Can't it be written in the more compact form max(SFLeft,SFRight)?*
**P8L200** *Why using log(Q) in this analysis? There ias writing mistalke. It will be modified. It is (Q).*
**P9L224** *What does IQR mean?*
**P9L226** *"r" is used here, while "R" is used in figures*
**Figure 2** *There is a "a" on the right below the colour legend*
**P11L267-269** *Please add a reference here to support this statement*
**P16L306-308** *How negative Q trends, just by themselves, suggest an effect on Tw?*

We believe that previous responses address this comment

 *"suggested" should be "suggesting".*

**P20L335-338** *First, despite it is clearly stated on the abstract of the paper Michel (2020) that: "The mean trends for the last 20 years are + 0.37 ± 0.11 ∘ C per decade for water temperature, resulting from the joint effects of trends in air temperature (+0.39 ± 0.14 ∘C per decade), discharge (−10.1 ± 4.6 % per decade), and precipitation (−9.3±3.4% per decade)", I think now that this paper does not show a real impact of Q on Tw, but rather a correlation between Q and Tw in summer. If I had to rewrite this paper today, I would not be so categorical (and this why I also question it in your paper). Second, when you say: "In contrast with our results, they found Tw trends lower than Ta trends due to influence of snow melt and glacier melt", this is not totally exact. Indeed, trends found in Alpine catchments are lower due to the mentioned effects. For the low-altitude catchments where snow plays no role, trends are indeed closer, but Tw trends remain slightly slower than Ta trends (compare Figures S17 and S18). However, on an annual basis, we are talking about a few tenths of a degree less in my article and a few tens of a degree more in yours, so taking into account all the uncertainties involved, I see no contradiction. In addition, different regions are studied.*

This comment is again rather surprising as the paper referred to by the reviewer (and signed by him) has been published quite recently. We clearly agree that the statement made in the referred paper is rather categorical, but the present manuscript conclusions are clearly much less categorical. Hence, our surprise is with respect to the comments on our main statement on Tw trends being stronger than Ta trends over the Loire basin.

***Introduction and Section 5.5*** *Nuclear plants cooling is never mentioned in the paper. This might not be relevant for the Loire (but Bustillo et al. (2014) mention some plants in the catchment), but in general in France the question of cooling nuclear plants in the future with increasing air and water temperature will be a real challenge and I think it is worth mentioning it (see e.g. Bourqui et al., 2011).*

Bustillo et al. (2014) mentioned that the impacts from the power plant is negligible in the Loire basin: "This effect can also be neglected, as the nuclear power stations are equipped with closed-circuit cooling towers that allow the heat to be dissipated directly into the atmosphere. Thus, the thermal input into the Loire River is notably low. Studies conducted by the electricity-generating authority (EDF) indicate that rises in daily temperature of the Loire River downstream from the Dampierre power station have a median of 0.1°C and a 90th percentile of 0.3°C, with the greatest increase being in winter."

Moreover, Moatar and Gailhard, 2006 mentioned: "Nuclear power stations are equipped with closedcircuit cooling towers, allowing the heat to be discharged directly into the atmosphere. Thermal waste into the Loire, essentially from purging the cooling towers, is very low. For example, studies carried out by EDF indicate that, 90% of the time, the daily temperature rise of the Loire downstream of the Dampierre power station is less than 0.3°C, the median rise being 0.1°C °C. Moreover, the greatest rise in temperature is in winter, which means that the statistics for summer increases result in even lower values. In this study, we therefore considered that temperature rises caused by nuclear power stations are of secondary importance in the trends that concerned us. This will be confirmed from the analysis of the series from Belleville, situated upstream of all the Loire power stations."

Also, trend at BelleVille, which is not influenced by nuclear power plant has the same magnitude as the other 3 stations located downstream nuclear power plant (see Figure 2).

To make this point more clear, the following sentence will be added in P21L368**: "Please note that the impacts from the power plant is considered negligible in the Loire basin according to Moatar and Gailhard, 2006, Bustillo et al., 2014. Moreover, the trend at Belleville, which is not influenced by nuclear power plant has the same magnitude as that at the other 3 stations located downstream nuclear power plants (see Figure 2), showing negligible influence of power plants."**

**P22L398** Shouldn't it be "small rivers"?

**References:**

Beaufort, A., Curie, F., Moatar, F., Ducharne, A., Melin, E. and Thiery, D., 2016. T-NET, a dynamic model for simulating daily stream temperature at the regional scale based on a network topology. *Hydrological Processes*, *30*(13), pp.2196-2210. https://doi.org/10.1002/hyp.10787.

Beaufort, A., Moatar, F., Curie, F., Ducharne, A., Bustillo, V. and Thiéry, D., 2016. River temperature modelling by Strahler order at the regional scale in the Loire River basin, France. *River Research and Applications*, *32*(4), pp.597-609. https://doi.org/10.1002/rra.2888.

Loicq, P., Moatar, F., Jullian, Y., Dugdale, S.J. and Hannah, D.M., 2018. Improving representation of riparian vegetation shading in a regional stream temperature model using LiDAR data. *Science of the total environment*, *624*, pp.480-490. https://doi.org/10.1016/j.scitotenv.2017.12.129.

Michel, A., Schaefli, B., Wever, N., Zekollari, H., Lehning, M., and Huwald, H.: Future water temperature of rivers in Switzerland under climate change investigated with physics-based models, *Hydrol. Earth Syst. Sci. Discuss.* [preprint], https://doi.org/10.5194/hess-2021-194, in review, 2021.

Wasson, J.G., Chandesris, A., Pella, H. and Blanc, L., 2002. Typology and reference conditions for surface water bodies in France: the hydro-ecoregion approach. *TemaNord*, *566*, pp.37-41. https://hal.archives-ouvertes.fr/hal-00475620/document

---

## Author Comment (AC2)

We thank the reviewer for very useful and generally positive comments. By addressing these comments, we believe the paper will be substantially improved, particularly with respect to refined focus and more detailed description of hydrological and thermal models. The reviewer comments are in italics and our responses are in normal font, the proposed text additions and modifications are in bold.

Please note that in the following, "P", "L" and "SM" stand for page number, line number and Supplementary Materials, respectively.

**#Reviewer 2:**

*General comments:*

*The only main concern I have is some lack of detail on the description of the hydrological and thermal models (see below). I also find that the manuscript is a bit too long, and I believe that its readability would increase if some of the less relevant results (say e.g. Figs. 6, 7, 9 and respective paragraphs commenting them) were moved to the supplementary material.*

We agree, and Figures 6 and 9 will be moved to SM.

*Main comments:*

*L. 95-112: Many details on the implementation of the EROS model are missing. What are the free parameters of the model? What method for model calibration was used? How were Nash-Sutcliffe index and overall bias combined in the objective function? What was the rationale of choosing sqrt(Q) instead of Q in the computation of the Nash- Sutcliffe index? Was the output of this model validated with respect to Q time series not included in the calibration dataset?*

We agree, and therefore the section "EROS hydrological model" (section 3.1.1), P4L95, will be modified as follows to include the principles of the EROS model, its input data, calibration process and variables, and applications in other studies:

**"The EROS semi-distributed hydrological model simulates the daily streamflow at the outlet of 368 sub-basins (ranges between 40 and 1600 km$^2$; mean drainage area = 300 km$^2$), designed to be as homogeneous as possible with respect to land use and geology. On each of these sub-basins, the water balance is modelled by a lumped model using three reservoirs and a routing function for propagation across sub-basins (Thiéry, 1988; Thiéry and Moutzopoulos, 1995; Thiéry, 2018). This hydrological model has already been used in several other studies including climate change impact studies (Ducharne et al., 2011; Habets et al., 2013; Bustillo et al., 2014).**

**The EROS model uses daily Ta (°C), solid and liquid precipitation (mm), and reference evapotranspiration (ET0, mm). Ta and precipitation are provided by the 8 km gridded Safran surface reanalysis from Meteo-France over the 1958–2019 period (Quintana-Segui**

et al., 2008; Vidal et al., 2010). ET0 was computed from Safran variables with the Penman-Monteith equation (Allen et al., 1998).

For calibration, 352 hydrometric stations with observed Q data were extracted from the French national Banque Hydro database (http://www.hydro.eaufrance.fr/) (Figure 1, right panel). Stations along the main Loire (14) and Allier (11) rivers are influenced by operations of 4 large dams for hydropower, flood control and low-flow management, notably through summer releases. Time series at these stations have been first naturalized by EDF (Électricité de France) based on variations in water storage in reservoirs due to operations based on target storage curves (Naussac, Villerest) and optimization of the hydropower energy generation for the electric grid needs (Grangent, Montpezat). EROS was then calibrated over the 1971-2018 period against daily Q at all 352 sub-basins with at least 10 years of daily observations. The calibration aimed at optimizing all unknown parameters (soil capacity, recession times, propagation times) through maximizing the Nash-Sutcliffe Efficiency (NSE) criterion on the square root of streamflow and minimizing the overall bias. Considering the NSE criterion on the square roots of the flows provides an estimate of model performance without favoring neither high flows nor low flows. The overall calibration criterion C is: C = mNSE – w.mRB where the weighting factor w is fixed at 0.05 .

A 3-year warm-up period (1971-1974) was discarded from the overall calibration period (1971-2018, which maximizes the number of streamflow observations). The calibrated model was then used to simulate streamflow over the whole 368 homogenous sub-basins in the Loire basin over the whole 1963-2019 period. Note that although meteorological variables are available from 1958 onwards, the first years (1958-1962) were discarded from the analysis to ensure the model convergence.”

Moreover, to avoid confusion (which happened for the other reviewer), P5L110 of the current manuscript will be removed and it will be added between P6L150 and L151 as following:

“- Streamflow: the daily streamflow simulated at the outlet of 368 homogeneous sub-basins by the EROS model (see section 3.1.1) are redistributed along the river network inside each sub-basin according to the reach drainage area for informing the T-NET model at the reach scale. The ratio of sum of the lengths of all reaches upstream of a reach to the sum of the lengths of all reaches located in a sub-basin is used as a proxy for the drainage area of a reach.”

*L. 113: If I understood correctly, the T-NET model does not have any free parameter, hence it does not require calibration. This should be stated explicitly.*

We agree. In this regard, the following paragraph will be added L90 in Section 3:

“We used two models to calculate stream temperature in the Loire River basin according to the method developed by Beaufort et al (2016). The first model is the semi-distributed hydrologic model EROS, which estimates daily streamflow at sub-basin outlets. The second model is the fully distributed, mechanistic temperature model Temperature-NETwork (T-

**NET; Beaufort et al. 2016 and Loicq et al. 2018) that uses streamflow from EROS and meteorological reanalysis data to estimate Tw at each reach in the Loire River basin. These models are briefly described below, and are detailed in Thiéry (1988), Thiéry and Moutzopoulos (1995), and Thiéry (2018) for EROS, and Beaufort et al. (2016) and Loicq et al. (2018) for T-NET."**

*LL. 115-119: The authors state that the first step for Tw estimation is the computation of the equilibrium temperature (Te). This is the stream temperature at which the net heat flux across the surface of the water body is null, and is a useful concept when one aims at producing a simplified, (semi-)mechanistic temperature model where Te is expressed e.g. as a linear or logistic function of Ta, which allows discarding the exact computation of the various non-advective heat fluxes (latent, short-wave radiation, etc..). However, here the non-advective heat fluxes are exactly computed (this is not explicitly mentioned here but is reported in Beaufort et al., 2016), and Te is calculated as the Tw value that nullifies the sum of non-advective heat fluxes. This is technically equivalent to a fully mechanistic model where non-advective heat fluxes are included in the energy budget at a reach (or cross-section) scale. Thus, I find it a bit confusing to invoke the concept of equilibrium temperature here.*

We agree, and in this regard, we will improve description of the thermal model in P5L115-116 as following:

**"TNET is a fully mechanistic, 1D model that simulates hourly $Tw_{i,j}$ at distance, i, along reach, j, by solving the local heat budget. The heat budget of each reach includes six fluxes (W m$^{-2}$): net solar radiation, atmospheric longwave radiation, longwave radiation emitted from the water surface, evaporative heat flux, convective heat flux, and groundwater heat inflow. Briefly, the model calculates the longitudinal change in Tw at time t (dTw/dx) for steady-state conditions in order to achieve thermal equilibrium (i.e., $\Sigma H_{i,j}=0$, where H is a heat flux), while accounting for confluence mixing. Detailed information about the T-NET model principles and calculation of six heat fluxes at the water-air and water-stream bed interfaces, thermal propagation were provided in Beaufort et al., 2016 and Loicq et al., 2018. Note that unlike the EROS hydrological model, the T-NET thermal model does not have any free parameter, hence, it does not require calibration and it is only validated."**

*L. 174: It is unclear why 72 stream temperature stations are mentioned here, but then validation is only performed on the 14 stations with long-term continuous data. Are the other 58 stations ever used in the analysis? If not, this information should be discarded.*

Indeed, all 72 stations were used for assessing model performance in terms of bias and RMSE while 14 of these stations with long-term data where used to assess the ability of the model to capture long-term data. To avoid this confusion, we will do some modifications to the text as follows:

L. 173-175: **"Therefore, model performance was assessed on 72 near natural Tw stations, which are weakly influenced by impoundments and with continuous daily data over the 2010–2014 period (see Fig. 1). These stations were identified using the thermal signatures approach that allows distinguishing between natural and altered thermal regimes"**

L.180-183**: "Long-term continuous data was available at 14 of the 72 near-natural stations, including 9 stations with 8-13 years data and 5 stations with 20-40 years data. They were used as a validation dataset for the seasonal and annual trend assessment (see Table 1, red points in Fig.1 for the location and Fig.S1 for annual Tw time series)".**

*Figure 2. It would be interesting to have an estimation of the performance of the T-NET model in terms of RMSE (or MAE), in addition to the mean bias (as the latter can obviously be close to zero even when absolute errors are very high).*

We agree. The following figure (including RMSE) will be added to the bias figure in the SM (Fig. S2).  The following results will be also added to P9L224:

**"The median RMSE of the T-NET thermal model, for small and medium rivers, ranges between 0.52 °C (Annual) and 0.91 °C (DJF and JJA) across seasons (see Figure S2, bottom left panel). For large rivers, the median RMSE of the T-NET thermal model ranges between 0.38 °C (annual) and 1.11 °C (JJA and SON) across seasons (see Fig. S2, bottom right panel). Moreover, 53-83% stations (resp. 50-100%) on small and medium (resp. large) rivers have a RMSE<1 °C across seasons."**

[Figure]

*LL. 260-261: "Indeed, where … for all seasons". This is an interesting observation, but, for the sake of fairness, one would need to check whether the majority of reaches where Ta trend > Tw trend also showed a decreasing Q trend.*

We agree. In this regard, we will modify Fig.5 as follows, which includes your suggested class, and two types of data (1) significant-non significant trends, and (2) only significant trends as requested by the other reviewer. With this modified Figure, P11L260-264 will be modified as follows:

**"Indeed, where Tw trends exceed Ta trends, decreasing Q trends occur coincidentally at the majority of reaches (43-94%) for all seasons (with the exception of winter) irrespective of whether all significant and non-significant trends are considered (Figure 5, top panel) or only significant trends of all three variables are considered (Figure 5, bottom panel). Of these specific reaches where all factors converge (trend in Tw higher than trend in Ta, and decreasing trend in Q), most are located in HER A irrespective of considering all significant and non-significant trends (52-90% of such reaches across seasons; see spatial figure, left panel in SM), or considering only significant trends of all three variables (100% of such reaches across seasons; see spatial figure, right panel in SM)."**

Please note that the spatial figures showing location of these reaches will be put in SM (between Figure S7 and S8 of the current manuscript).

[Figure]

Significant & non-significant trends    Significant trends

[Figure]

Trends
— Tw<Ta & Q>0
— Tw<Ta & Q<0
— Tw>Ta & Q>0
— Tw>Ta & Q<0

Spatial figures

*L. 274: "shifting by approximatively +2 °C". I find this unclear. Is the shift observed by comparing current Tw and Ta values with those observed at the change-points (~1988)? Please state this more clearly.*

We agree this was not clear. We will clarify the text on P13L273 to state that**:**

**"Tw and Ta anomalies exhibited clear negative-to-positive change-points in the late 1980s at nearly all reaches, with median values increasing by approximately +2 °C after the change-point.**

*LL. 288-289: Strahler order is significantly and positively correlated with Tw only in Spring for all HERs (and in summer only for HER A). The way this sentence is written, one is led to think that this correlation is strong more often than it is the case by looking at Fig. 9.*

We agree and it will be modified in the revised manuscript.

*Minor comments*

**We agree with all comments except the ones to which we have responded.**

*L. 58: "in meeting these goals"*

*LL. 92-93: I think it would be clearer to mention "meteorological seasons", and then use the season names in lieu of the acronyms DJF, MAM etc. in the figures. This would make the figures much more intuitive.*

*L. 108: if 1971-1974 is the warm-up period, then the calibration period should be 1974 (or 1975)-2018.*

*L. 109: Why is the number of subcatchments here (368) higher than 352 (mentioned in L. 104)?*

**There were observed Q at these 352 stations but after calibrating the hydrological EROS model at these 352 stations, the EROS hydrological model simulated Q at 368 sub-basins. This will be added to the revised manuscript.**

L. 112: "were discarded". Moreover, "(1958-1962)" is four (or five) years, not three years.

LL. 140-143: Eqs. (3-5) can be condensed in one equation: SF = max((W_shaded)_left * vc_left, (W_shaded)_right * vc_right)

L. 178: I suggest indicating the range rather than SD for drainage area values (when SD > mean, this does not make much sense)

L. 201: verb missing. Perhaps "used to detect/assess synchronicity".

L. 227: "The trends of both modeled and observed Q"

L. 247: "The highest (resp. lowest) Ta trend values"

L. 248: In Fig. S6, the season with lowest Ta values seems to be fall, not spring, with 0% of reaches showing values > 0.4 °C /decade.

L. 257: "The medians of Tw… than those of Ta"

L. 268: "they are either warm and wet"

L. 308: "strongly suggesting an effect on Tw" or similar

L. 319: "comparing trends… gives us a comprehensive view"

L. 364: "The warming effect… seems more significant"

L. 370: "an increase of >25%". Increasing from 15% to 40% (this is what I believe the authors are referring to, see LL. 299-301) is actually a 267% increase. Perhaps it would be best to reformulate as "increasing riparian shading from 15% to 40%".

---

## Author Response (AR1)

We thank the reviewer for very useful and generally positive comments. By addressing these comments, we believe the paper will be substantially improved, particularly with respect to refined focus and more detailed description of hydrological and thermal models (section 3 of the article). The reviewer comments are in italics and our responses are in normal font, the proposed text additions and modifications are in bold.

Please note that in the following, "P", "L" and "S" stand for page number, line number and Supplementary Materials, respectively.

**#Reviewer 2:**

*General comments:*

*The only main concern I have is some lack of detail on the description of the hydrological and thermal models (see below). I also find that the manuscript is a bit too long, and I believe that its readability would increase if some of the less relevant results (say e.g. Figs. 6, 7, 9 and respective paragraphs commenting them) were moved to the supplementary material.*

We agree, and Figures 6 and 9 of the first version of the manuscript was moved to S.

*Main comments:*

*L. 95-112: Many details on the implementation of the EROS model are missing. What are the free parameters of the model? What method for model calibration was used? How were Nash-Sutcliffe index and overall bias combined in the objective function? What was the rationale of choosing sqrt(Q) instead of Q in the computation of the Nash- Sutcliffe index? Was the output of this model validated with respect to Q time series not included in the calibration dataset?*

We agree. We responded to all these questions of the reviewer in the revised manuscript. Please see section 3.1.0 (P5L100-106), section 3.1.1, Figure S2, and section 3.2.1 in the revised manuscript.

Moreover, to clarify that simulated Q by EROS used in T-NET and there is no routing for simulating Q in T-NET, please see 3.1.0 (P5L100-106) and P6L161-165 ("Reach streamflow") in the revised manuscript.

*L. 113: If I understood correctly, the T-NET model does not have any free parameter, hence it does not require calibration. This should be stated explicitly.*

We agree. Please see P8L220-221 in the revised manuscript.

*LL. 115-119: The authors state that the first step for Tw estimation is the computation of the equilibrium temperature (Te). This is the stream temperature at which the net*

*heat flux across the surface of the water body is null, and is a useful concept when one aims at producing a simplified, (semi-)mechanistic temperature model where Te is expressed e.g. as a linear or logistic function of Ta, which allows discarding the exact computation of the various non-advective heat fluxes (latent, short-wave radiation, etc..). However, here the non-advective heat fluxes are exactly computed (this is not explicitly mentioned here but is reported in Beaufort et al., 2016), and Te is calculated as the Tw value that nullifies the sum of non-advective heat fluxes. This is technically equivalent to a fully mechanistic model where non-advective heat fluxes are included in the energy budget at a reach (or cross-section) scale. Thus, I find it a bit confusing to invoke the concept of equilibrium temperature here.*

We agree, and in this regard, we improved description of the thermal model in the revised manuscript. Please see P5L120-126 in the revised manuscript.

*L. 174: It is unclear why 67 stream temperature stations are mentioned here, but then validation is only performed on the 14 stations with long-term continuous data. Are the other 58 stations ever used in the analysis? If not, this information should be discarded.*

Indeed, all 67 Tw stations were used for assessing model performance in terms of bias and RMSE while 14 of these stations with long-term data where used to assess the ability of the model to capture long-term data.

To avoid this confusion, we did some modifications to the revised manuscript. Please see P8L219-225 in the revised manuscript.

*Figure 2. It would be interesting to have an estimation of the performance of the T-NET model in terms of RMSE (or MAE), in addition to the mean bias (as the latter can obviously be close to zero even when absolute errors are very high).*

We agree. We also considered RMSE in the revised manuscript. Please see P8L219-221, P11L278-285, and Figure S9 in the revised manuscript.

*LL. 260-261: "Indeed, where … for all seasons". This is an interesting observation, but, for the sake of fairness, one would need to check whether the majority of reaches where Ta trend > Tw trend also showed a decreasing Q trend.*

We agree. In this regard, we modified Figure 5 and it was provided as Figure 6 in the revised manuscript. Please also see P13L324-335.

*L. 274: "shifting by approximatively +2 °C". I find this unclear. Is the shift observed by comparing current Tw and Ta values with those observed at the change-points (~1988)? Please state this more clearly.*

We agree this was not clear. P15L343-345 in the revised manuscript clarified it**:**

**"Tw and Ta anomalies exhibited clear negative-to-positive change-points in the late 1980s at nearly all reaches, with median values increasing by approximately +2 °C** **after the change-point.**

*LL. 288-289: Strahler order is significantly and positively correlated with Tw only in Spring for all HERs (and in summer only for HER A). The way this sentence is written, one is led to think that this correlation is strong more often than it is the case by looking at Fig. 9.*

We agree and we modified in the revised manuscript. Please see P17L357-362.

*Minor comments*

**We agree with all comments except the ones to which we have responded.**

*L. 58: "in meeting these goals"*

*LL. 92-93: I think it would be clearer to mention "meteorological seasons", and then use the season names in lieu of the acronyms DJF, MAM etc. in the figures. This would make the figures much more intuitive.*

We believe that it would be better to use the acronyms in the figures as seasons are different in northern and southern hemisphere.

*L. 108: if 1971-1974 is the warm-up period, then the calibration period should be 1974 (or 1975)-2018.*

*L. 109: Why is the number of subcatchments here (368) higher than 352 (mentioned in L. 104)?*

There were observed Q at these 352 stations but after calibrating the hydrological EROS model at these 352 stations, the EROS hydrological model simulated Q at 368 sub-basins. Please see section 3.1.1 and 3.2.1 in the revised manuscript.

L. 112: "were discarded". Moreover, "(1958-1962)" is four (or five) years, not three years.

LL. 140-143: Eqs. (3-5) can be condensed in one equation: SF = max((W_shaded)_left * vc_left, (W_shaded)_right * vc_right)

L. 178: I suggest indicating the range rather than SD for drainage area values (when SD > mean, this does not make much sense)

L. 201: verb missing. Perhaps "used to detect/assess synchronicity".

L. 227: "The trends of both modeled and observed Q"

L. 247: "The highest (resp. lowest) Ta trend values"

L. 248: In Fig. S6, the season with lowest Ta values seems to be fall, not spring, with 0% of reaches showing values > 0.4 °C /decade.

L. 257: "The medians of Tw… than those of Ta"

L. 268: "they are either warm and wet"

L. 308: "strongly suggesting an effect on Tw" or similar

L. 319: "comparing trends… gives us a comprehensive view"

L. 364: "The warming effect… seems more significant"

L. 370: "an increase of >25%". Increasing from 15% to 40% (this is what I believe the authors are referring to, see LL. 299-301) is actually a 267% increase. Perhaps it would be best to reformulate as "increasing riparian shading from 15% to 40%".

We thank the reviewer for his very useful comments. By addressing these comments, we believe the paper will be substantially improved, particularly with respect to a refined focus and a more detailed description of hydrological and thermal models (section 3 of the article). The reviewer comments are in italics, our responses are in normal font, and the proposed text additions and modifications are in bold.

Please note that in the following, "P", "L" and "S" stand for page number, line number and Supplementary Materials, respectively.

**#Reviewer 1:**

General comments:

*The submitted paper presents a large-scale modelling effort of river discharge and temperature for the Loire River basin in France. The modelling is carried out in two steps, first for discharge and then for water temperature. The models are forced with reanalysis data provided by Météo-France and validated on a large number of measurement stations. The model results are then used to analyse and discuss the trends in discharge and water temperature, the factors influencing the modelled trends in water temperature, the spatial patterns of the resulting trends, and the role of riparian vegetation and Strahler order.*

*This article presents an important work and a valuable contribution to his field of research. In general, the paper is well written in terms of language and has clear and nice figures.*

The authors would like to thank the reviewer for this statement.

*However, I have some concerns which I detail below. The first concern is the clarity, completeness, and organisation of section 3 "Method and data", where, in my opinion, some crucial information is missing.*

We believe that the responses to the following comments address this comment.

*The second concern is the robustness of some of the results, in particular the validation of the model and the strength of some of the analysis performed. Indeed, the results and discussion sections tend to analyse many different aspects and perhaps only the most robust ones could be kept. In addition, since quite simple models are used, I would be more careful for some interpretation of the results.*

We believe that the responses to the following comments address this comment.

*I also regret that no data or models are shared along the article. I imagine that sharing of forcing data may not be allowed, however I would strongly encourage the addition of a "Data*

*and code availability" section to detail how this work can be replicated. If possible, sharing with the community the time series produced by the model would also be appreciated. In the same vein, the supplementary material could be improved to provide interested readers with a more complete overview of the large number of results obtained.*

We completely agree on the data sharing idea. Please look at the "Data and code availability" section in the revised manuscript.

Please note that we indeed have no right to publish the Safran meteorological reanalysis data belonging to Météo-France.

*Main comments*

*1.1 Data description*

*My first recommendation is to create a "Data" section (with sub-sections) where all details on forcing, validation and geographical data are grouped. Currently, the information is spread over P3L83-86, P4L97-P5L104, P5L118-121, L123-124, L125-128, P7L165-167 (I may have forgotten some). Grouping them together would make for easier reading as many details now appear in the models' description sections.*

Please first note that P3L83-86 is about HERs (Hydro EcoRegions) that are not input of any of the models. This confusion has been repeated in other comments by the reviewer. They are simply regions within which land use/land cover, and climatic conditions are rather homogeneous. They were developed for the Water Framework Directive to give ecological reference conditions. Such information has been already specified in P3L80 of the first version of manuscript. HERs were used in the current study to be able to consider the similar landscape characteristics for reaches belonged to the same HER. Thus, it is used to compare reaches with the same landscape characteristics together like what has been done in assessing the influence of reach size and vegetation cover. Moreover, it helps to report and compare the trends over the basin in a simple way at this large scale. To avoid this confusion, we modified this part in the revised manuscript. Please see P3L82 in the revised manuscript.

Please note that the rest of the data pointed out by the reviewer (*P4L97, P5L118-121, L123-124, L125-128, P7L165-167*) are the part of inputs of the hydrological and thermal models. We believe that we cannot talk about such data before explaining each model. Nevertheless, in the revised manuscript, we rearranged the sections 3.1 and 3.2 and tried to better explain principles, input data, and calibration and validation dataset. In this regard, please see sections 3.1 and 3.2 in the revised manuscript.

*In addition, a comprehensive table of all stations used for calibration and validation should be provided in the supplementary material (SM later), with coordinates and station names, and also indicating the data provider.*

We agree. The data source of hydrometric stations used for both calibration and validation was already mentioned in P5L100 of the first version of manuscript. Nevertheless, the data source of hydrometric stations and a list of these stations with coordinates was provided in P7L180 and Table S2 in the revised manuscript.

The data source of Tw stations and a list of these stations with coordinates was provided in P8L216-219, Table 1 and Table S5 in the revised manuscript.

*The coordinates should be added in Table 1. For long-term stations, one could even consider numbering them in the table and then indicating the numbers on Figure 1.*

We agree. Please see the Table 1 and Figure 1 in the revised manuscript.

*Maps similar to Figure 3 and S5, but showing the annual and seasonal average of Tw, Ta and Q over the whole catchment area could be a useful addition to the SM to capture the different local conditions in the catchment.*

We agree. Please see the Figures S1 and S5 in the revised manuscript.

*A map similar to Figure 2, middle panel, with a colour indicating the Strahler order could be added in the SM.*

We agree. Please see the Figure S3 in the revised manuscript. Such information can also be seen on the Figure S5 in the revised manuscript.

*For non-French readers, a map showing the location of the Loire basin in France (or in Europe) should be added. It could be integrated as a new panel in figure 1.*

It has been already presented in the first version of the manuscript, in the top right corner of Figure 1, left panel. Nevertheless, it was made explicit in the corresponding caption in the revised manuscript.

*In P5L103 it is said that the Q time series are "naturalised". However, no reference or details of the procedure used are provided. This information should be included because, as you note later in section 5.4, anthropogenic disturbances are of major importance.*

We agree. Please see P7L181-185 in the revised manuscript.

*In addition, I did not find in the document the source of the time series for water temperature. Is it the same supplier as for the discharge? Is it also "naturalized"?*

The data providers of stream temperature were already mentioned in P23L410 of the first version of the manuscript. However, as mentioned before, the data source of Tw stations and a list of these stations with coordinates was provided in P8L216-219, Table 1 and Table S5 in the revised manuscript.

Please note that Tw data are raw observations, i.e. not "naturalized", as already made explicit in the manuscript P7 L172 of the first version of the manuscript. However, stations with near-natural Tw data was used in the current study. Please see P8L209-213 in the revised manuscript.

*1.2 Description of the EROS model*

*Only a few details are given on how the model actually works. In addition, the two main references given for the model are in French. Although it is not necessary here to describe all the details of the models, the main points should be provided to the reader. In particular, details of the mass balance should be given. Is it precipitation - evaporation, or can some of the mass be lost through deep soil infiltration? How is the water transported through the soil? Does EROS use a reservoir model? Finally, is there any routing of water into the stream network carried out in EROS or is only the release of water at the sub-catchment scale simulated? P5L109-111 suggest that routing is done in T-NET, but this is not clearly stated, nor is it mentioned in section 3.1.2. On the other hand, section 3.2.1 suggests that Q is obtained directly from EROS, which means that the routing is done in EROS.*

*It is not mentioned which parameters are calibrated (this should be stated), which values are tested for calibration, and which values are finally chosen. This should be stated in the SM to allow reproducibility. P5L105-107 states that: "The calibration aimed at maximizing the Nash-Sutcliffe efficiency criteria (Nash and Sutcliffe, 1970) on the square root of streamflow and minimizing the overall bias, in order to simulate correctly the whole range of Q values". How are the two metrics combined (NSE and bias) to assess the quality of the calibration? Why is the square root used? How many calibrations are performed, how are the calibration values chosen (pure chance or more advanced algorithm)? Again, the only non-French peerreviewed source provided is Thiéry (1988). This paper describes a water level model for an unconfined aquifer and I imagine that EROS is a model based on this work, but rather different from this 1988 version.*

*Overall, more details about the model are needed to enable the reader to understand how the EORS model works. The main assumptions and principle of the model should be stated, as well as details on the calibration procedure and parameters (a part can be added to the SM). I have found many applications of this model in the literature; these could also be cited as application examples.*

We agree. We responded to all these questions of the reviewer in the revised manuscript. Please see section 3.1.0 (P5L100-106), section 3.1.1, Figure S2, and section 3.2.1 in the revised manuscript.

Moreover, to clarify that simulated Q by EROS used in T-NET and there is no routing for simulating Q in T-NET, please see 3.1.0 (P5L100-106) and P6L161-165 ("Reach streamflow") in the revised manuscript.

*There is also no mention of the other input parameters of the model. The authors describe three different HydroEco regions (HER), but there is no information on how the region influences the model parameters (and thus the model outputs).*

Please see response to a comment above. HERs (Hydro EcoRegions) are not input of any of the models

*1.3 Description of the T-NET model*

*Here too, important pieces of information are missing. This is even more problematic than for the EROS model as no references are given for this model.*

We actually did provide references for the T-NET model P5L120 of the first version of the manuscript. Nevertheless, we clarified this point in the revised manuscript. Please see section 3.1.0 (P5L100-106) in the revised manuscript.

*P5L115-116 says: "To simulate Tw, the equilibrium temperature (Te) is first computed, the temperature at which the net heat flux across the surfaces of the stream is null". Then details of how some of the energy fluxes are calculated are given, but the above sentence is never followed by a "then" or "next". So, if I understand correctly, the temperature of the water being modelled is the equilibrium temperature? Or are you using a formulation similar to Bustillo's (2014) eq. (6). A crucial piece of information is missing here. And if only Te is used, this critical assumption and its consequences need to be discussed.*

We agree, and in this regard, we improved description of the thermal model in the revised manuscript. Please see P5L120-126 in the revised manuscript.

*In section 3.1.1 it says that EROS is used to calculate discharge over 368 sub-catchments, whereas in section 3.1.2 52'278 reaches are mentioned. So, there are several reaches per sub-catchment and within each sub-catchment, the water supply simulated by EROS is distributed to the reaches using the drainage area? But again, how and in which model is the routing calculated? P6L156-157 clearly states that "Q is the daily mean streamflow provided by the EROS model". But how can this be done if there is no routing in EROS? P7L160-161 explains that the travel time is calculated in T-NET, but this information is not used for the calculation of the water temperature (at least that is my understanding), but would be mandatory information for routing.*

We believed that responses to previous comments address these comments as well.

*No details are provided on how sensible, latent and groundwater heat fluxes are calculated. Equations or references should be provided.*

We believe that the modifications proposed in the previous comments address this comment. Please see P5L120-126 in the revised manuscript.

*I understand that no calibration is done for TNET, is this correct?*

Yes, there is no calibration for T-NET. Please see P8L220-221 in the revised manuscript.

*Are there any other parameters used in the model? For example, as with EROS, how are the properties of HERs taken into account in the model? The results are discussed for the different HERs, but this discussion only makes sense if the HERs are somehow parameterized in the model.*

We believed that responses to previous comments address these comments.

*Finally, in addition to my questions about routing, there is no information about reach-to-reach heat advection. Is there reach-to-reach heat transfer? If not, this would considerably weaken the analysis carried out in terms of the Strahler order and of the whole model in general. To conclude this first part of the discussion, I would really encourage the authors to add a few paragraphs better detailing the data sources, the model workflow, and the main assumptions of the models. As I say below, it would also be important to indicate the limitations of the models.*

Indeed, there is an upstream-downstream thermal propagation in T-NET thermal model. Please see Beaufort et al., 2016 page 5.  Moreover, we believe that the modifications proposed in the previous comments address this comment.

2.1 Calibration and validation of EROS and T-NET

*Firstly, details of the calibration procedure and parameters should be provided (see above). Secondly, why is no validation period used to infer the quality of the calibration? Indeed, time series are usually divided into a calibration period and a validation period. By using only a calibration period without validation, we have no information on the potential overfitting of the model at the calibration station during the calibration period. Depending on the modelling effort required, I would strongly recommend recalibrating the model over a shorter period and using a few years for validation.*

Modifications related to EROS model proposed in the previous comments, were already dealt with these comment. Please see section 3.2.1 in the revised manuscript.

*Furthermore, looking at Figure 3 of Thiéry (1988), we see a clear decrease in the performance of the model during the validation period.*

Several points should be noted here: first, in Thiéry (1988), the period was much shorter (11 years) leading to a necessarily less robust (with respect to interannual varaibility) model response. Second, the assessment is here done on aquifer levels, not streamflow like in the present study. Third, validation results in Thiéry (1988) are quite satisfying with the exception of two sites impacted by one specific precipitation input, i.e. the station observation at Abbeville, which may point to a possible deficiency in these inputs. These inputs are indeed different from those used in the present study, which are provided by the reference Safran gridded surface reanalysis over France (Vidal et al., 2010).

*In section 3.2.1, NSE on Q, ln(Q) and sqrt(Q) are mentioned, does this correspond to the "the mean NSE criteria for low, medium and high flows" mentioned in P9L221?*

Yes. We used the same configuration (Q, ln(Q) and sqrt(Q)) in the results instead of using low, mean and high flows in the revised manuscript. Please see P11L275-276 in the revised manuscript.

*Little detail is given on the quality of the calibration of Q. Indeed, only the mean NSE is given (no details on the variance), a graph of the distribution of values should be added in SM, as well as a graph showing the simulated and measured time series for some stations. In Figure S2, the bias is only shown for the 44 NHN stations, why not show all 352 stations (using a shorter time period)?*

We agree and we provided more figures in the revised manuscript showing the performance of EROS model. Please see P8L200-207, P10L270-277, Figures S6, S7 and S8 in the revised manuscript.

*In Section 4.1, it is shown that most of the simulated summer trends are not significant and that the simulated summer trends are poorly correlated with the observed trends. However, these summer trends are used extensively later in the document (e.g. Figure 5, Table 2). Given the uncertainty around these trends over the calibration period, I doubt that they are robust enough to perform such an analysis.*

We discussed about low correlation between observed and simulated trend found in summer in the revised manuscript. Please see P19L374-379 in the revised manuscript.

*For Tw, only the performance in terms of trends and biases is presented. A presentation of performance in terms of mean square error would also be informative in assessing the performance of the model as this metric is commonly used in the literature.*

We agree. We also considered RMSE in the revised manuscript. Please see P8L219-221, P11L278-285, and Figure S9 in the revised manuscript.

*The bias is discussed for Q and Tw. However, as the subsequent analysis is mainly trend based, and bias has no impact on trend, I am not sure of the relevance of this metric here.*

We agree that a model with bias can also simulate temporal trends well. Nevertheless, the first step of using a model is to assess its biases. Please note that the model performance in simulating seasonal Tw and Q was not assessed before in Beaufort et al., 2016. Moreover, such a comment is rather surprising after the previous one requesting an assessment based on the RMSE, which is highly influenced by any bias.

*Section 5.1 does not add much to section 4.1. I would like to see in these sections more discussion of where the models are underperforming and therefore the limitation of the dataset obtained. Indeed, the models used, like all models, are not perfect. Identifying the limitations and thus focusing the analysis and discussion on the part that proved to be within the radius of validity of the models is a mandatory step in modelling studies.*

We agree and the limitations of the models was discussed in the section 5.1 in the revised manuscript. Please also note that the section 5.4 also discusses about the limitation of models outputs. Indeed both our hydrological and thermal models produced natural regimes (see P5L9-113 and P8L209) and therefore, the influence of impoundments on both magnitude and trends at regulated streams cannot be assessed.

*Finally, the part concerning riparian vegetation seems to be a new addition to the model in this article. My concern is that no validation is shown. The approach should be validated using a Tw measurement station in a shaded area and compare the model performance with and without shading. In the absence of validation and evaluation of the effectiveness and limitations of the approach, I find it difficult to proceed with the analysis.*

T-NET improvement by this riparian shading method was already shown by Loicq et al., 2018. This was explicit in the revised manuscript. Please see P6L154-155 in the revised manuscript.

*2.2 Link between Ta, Tw, and Q*

*The comparison of Ta and Tw trends and the potential impact of Q are widely discussed. I appreciate this effort and think that there is still much to understand about the interaction*

*between Tw and Q.*

The authors thank the reviewer for this statement.

*First of all, great importance is attached to the finding that Tw increases faster than Ta for most seasons and on an annual basis. This result has already been found in some regions (see e.g., Webb and Nobilis, 2007; and Arora et al., 2016), but is in contradiction with other studies (see e.g., Moatar and Gailhard, 2006; Orr et al., 2015; and Michel et al., 2020). I believe this result would merit further discussion to assess its strength.*

We agree that such findings depend on the study area and even on the study period. In this regard, we decided to specify the study area in the new manuscript as following: "**Regional, multi-decadal analysis on the Loire River basin reveals that stream temperature increases faster than air temperature**".
Moreover, in the revised manuscript, we better discussed Tw and Ta trends observed in the other regions like what the reviewer pointed out. Please see section 5.3 in the revised manuscript.

 Please also note that in Moatar and Gailhard, 2006, Tw increases faster than Ta in spring and summer and at annual scale for the whole 4 stations on the Loire River.

*The main factor used to explain why the trends in Tw are more important than those in Ta is the discharge. However, trends in other forcing variables should be shown. Indeed, in winter, Figures 3 and S5 show that the variable explaining why TW trends > TA is probably not discharge (since no significant trend in discharge is found).*

We agree with the reviewer that many factors affect the spatio-temporal variability of Tw. However, in the current study, to understand variability in Tw trends, we considered Ta as a proxy for heat fluxes and meteorological variables, and Q as a proxy for thermal inertia and hydraulic geometries. These points were made explicit P9L239-247 in the revised manuscript.

Moreover, please note that agreed with the reviewer, we already excluded winter from this statement (P11L260 of the first version of the manuscript). Please also see P13L324-334 in the revised manuscript.

*Similarly, summer and spring show marked negative trends in discharge in some parts of the catchment, whereas these are the seasons where the trends in Ta and Tw are most similar. The addition of the seasonal discharge trend in the boxplot in Figure 4 would facilitate the analysis.*

First of all, please note that Figure 5 is Figure 6 in the revised manuscript.

Figure 3 clearly shows reaches in the upstream of the basin for which Ta trend (middle panel) are in range of 0.4-0.6 °C/decade (in orange) while Tw trends are >0.6 °C/decade (in red), so we do

not agree that Tw and Ta trends are similar in spring and summer. Yes, the median values across the basin for Tw and Ta trends is similar (Figure 4), but the spatial pattern is clearly not, hence the added value of Figure 3. The difference between Tw and Ta trend at reach scale can also be seen in Figure 5 in the revised manuscript. Please note that, Figure 4 compares dispersion and median values of Tw and Ta trends across the basin and not reach by reach while the modified Figure 5 (shown as Figure 6 in the revised manuscript) compares trends reach by reach. Figure 4 is just the first step to see whether Tw are more spatially variable than Ta trends. It therefore suggests that other factors than Ta are affecting Tw trends. Looking at the spatial pattern of trends in Figure 3 suggests spatial links between an additional increase in Tw and a decrease in Q, mainly in the upstream part of the basin (considering both significant and non-significant trends). Finally, the modified Figure 5 (shown as Figure 6 in the revised manuscript) compares Tw, Ta trends and the sign of Q trends reach by reach to test the hypothesis of decreasing Q as a possible controlling factor. Indeed the modified Figure 5 (shown as Figure 6 in the revised manuscript) clearly shows that regardless of the significance level, for the majority of reaches with Tw trends> Ta trends, a decrease in Q occurs coincidentally across seasons, with again the exception of winter. Moreover, it also shows that wherever Tw trends< Ta trends, the increasing Q trend occurs coincidentally.

Please see P13L322-335 for new result found by modifying Figure 5 (as Figure 6 in the revised manuscript).

*Figure 5 is used to support the hypothesis that Q is the main driver, see P11L259-263: "Overall, Tw trends were more spatially variable than Ta trends, suggesting the conditional influence of Q trends (Fig. 4). Indeed, where Tw trends exceeded Ta trends, decreasing Q trends occurred coincidentally at the majority of reaches for all seasons – with the exception of winter – (43-72 %, depending on season; Fig. 5. Of these specific reaches where all factors converged (trend in Tw higher than trend in Ta, and decreasing trend in Q)" (please note that there are some grammatical problems in the second sentences). There are many shortcuts here. Firstly, the larger scatter suggests that factors other than Ta have an impact on water temperature, but this does not in itself show that Q is responsible, it could be any other forcing variable. Secondly, and more importantly, Figure 5 shows the percentage of reaches where "all factors converge", which I interpret as "all trends are significant".*

First of all, in the paragraph that the reviewer referred to, we already explained what "all factors converge" means by adding "(trend in Tw higher than trend in Ta, and decreasing trend in Q)" after the sentence.

We of course acknowledge that other factors are at play in the trends, and this is also commented in the Discussion and Method sections. However, as it was also mentioned in the previous comment, in the current study, we considered Ta as a proxy for heat fluxes and meteorological variables, and Q as a proxy for thermal inertia and hydraulic geometries (P9L230-242 in the revised manuscript), and we tried to understand variability in Tw trends with respect to these selected variables.

The modified Figure 5 (shown as Figure 6 in the revised manuscript) is the strong supporter of our hypothesis that increasing Tw can be due to the joint effects of increasing Ta and decreasing Q. We do not claim that this is the main driver. At this point, we just stated what we observed as the reviewer also referred to "…decreasing Q trends occurred coincidentally at the majority of reaches for all seasons". P23L434-441 in the revised manuscript made explicit that we did not assess the causality relationship here. We just assessed the spatial and temporal links between Tw and Ta and Q as the main hydroclimate drivers of Tw.

*Please clarify this point, as when comparing Figures 5 and 3, I rather understand that all trends are included in Figure 5. Including non-significant trends would be a significant bias here, as many non-significant trends are just above or just below zero and the figures are based on trend signs. For Figure 5 to be complete, the distribution between Q>0 and Q<0 should also appear in the blue part. Indeed, the figure now shows that Tw>Ta in most cases when Q<0, however Figure 4 shows that Tw>Ta in most cases anyway.*

We agree and in the revised manuscript, we modified caption of Figure 3, and we also modified the Figure 5 (shown as Figure 6 in the revised manuscript).
Again, please note that Figure 4 is pool of all reaches and so looking at this figure, we cannot say anything about reach by reach trends, and consequently, it is not possible to compare Figure 4 and Figure 5 (shown as Figure 6 in the revised manuscript).
The Figure 5 in the revised manuscript clearly shows that Tw trends are greater than Ta trends for the majority of the reaches across seasons except for the summer when it is the case for the half of the reaches.

*This figure could be used to show the impact of Q if, and only if, we can see that the proportion of Ta>Tw vs Ta<Tw changes if Q>0 and Q<0. Also, the number of catchments used probably differs significantly between seasons (at least if only significant trends are shown). Here I see that there may be something irreconcilable in this figure:*
*- Either all reaches are kept, including those with insignificant trends, but then the figure itself will lose much of its meaning for the reason mentioned above.*
*- Either only significant trends are retained, but only a small minority of reaches show significant Q trends in certain seasons, and the figure would then only show a small subset of reaches.*

These points were already considered in the modified Figure 5 (shown as Figure 6 in the revised manuscript), and what we stated is still valid. Please see again P13L323-335

*This question of the relationship between Tw and Q trends is not straightforward (you can look at the introduction of Arora et al. (2016) for a good review of the literature available by then) and, it is important to note, seeing a correlation between Tw trends and Q trends does not imply any causality.*

Thanks you for this suggestion, but we are already aware of the paper referred to here as we already cited it several times in the manuscript. And of course, we are aware that correlation does not imply causality, hence our position only suggesting this causality, as it is supported by coherent spatial patterns, temporal patterns, and well-understood physical processes. The aim of this article was just to find out the magnitude of trends in Tw, and possible explanations for discrepancy found between Tw trends and Ta trends by assessing the spatial (Figures 3, 4, 5 and 6) and temporal links (Figures 7 and 8, and Table 2) between trends in Tw and Ta and/or Q. P23L434-441 in the revised manuscript made explicit that we did not assess the causality relationship here. We just assessed the spatial and temporal links between Tw and Ta and Q as the main hydroclimate drivers of Tw.

*This is what we see in the subsection "Synchrony of annual anomalies" and in Figure 8: low Q summers correspond to high Ta summers, so it is difficult to assess which of the two, or the combination, leads to an increase in Tw.*

As you are probably aware, summer streamflow does not only depends on summer temperature (through summer evapotranspiration), but also from baseflow originating from precipitation in the previous months, during the recharge season. Distinguishing between Ta and Q factors on Tw in this study derives from that they are the two main inputs of the T-NET model. Of course as it is mentioned in P23L434-441 in the revised manuscript, one could devise a formal attribution framework where one may e.g. remove trends in Q and trends in Ta alternatively in T-NET inputs, but this would require much more work than the objectives set up for this manuscript.

*Furthermore, Figures S10 and S11 suggest that the negative discharge trend is caused by an increase in ET rather than a decrease in P. The regions concerned (see Figure 3), are those where the increase in Ta is most significant. Thus, the causal chain here appears to be Ta increasing → ET increasing → Q decreasing. So, even if Q is shown to be a factor influencing Tw, it originates (mostly) in the increase in Ta. Thus, it might be misleading to say that a decrease in Q is a contributing factor, as I think that for most readers a decrease in Q would mean a decrease in precipitation, whereas here no significant decrease in precipitation is found, and thus no impact of the precipitation regime on Tw can be assessed. The real impact of ET can be assessed by comparing the measured trends of discharge to the trends on measured precipitation in the catchment, in order to confirm if the increase in ET modelled is correct.*

We refer the reviewer to the response to a pervious comment on summer baseflow originating from previous recharge season. The causal chain presented by the reviewer may be effective when looking at the annual scale, but this is made much more complex with the seasonality in T and P, and of course the fact that catchment integrate these signals in time across seasons. Moreover, we find it surprising that the reviewer suggests that " it might be misleading to say that a decrease in Q is a contributing factor" while a recent paper he was the main author of actually says right in the abstract "The mean trends for the last 20 years are $+ 0.37 \pm 0.11$ ∘ C per decade for water temperature, resulting from the joint effects of trends in air temperature (+0.39

± 0.14 ◦ C per decade), discharge (−10.1 ± 4.6 % per decade), and precipitation (−9.3 ± 3.4 % per decade).” (Michel et al., 2020). Such an assessment goes much further in the attribution of Tw trends than we aim to, while we only gather clues (spatial patterns, temporal patterns, physical reasoning) towards such a still distant formal attribution.

*With all this discussion, I really question this sentence in the abstract: “Importantly, air temperature and streamflow exerted joint influence on stream temperature trends, where the greatest stream temperature increases were accompanied by similar trends in air temperature (up to +0.71 °C/decade) and the greatest decreases in streamflow (up to -16 %/decade)”. Indeed, as discussed above the discharge decrease might just be a “side-effect” of the increase in Ta through increased ET, but since it is also the region with the highest Ta increase (P16L229), I don’t think that we can conclude that “air temperature and streamflow exerted joint influence on stream temperature”.*

Again, according to our responses to the previous comments, we respectfully disagree with the reviewer and at the same time agree with Michel et al. (2020) which asserts such a joint influence while we clearly only bring forward here this joint influence.

*Continuing with the subsection "Synchrony of annual anomalies", I do not really see the added value of the change point analysis. Furthermore, as it seems to illustrate an abrupt change in the late 1980s, I would think that a trend analysis is then not appropriate for these time series.*

This is a good remark indeed. However, trend assessment and change-point assessment are definitely not incompatible, and both analyses bring forward their own contribution of our understanding. More specifically here, the trend analysis shows the increase/decrease of the variables and the change-point analysis critically brings some more information on the relationship between Ta, Q, and Tw in their temporal synchronicity, again towards a still distant attribution.

Moreover, please note that a time series can have both a change-point and a trend. A visual assessment can clearly shows this point (e.g. see Tw and Ta anomalies in Figure S17 in the revised manuscript).

*Going back to the Tw>Ta question more generally (sorry, all the variables are related and I had trouble organising my comments), it would be interesting to calculate trends at long-term water temperature stations to see if the same trend is observed, to reinforce the results.*

It is a good idea, but we do not have Tw long-term stations in the upstream part of the Loire catchment (HER A) where mostly Tw trend> Ta trends (see Figure 3 and modified Figure 5 above). We have long-term data for the Loire River in the downstream part of the basin, which have already been analyzed in this study and previous ones (Moatar and Gailhard, 2006; Avarello et al., 2020). Nevertheless, consistent with our findings, Moatar and Gailhard, 2006

found Tw trends > Ta trends in spring, summer and at the annual scale for the whole 4 stations on the Loire River used also in the current study (see Table 1).

Moreover, it would be nice to work with observations and have less uncertainties about the results of simulations, but as seen in Table 1, only 5 stations have long-term Tw data over the whole basin. In France, the National Water temperature network was established only in 2009 (https://hubeau.eaufrance.fr/page/api-temperature-continu#/). We are facing lack of long-term and detailed observed Tw in the basin, and the point of developing models like T-NET model was precisely to overcome this lack of data.

*In addition, a map like the one in Figure 3, but showing the difference between Ta and Tw, would be very informative about the spatial distribution of catchments where Ta trends are more important than Tw and help the comparison with discharge trends.*

We agree. In this regard, please see Figures 5 and S16 in the revised manuscript.

*This result receives a lot of attention in the article (see for example the title). The details (Figure 4) show that this is mainly due to the winter and autumn seasons (although the difference is statistically significant in spring, it is still really small).*

Please note the statement is mainly based on the reach by reach assessment i.e. Figures 5 and 6 in the revised manuscript not Figure 4 which only compares median and spatial variability of trends in Tw and Ta. Moreover, the fact that Tw trends are stronger than Ta trends is also valid when median over the basin is considered (Figure 4) or for reach by reach assessment (Figures 5 and 6) at the whole annual scale. We therefore believe that the title is appropriate.

*However, Figure S4 and the indicated R coefficient shows that the simulated trends in autumn have the lowest correlation with the measured trends (this is also stated in P9L232), so autumn is the season where we have the lowest confidence in the results. In any case, I would condition the general statement of Tw>Ta on the seasonal aspect since it is not general (and I would add this information in the title) and I would really stress the uncertainties about it. This result is interesting, and certainly deserves attention, but in my opinion not robust enough to be asserted in the way it is in the title of the paper (at least with what is currently shown). The discussion around the cause can also be enhanced.*

Again as it is mentioned in the previous comment, the statement "Tw increases faster than Ta" is mainly based on the reach by reach assessment i.e. Figures 5 and 6 in the revised manuscript not Figure 4 which only compares median and spatial variability of trends in Tw and Ta. Moreover, as we discussed in the section 5.1 in the revised manuscript, in fall, we have a good coherence between Tw and Ta trends at the stations with long-term observed data (>20 years) and only two stations with lower data (8-13 years) show discrepancy between trends in observed and simulated Tw. Therefore, poor correlation in fall can be due to insufficient Tw data at these two stations.

Therefore, we believe that our statement is still valid for fall.

*2.3 HER, Strahler order and riparian vegetation*

*The study is complemented by an assessment of the influence of HERs, Strahler order and riparian vegetation on Tw. I have already raised some concerns about HERs (how they are accounted for in the models), and about riparian vegetation (the model is not validated for this). These two points need to be addressed in order to present the analysis.*

Responses to previous comments clearly stated that HER are not a conditioning/explaining factor and that the riparian vegetation influence has already been validated by Loicq et al. (2018).

*In general, all these topics are interesting, but I have the impression that they are treated only superficially and not with the necessary rigour.Moreover, there are already a lot of results presented and I think the article would still be interesting, and perhaps easier to read, if these parts were removed.*

We would like the reviewer to suggest a more "rigorous" approach than the one we used, which is rather basic and robust, derived only from model outputs and classes of vegetation and stream order. We therefore strongly disagree with the reviewer statement on a lack of rigour.

*Regarding the Strahler order analysis, this brings back to the issue of the reach-to-reach heat advection mechanism that is missing in the model description. It is answered previously. But, there is upstream-downstream thermal propagation in the model.*

This comment has already been answered previously. There is an upstream-downstream propagation of thermal signal. Please see page 5 of Beaufort et al. (2016). Please also see P5L120-126 in the revised manuscript.

*Also, showing a Strahler order map for each reach in the SM would be really informative.*

As mentioned in the previous comments, we agree. Please see Figure S3 in the revised manuscript.

*As mentioned in P22L397-398, the correlation between Strahler order and Tw may be due to riparian vegetation (and not a concentration of the warming when going downstream). However, this should be analysed in more details, e.g., by looking at the correlation between Strahler order and Tw separately for different shading factor values (e.g., the categories of Figure 10).*

This comment is somewhat contradictory with the previous comment relative to the lack of rigour suggested by the reviewer on this topic. Moreover, It was already shown by other studies that riparian shading influence is more important on small rivers and has no influence on large

rivers (e.g., Moor et al., 2005, Loicq et al., 2018). Of course, additional analyses are always possible, and we would appreciate any help on this issue for further assessments.

*For riparian vegetation, do you have a mechanism for why it would change the trend in Tw? I understand that adding or removing vegetation would change the absolute value of temperature, but by what mechanism would the trends be affected? If the riparian temperature sections were to be retained, this should be addressed.*

We believe that the reviewer misunderstood this part of the analysis (which may have led to his previous comments) and we'll make sure to make the revised version clearer. In this analysis, we compare Tw trends on headwater streams (in the same HERs which implies same type of geology, proxy of groundwater contribution) with different degrees of shading, resulting from vegetation and reach orientation. It is observed that effect of increase on radiative fluxes are less important on streams with high shading, creating resilience to warming for these type of rivers.

*To conclude this second part, I would recommend that the authors revise and strengthen the analysis of Ta vs Tw trends and the impact of Q. With the results presented, such strong statements as those in the title and abstract do not entirely hold water. I would also recommend mentioning in the title the location where the study was conducted (Loire, France). Perhaps some of my reservations stem from a lack of understanding of exactly how the models work and a better description might help to alleviate these reservations.*

The reviewer is right and we hope that our responses and proposed modifications to the revised manuscript put some light on the understanding of the models.

*Along with the emphasis on the main message that I recommend, perhaps some of the "secondary analyses" could be removed from the document. I think that a more thorough discussion of the limitations of the methods and results should be provided. I have not commented in detail on all the discussion, summary and conclusion sections, but certainly some of them are relevant to my comments above.*

We believe that the previous responses address these comments and that the analysis made on e.g. riparian vegetation may be interesting to readers as they suggest that increasing riparian vegetation might help reducing the impact of climate change on stream temperature (of course, independently on other consequences, notably on water use by such riparian vegetation).

*A final comment concerns the real added value of such a major modelling effort. A significant part of France is modelled; however, I feel that some of the results could be obtained by analysing only past measurements (and getting rid of all the modelling uncertainty).*

This comment is rather surprising for two reasons. First, the manuscript clearly states that too few Tw observations are available for assessing long-term trends over the whole Loire basin. Second, the reviewer is the main author of a HESSD preprint precisely using physically-based models to assess the impact of climate change in Switzerland (Michel et al., 2021, https://doi.org/10.5194/hess-2021-194) and should understand why such a modelling effort may be needed, i.e. to infer future changes in Tw thanks to such models and climate projections (which is precisely the objective of an upcoming manuscript).

*The added value could come from the whole analysis of the riparian vegetation for example, but as mentioned these analyses need to be strengthened. One solution (but which would involve a lot of extra work), could be to first publish just the data set in a journal like ESSE (https://www.earth-system-science-data.net/), where all the modelling and validation aspects are discussed in detail, and then have an article in HESS focused on the data analysis only allowing for an in-depth analysis. This would also allow for elaboration on some aspects that are not yet discussed (e.g., elevation, impact of snow if relevant for this catchment, more detailed spatial analysis). I think this option would really increase the potential impact of the significant modelling effort that has been made. But I would understand if the authoring team do not want to go through this extra work.*

We really thanks the reviewer for these suggestions, and we will take theme into account for further analyses, as this goes much further than the objective of the present manuscript.

*Minor comments:*

We agree with all comments except with ones to which we responded.

*P1L38 [Q] should be (Q)*

*P1L52-52 "in the face of a changing climate", maybe just say "to climate change"*

*P3L79-80 Maybe define exactly what HERs are, or give a reference*
*Figure 1 Add HERs "borders" also in left panel. In the manuscript, this figure is not vectorized and small text are really pixelized when zooming to reads them. Maybe provide a vectorized figure or a higher resolution bitmap figure. Add in the figure or caption the source for the maps shown.*

Please see Figure 1 in the revised manuscript.

**P5L101** *"bottom" should be "right"*

**P5EQ(5)** *Can't it be written in the more compact form max(SFLeft,SFRight)?*

**P8L200** *Why using log(Q) in this analysis?*

There was a mistake in writing. It was modified. It is (Q).

**P9L224** *What does IQR mean?*

**P9L226** *"r" is used here, while "R" is used in figures*

**Figure 2** *There is a "a" on the right below the colour legend*

**P11L267-269** *Please add a reference here to support this statement*

**P16L306-308** *How negative Q trends, just by themselves, suggest an effect on Tw?*
*"suggested" should be "suggesting".*
It was removed in the revised manuscript.

**P20L335-338** *First, despite it is clearly stated on the abstract of the paper Michel (2020) that: "The mean trends for the last 20 years are + 0.37 ± 0.11 ◦ C per decade for water temperature, resulting from the joint effects of trends in air temperature (+0.39 ± 0.14 ◦C per decade), discharge (−10.1 ± 4.6 % per decade), and precipitation (−9.3±3.4% per decade)", I think now that this paper does not show a real impact of Q on Tw, but rather a correlation between Q and Tw in summer. If I had to rewrite this paper today, I would not be so categorical (and this why I also question it in your paper). Second, when you say: "In contrast with our results, they found Tw trends lower than Ta trends due to influence of snow melt and glacier melt", this is not totally exact. Indeed, trends found in Alpine catchments are lower due to the mentioned effects. For the low-altitude catchments where snow plays no role, trends are indeed closer, but Tw trends remain slightly slower than Ta trends (compare Figures S17 and S18). However, on an annual basis, we are talking about a few tenths of a degree less in my article and a few tens of a degree more in yours, so taking into account all the uncertainties involved, I see no contradiction. In addition, different regions are studied.*

This comment is again rather surprising as the paper referred to by the reviewer (and signed by him) has been published quite recently. We clearly agree that the statement made in the referred paper is rather categorical, but the present manuscript conclusions are clearly much less categorical. Hence, our surprise is with respect to the comments on our main statement on Tw trends being stronger than Ta trends over the Loire basin.

Moreover, please note that the difference between Tw and Ta trends is not a few tens of a degree as the reviewer mentioned. Figure 5 in the revised manuscript shows that such difference can go up to 0.38 °C (i.e. up to 2.2°C over the whole 1963–2019 period).

**Introduction and Section 5.5** *Nuclear plants cooling is never mentioned in the paper. This might not be relevant for the Loire (but Bustillo et al. (2014) mention some plants in the catchment), but in general in France the question of cooling nuclear plants in the future with increasing air and water temperature will be a real challenge and I think it is worth mentioning it (see e.g. Bourqui et al., 2011).*

Please see P23L459-462 in the revised manuscript.

**P22L398** Shouldn't it be "small rivers"?

**References:**

Beaufort, A., Curie, F., Moatar, F., Ducharne, A., Melin, E. and Thiery, D., 2016. T-NET, a dynamic model for simulating daily stream temperature at the regional scale based on a network topology. *Hydrological Processes*, *30*(13), pp.2196-2210. https://doi.org/10.1002/hyp.10787.

Beaufort, A., Moatar, F., Curie, F., Ducharne, A., Bustillo, V. and Thiéry, D., 2016. River temperature modelling by Strahler order at the regional scale in the Loire River basin, France. *River Research and Applications*, *32*(4), pp.597-609. https://doi.org/10.1002/rra.2888.

Loicq, P., Moatar, F., Jullian, Y., Dugdale, S.J. and Hannah, D.M., 2018. Improving representation of riparian vegetation shading in a regional stream temperature model using LiDAR data. *Science of the total environment*, *624*, pp.480-490. https://doi.org/10.1016/j.scitotenv.2017.12.129.

Michel, A., Schaefli, B., Wever, N., Zekollari, H., Lehning, M., and Huwald, H.: Future water temperature of rivers in Switzerland under climate change investigated with physics-based models, *Hydrol. Earth Syst. Sci. Discuss.* [preprint], https://doi.org/10.5194/hess-2021-194, in review, 2021.

Wasson, J.G., Chandesris, A., Pella, H. and Blanc, L., 2002. Typology and reference conditions for surface water bodies in France: the hydro-ecoregion approach. *TemaNord*, *566*, pp.37-41. https://hal.archives-ouvertes.fr/hal-00475620/document